# Immune selection determines tumor antigenicity and influences response to checkpoint inhibitors

Luis Zapata [1] ✉, Giulio Caravagna [1,2], Marc J. Williams [3], Eszter Lakatos [4], Khalid AbdulJabbar[1], Benjamin Werner [4], Diego Chowell[5,6], Chela James[1,7], Lucie Gourmet[8], Salvatore Milite[7], Ahmet Acar[9], Nadeem Riaz [10], Timothy A. Chan [11], Trevor A. Graham [1,4] ✉ & Andrea Sottoriva [1,7] ✉

In cancer, evolutionary forces select for clones that evade the immune system. Here we analyzed >10,000 primary tumors and 356 immune-checkpoint-treated metastases using immune dN/dS, the ratio of nonsynonymous to synonymous mutations in the immunopeptidome, to measure immune selection in cohorts and individuals. We classified tumors as immune edited when antigenic mutations were removed by negative selection and immune escaped when antigenicity was covered up by aberrant immune modulation. Only in immune-edited tumors was immune predation linked to CD8 T cell infiltration. Immune-escaped metastases experienced the best response to immunotherapy, whereas immune-edited patients did not benefit, suggesting a preexisting resistance mechanism. Similarly, in a longitudinal cohort, nivolumab treatment removes neoantigens exclusively in the immunopeptidome of nonimmune-edited patients, the group with the best overall survival response. Our work uses dN/dS to differentiate between immune-edited and immune-escaped tumors, measuring potential antigenicity and ultimately helping predict response to treatment.

The immune system shapes tumor genomes by selecting for neoantigen-depleted clones (immune edited) or clones with an immune evasion strategy that allows neoantigen accumulation ('immune escaped')[1–3]. Immune checkpoint inhibitors (ICIs) work by reactivating immune predation against malignant cells by removing the 'invisibility cloak' provided by overexpression of immune checkpoint pathways, such as PD1 and CTLA-4. ICIs have been widely applied to treat cancer, especially in melanoma, where studies show an extraordinary 30% objective response rate[4]. However, low response rates for some tumor types and the highly toxic side effects of costly ICI treatments have

[1]Centre for Evolution and Cancer, The Institute of Cancer Research, London, UK. [2]Cancer Data Science Laboratory, Dipartimento di Matematica e Geoscienze, Università degli Studi di Trieste, Trieste, Italy. [3]Computational Oncology, Department of Epidemiology and Biostatistics, Memorial Sloan 10 Kettering Cancer Center, New York, NY, USA. [4]Centre for Genomics and Computational Biology, Barts Cancer Institute, Barts and the London School of Medicine and Dentistry, Queen Mary University of London, London, UK. [5]The Marc and Jennifer Lipschultz Precision Immunology Institute, Icahn School of Medicine at Mount Sinai, New York, NY, USA. [6]Department of Oncological Sciences, Tisch Cancer Institute, Icahn School of Medicine at Mount Sinai, New York, NY, USA. [7]Computational Biology Research Centre, Human Technopole, Milan, Italy. [8]UCL Genetics Institute, Department of Genetics, Evolution and Environment, University College London, London, UK. [9]Department of Biological Sciences, Middle East Technical University, Universiteler Mah, Ankara, Turkey. [10]Department of Radiation Oncology, Memorial Sloan Kettering Cancer Center, New York, NY, USA. [11]Center for Immunotherapy and Precision Immuno-Oncology, Cleveland Clinic, Cleveland, OH, USA. ✉e-mail: luis.zapata@icr.ac.uk; trevor.graham@icr.ac.uk; andrea.sottoriva@fht.org

fueled the search for better predictive biomarkers. To date, the US Food and Drug Administration-approved biomarkers are tumor mutation burden (TMB), microsatellite instability (MSI) and PDL1 expression. However, TMB has technical limitations, including low predictive power for some tumors, the absence of a universal threshold to predict response and a strong dependency on the sequencing technology and depth[5–8]. MSI-associated response and PDL-1 expression have also been challenged, as microsatellite stable (MSS) and PDL-1-negative patients may also display clinical benefit upon ICI treatment[9,10]. As these metrics neglect the underlying tumor evolutionary dynamics, we hypothesize that stratifying patients based on immune selection will improve patient management.

An evolutionary metric[11,12] commonly used to detect selection in cancer studies is the nonsynonymous to synonymous mutations ratio, dN/dS[13–17]. dN/dS has been used to detect driver genes[18], measure selective coefficients at different clone sizes[19] and show positive selection during subclonal expansions[20,21]. As nonsynonymous mutations can also generate neoantigens by transforming self-peptides, which do not elicit an immune response due to central tolerance[22], to non-self-peptides, which can potentially initiate an immune reaction, we hypothesized that immune selection can be measured by calculating dN/dS on the self-immunopeptidome[13]. The self-immunopeptidome can be defined as all genomic regions that generate peptides natively exposed to the immune system through the individual major histocompatibility complex (MHC). Despite wealthy literature on immune selection against neoantigens[23–27], few studies have challenged the application of MHC-based predictions to detect immune selection[28], raising the important question whether negative selection is truly absent[29], inefficient during somatic evolution[30] or computational predictions of MHC-biding peptides are poor[31]. Beyond these possibilities, the impact of immune evasion on immune selection signals remains unexplored.

Here, we calculate dN/dS inside the immunopeptidome, immune dN/dS, to validate the association between immune selection and T cell infiltration in primary immune-edited tumors. We demonstrate that immune escape enables neutral neoantigen accumulation, ultimately masking selection on cohort estimates. We hypothesize and demonstrate that immune-edited tumors do not respond to immunotherapies by analyzing 356 immunotherapy-treated metastatic cancers. Finally, in a longitudinal set of ICI-treated metastatic tumors, we show that immune dN/dS in combination with escape status predicts ICI response better than clonal TMB. Our study highlights the importance of considering tumor evolutionary dynamics for the future of personalized medicine.

## Results

### Signals of immune selection in mixed primary tumor cohorts

To measure selection using genomic data, we developed SOPRANO (selection on protein annotated regions), an algorithm that calculates trinucleotide context corrected dN/dS inside (ON target or ON-dN/dS) and outside (OFF target or OFF-dN/dS) any target genomic region. SOPRANO allows for cohort- and patient-specific dN/dS enabling comparisons between multiple or single individuals (Fig. 1a). Our method uses point mutations (missense and/or truncating single-nucleotide variants) in target immunopeptidomes made of (1) a single HLA allele (that is, HLA-A0201), (2) a proto-HLA allele consisting of 6 HLA haplotypes or (3) a patient's specific six HLA class I alleles (Fig. 1b). We applied SOPRANO in different settings to quantify immune selection (Fig. 1c and Table 1) and determine the impact of immune evasion (Fig. 1d).

First, we applied SOPRANO to 33 cancer types from The Cancer Genome Atlas (TCGA) (Supplementary Tables 1 and 2 and Supplementary Fig. 1a–c) using two immunopeptidomes: HLA-A0201, the most common allele, and a proto-HLA[28] composed of the six most frequent HLA alleles (HLA-A01:01, HLA-A02:01 HLA-B07:02, HLA-B08:01, HLA-C07:01 and HLA-C07:02). We used the ratio between ON-dN/dS and

OFF-dN/dS (ON/OFF) to conservatively demonstrate immune-specific selection and defined it as immune dN/dS. Bladder cancer (BLCA; 95% confidence interval (CI), 0.71–0.98), lung adenocarcinoma (LUAD; 95% CI, 0.61–0.84), lung squamous cell carcinoma (LUSC; 95% CI, 0.70–0.98), melanoma (SKCM; 95% CI, 0.75–0.92) and uterine corpus and endometrial carcinoma (UCEC; 95% CI, 0.85–0.98) displayed significant nonsynonymous depletion inside the HLA-A0201 immunopeptidome (Fig. 2a), and head and neck squamous cell carcinoma (HNSC; 95% CI, 0.85–0.95), cervical squamous cell carcinoma (CESC; 95% CI, 0.88–0.99), LUAD (95% CI, 0.84–0.93) and LUSC (95% CI, 0.86–0.95) inside the proto-HLA (Fig. 2b). Edited neoantigens varied from 0 to 151 (Supplementary Table 1) when calculated cohort-wise inside the HLA-A0201 immunopeptidome and from 0 to 2113 when calculated per patient, suggesting that cohort estimates may mask immune selection at the individual level. UCEC (four) and colorectal cancer (CRC) (two) were the tumor types with the highest number of edited neoantigens per patient. We confirmed the robustness of our findings by resampling mutations and individuals (Supplementary Fig. 1d,e) and by estimating immune dN/dS with mutations from different somatic callers (Supplementary Fig. 1f). Overall, results were consistent between different callers and, thus, we further used only MuTect2 calls. Cholangiocarcinoma, mesothelioma and esophagus carcinoma had also traces of immune selection but with large CIs due to low mutation burden (Supplementary Fig. 1c).

Next, we compared cohort immune selection between SOPRANO immune dN/dS and a published HLA-binding mutation ratio (HBMR)[28]. There was a significant correlation between immune dN/dS and HBMR for the proto-HLA (Pearson $r$ or $r$ = 0.77, $P$ = 0.00054; Fig. 2c), but not when comparing HLA-A0201 to the proto-HLA immunopeptidome ($r$ = 0.37, $P$ = 0.15; Supplementary Fig. 1g). The latter correlation was expectedly lower given that proto-HLA includes a larger genomic region where immune selection might not be acting for patients without those HLA alleles. Both metrics used similar burden of nonsynonymous mutations inside (Supplementary Fig. 1h, $r$ = 0.93) and outside the immunopeptidome (Supplementary Fig. 1i, $r$ = 0.98). When exploring alternative immunopeptidomes based on HLA-A0201 patients, immune selection was weaker (immune dN/dS ~ 1) when including unexpressed regions or weak binders, whereas in both cases, OFF-dN/dS remained the same. Further immunopeptidome benchmarking revealed the strongest immune selection when using patient-specific data (Supplementary Fig 2). Overall, these results suggest that immune selection discrepancies can arise due to different immunopeptidomes, especially if including unexpressed or false MHC-binding peptides.

### Immune escape abrogates CD8 T cell-mediated immune selection

We hypothesized that weak immune selection is due to immune-escaped tumors masking the signal. We compared the association between immune selection, measured by immune dN/dS and HBMR, and immune infiltrates when including or excluding three tumor types with a high frequency of microsatellite-unstable (MSI) cases, and therefore a high frequency of evasion mechanisms: CRC, stomach adenocarcinoma (STAD) and UCEC[32]. To minimize possible biases from bulk cell deconvolution methods, we obtained immune infiltration estimates from three different TCGA studies[26,33,34]. When comparing all tumor types with HBMR available, median CD8 T cell abundance score was negatively correlated with immune dN/dS in the HLA-A0201 immunopeptidome ($r$ = −0.7, $P$ = 0.0035; Fig. 2d), but not to HBMR in the proto-HLA ($r$ = −0.38, $P$ = 0.16; Fig. 2e). When excluding CRC, STAD and UCEC, immune dN/dS and CD8 T cells correlation increased significance for both immunopeptidomes, supporting our hypothesis of immune escape masking immune selection (HLA-A0201: $r$ = −0.78, $P$ = 0.0017, Fig. 2f; proto-HLA: $r$ = −0.61, $P$ = 0.028, Fig. 2g). Moreover, similar results were observed when using alternative immune estimates such as cytolytic activity (Supplementary Fig. 3a), CD8 T cell infiltration

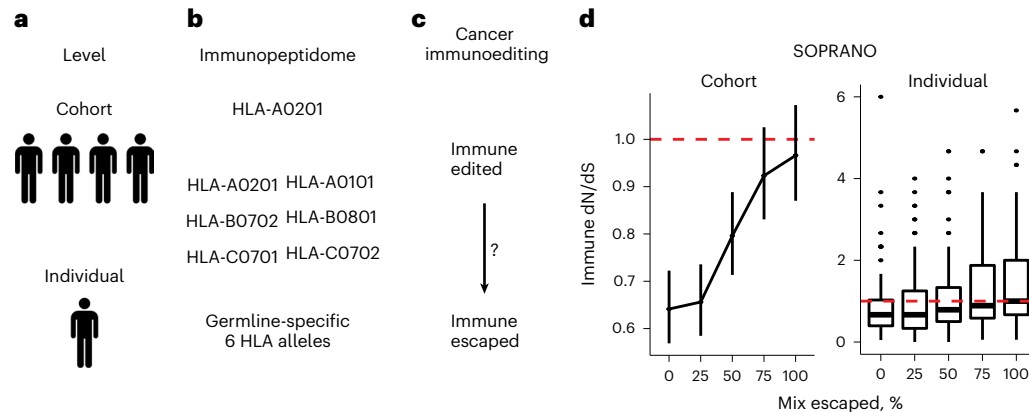

**Fig. 1 | Overview of immune selection calculation using SORANO.**
**a**, Estimates can be performed at the cohort or at the single individual level.
**b**, In each case, it is possible to estimate immune selection on a single HLA allele (that is, HLA-A0201), a generic combination of HLA alleles (proto-HLA) or the germline-specific HLA-immunopeptidome. **c**, Immune selection determines the evolutionary trajectories of clonal growth; fully immune-edited tumors with strong immune selection signals can transit towards fully immune-escaped

tumors where signals are absent. **d**, Toy model of mixing immune-edited and immune-escaped tumors. It is possible to estimate immune dN/dS by aggregating all mutations and generate a single cohort estimate or to estimate a distribution of values per patient. In both cases, we hypothesize that mixing escaped with edited tumors leads to loss of immune selection signals reflected by immune dN/dS values closer to one (depicted as red dashed lines in the figure).

## Table 1 | Analysis of tumor datasets with SOPRANO

| Dataset | Tumor type | No. of individuals | Immunopeptidome |
|---|---|---|---|
| TCGA | 33 tumor types | 10,172 | HLA-A0201, proto-HLA, patient specific |
| Lakatos | CRC, STAD, UCEC | 879 | Patient specific |
| Hartwig | Multiple | 308 | HLA-A0201, patient specific |
| Riaz | Melanoma | 68 | Patient specific |

Four datasets were analyzed in this study, including a large cohort of 33 primary untreated tumor types obtained from TCGA; a cohort of primary untreated CRC, STAD and UCEC from TCGA with curated escaped mechanisms (Lakatos); an immunotherapy-treated metastatic tumor cohort from the Hartwig Medical Foundation (Hartwig); and a metastatic melanoma tumor cohort with genomic data pre and during immunotherapy (Riaz). The analysis was performed using different immunopeptidomes as described in the paper.

from a different study[33] (Supplementary Fig. 3b) and lymphocyte infiltration score[34] (Supplementary Fig. 3c), but not when comparing to TMB (Supplementary Fig. 3d). When including all 33 tumor types, we also confirmed that immune infiltration correlated with ON-dN/dS and immune dN/dS, but not with OFF-dN/dS (Supplementary Fig. 4a–c).

We then run SOPRANO on a subset of 6,858 primary untreated tumors from TCGA with at least a single mutation in the immunopeptidome using each patient's six HLA alleles, and we classified each individual into escaped (escaped+) or non-escaped (escaped−) based on a missense or a truncating mutation in a preselected 'escape' gene associated to the antigen presenting machinery (Supplementary Tables 3–5). Among 88 preselected escape genes, we found significant positive selection on missense mutations in 40 genes, including *HLA-B*, *B2M*, *IFNG* and KIR-like genes, and on truncating mutations in eight genes, including *B2M*, *HLA-A*, *B* and *C* (Fig. 2h and Supplementary Table 6). Next, we compared immune dN/dS and mutation burden between escape strata for different immune categories[34]. Although patient-specific immune dN/dS and CD8 T cell infiltration were not correlated, we found significantly higher TMB in escaped+ tumors in all categories (Supplementary Fig. 4d), while immune dN/dS was significantly higher for escaped+ tumors in C1 and C2 categories, characterized by high proliferation and high intratumor heterogeneity, and was not different in C3, characterized by low proliferation, and C4, characterized by lymphocyte absence (Supplementary Fig. 4e).

Additionally, escaped+ harbored more splice, stop-loss and nonsense mutations (Supplementary Fig. 5a) and more amplifications and deletions (data obtained from a previous study[35]; Supplementary Fig. 5b) than escaped− tumors. There was a significant correlation between immune dN/dS distance to neutrality and TMB in escaped+ ($P = 2.4e\text{-}12$) but not in escaped− tumors (Supplementary Fig. 5c), supporting the hypothesis that neoantigenic mutations accrue neutrally in escaped+ tumors. Intriguingly, such effect was not observed when deletions or amplifications in escape genes were considered (Supplementary Fig. 5d). Importantly, mutation burden increase was specifically due to missense or truncating mutations and not to synonymous events on the same escape genes (Supplementary Fig. 6).

To determine which immune subpopulations were associated to immune selection, we applied a linear mixed model to determine their contribution to immune-, ON- and OFF-dN/dS, including (Supplementary Fig. 7a) or excluding CRC, STAD and UCEC (Fig. 2i). The best-performing model for predicting immune dN/dS ($R^2$adj = 0.89, Akaike information criterion (AIC) = −83, $P = 0.01$) had CD8 T cells as the most significant explanatory variable. Importantly, no variable could explain OFF-dN/dS, and seven out of ten immune variables tested were significantly associated to ON-dN/dS. Moreover, when expanding to more tumor types with available immune cell scores, we found a significant association between immune dN/dS and leukocyte infiltration in escaped− tumors (Supplementary Fig. 7b, no truncating mutations: $P = 0.007$; Supplementary Fig. 7c, no missense/truncating: $P = 0.011$), but not in escaped+ tumors (Supplementary Fig. 7d, missense/truncating: $P = 0.58$; Supplementary Fig. 7e, truncating: $P = 0.25$). These results were confirmed in a multivariate model when controlling for stromal fraction (Supplementary Fig. 7f) or including multiple other cellular states (Supplementary Fig. 7g). Interestingly, PD1 and PDL1 expression were associated to immune selection in a univariate analysis (Supplementary Fig. 7h,i PD1 $P = 0.021$, PDL1 $P = 0.057$), but not in a linear mixed model that included CD8 T cells (Supplementary Fig. 7j,k), highlighting CD8 T cell lymphocytes as the main driver of immune selection.

### MSS escaped tumors show neutral immune dN/dS

To further characterize genetic differences between escaped and edited tumors, we focused on 879 previously curated primary untreated CRC, STAD and UCEC[32]. These tumors had multiple annotated escape mechanisms, including immune checkpoint inhibition derived from RNA sequencing, loss of heterozygosity of HLA genes and copy number

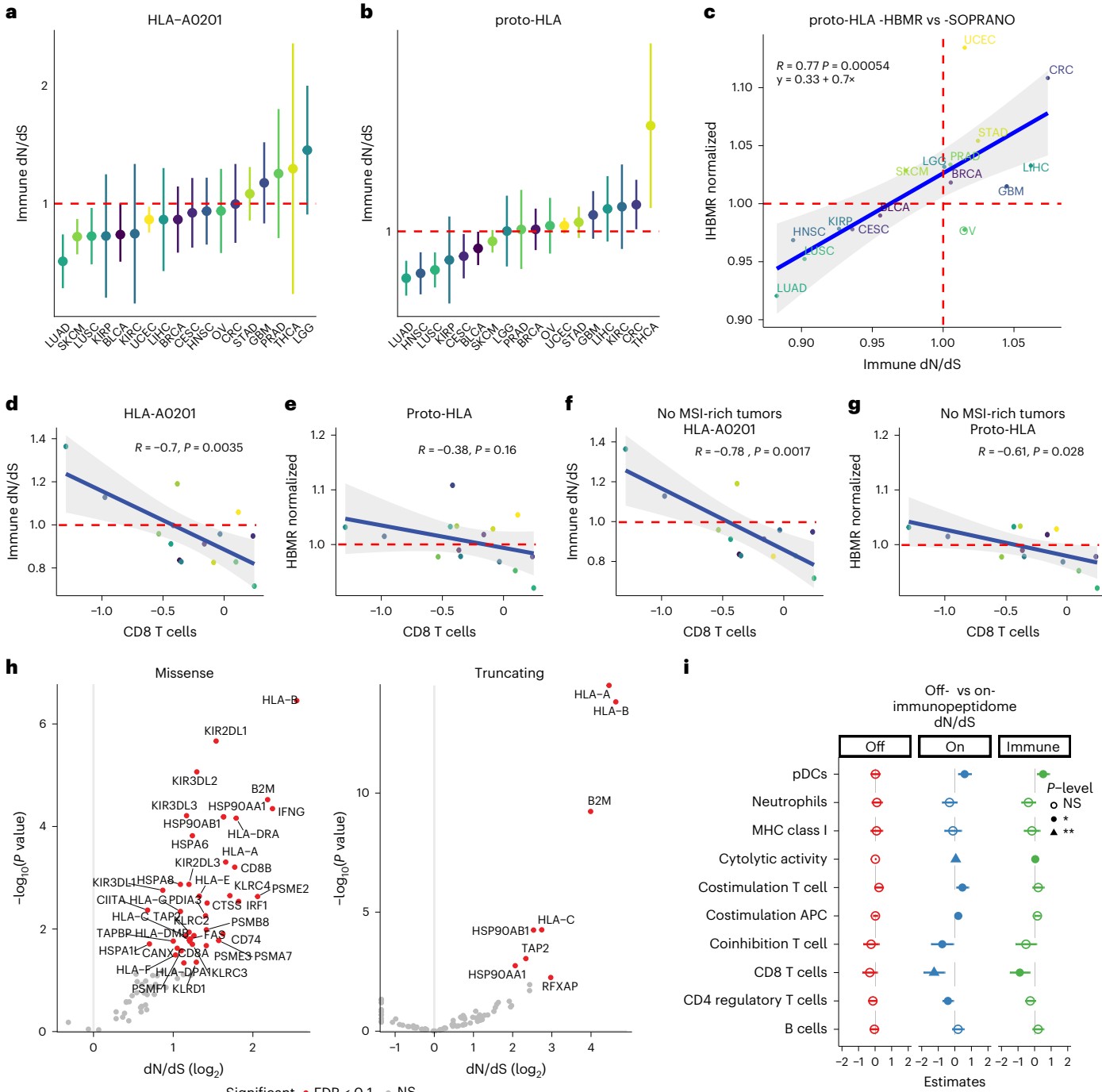

**Fig. 2 | Immune dN/dS landscape across multiple tumor types. a,b,** Immune dN/dS (ON-dN/dS/OFF-dN/dS ratio) in multiple tumor types using either a curated HLA-A0201 based immunopeptidome (**a**) or a proto-HLA consisting of the most common HLA haplotypes in the population (**b**). Error bars indicate 95% confidence interval to the point estimate obtained with SOPRANO and number of samples for each tumor type is described in Supplementary Table 1. **c,** Proto-HLA immune dN/dS from SOPRANO versus proto-HLA normalized HBMR reported previously[28]. **d,e,** Immune dN/dS on HLA-A0201 (**d**) and HBMR on the proto-HLA versus median CD8 T cell infiltration, including microsatellite-unstable (MSI)-rich tumors (**e**). **f,g,** Immune dN/dS on HLA-A0201 (**f**) and HBMR on the proto-HLA versus median CD8 T cell infiltration, excluding MSI-rich tumors (**g**). We assumed that MSI-rich tumors are also escape-rich tumors. *P* values and correlation coefficients were calculated using Pearson's correlation (two-sided *t*-test). Gray

shaded areas represent error bands indicating the 95% confidence interval. Red dashed lines indicate neutral dN/dS at one. **h,** log₂ dN/dS versus -log₁₀(*P* value) of selected escape genes (Supplementary Table 3) using missense and truncating mutations. **i,** Linear mixed model using dN/dS as the dependent variables and all immune metrics as independent variables. *x* axis shows the B coefficients. Model selection using the AIC revealed that immune dN/dS is strongly associated to the median abundance of CD8 T cells. No immune subpopulation was significantly associated to OFF-dN/dS. For ON-dN/dS, adjusted $R^2 = 0.927$, F-statistic = 18.8 on 10 and 4 degrees of freedom, $P = 0.00617$. For OFF-dN/dS, adjusted $R^2 = 0.898$, F-statistic = 13.3 on 10 and 4 degrees of freedom, $P = 0.0117$ (significance codes are described as '***' for $P < 0.001$, '**' $P < 0.01$, '*' $P < 0.05$, and NS, for not significant $P > 0.05$). APC, antigen presenting cell; FDR, false discovery rate; pDC, plasma dendritic cell. *R* indicates Pearson *r* correlation coefficient.

(CN) status for escape genes (Supplementary Fig. 8). In all three tumor types, TMB (Supplementary Fig. 9a) and immune dN/dS (Supplementary Fig. 9b) were significantly higher in MSI and *PolE* mutated (POLE)

subtypes compared to MSS tumors. When stratifying by escape status, escaped+ tumors had significantly higher TMB than escaped− tumors (MSS: $P = 0.0018$ and MSI: $P = 0.00027$, Fig. 3a), and an immune dN/dS

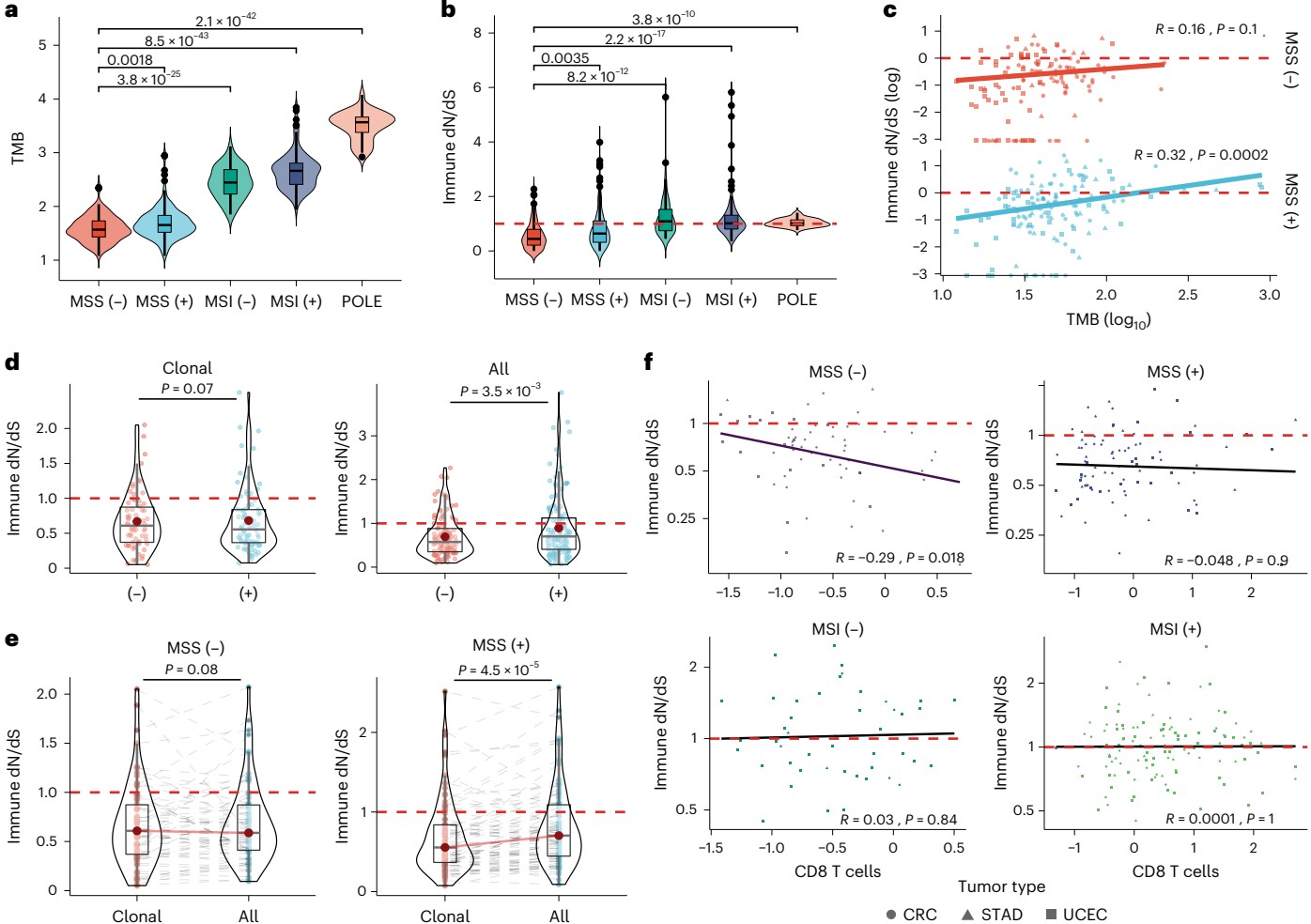

**Fig. 3 | Immune dN/dS analysis of a curated set of individuals from three tumor types.** Analysis of a curated set of individuals from three MSI-rich tumor types. Patient-specific analysis of primary untreated colorectal (CRC), stomach (STAD) and uterine/endometrial cancer (UCEC) (n = 879) with annotated escape mechanisms obtained from Lakatos et al.[32]. **a,b**, TMB (**a**) and immune dN/dS (**b**) for different subtypes of cancers, including MSS escaped− (MSS−, n = 130) and escaped+ (MSS+, n = 144), microsatellite-unstable escaped− (MSI−, n = 53) and escaped+ (MSI+, n = 125) and POLE mutants (n = 38). **c**, Immune dN/dS versus TMB for immune-escaped and immune-edited MSS groups using all mutations (MSS− n = 107, MSS + n = 133). **d**, Immune dN/dS comparison between escaped− and escaped+ tumors using clonal (MSS−, n = 93, MSS+, n = 94) or all mutations

(MSS−, n = 130, MSS+, n = 144). **e**, Immune dN/dS comparison between clonal versus all mutations in escaped− and escaped+ MSS tumors. Reported P values from paired two-samples two-sided Wilcoxon signed rank test after multiple test correction using Holm method. **f**, Relationship between immune dN/dS and the reported CD8 T cell infiltration for escaped− and escaped+ in MSS and MSI tumors. Boxplots represent the median, 25th percentile and 75th percentile, and whiskers correspond to 1.5 times the interquartile range. CRC, circle; STAD, triangle; UCEC, square. For two-sample comparisons, P values were calculated using a nonparametric two-sided Mann–Whitney U test. For linear correlations, P values and coefficients were calculated using Pearson's correlation (two-sided t-test). Red dashed lines indicate immune neutral dN/dS = 1.

closer to one (Fig. 3b), suggesting that most mutations post-escape accumulate neutrally.

To test whether antigenic mutations accumulate 'immune' neutrally after escape (absence of immune selection), we compared mutation burden and immune dN/dS. Indeed, only in MSS escaped+ tumors, TMB and immune dN/dS were significantly correlated (P = 0.0001; Fig. 3c), suggesting that immune selection was still active in MSS escaped− patients. To rule out that this association was driven by high-TMB escaped+ tumors, we exclude individuals with a mutation burden above the maximum TMB observed in MSS escaped− and confirmed the correlation considering all mutations (P = 0.0092; Supplementary Fig. 9c) and clonal mutations (P = 0.032; Supplementary Fig. 9d). Immune selection was not associated with TMB in escaped+ or escaped− MSI/POLE tumors (Supplementary Fig. 9e). This finding suggests that evasion mechanisms in these tumor subtypes develop early in carcinogenesis, leading tumors to be immune neutral, highly antigenic and thus potentially more responsive to immunotherapies.

Indeed, it is known that patients with MSI or POLE subtypes have the best clinical response rates to checkpoint inhibitors[36]. We corroborated these results using at least three synonymous mutations in the immunopeptidome to minimize the risk that patients were incorrectly classified as edited (P = 2.8 × 10⁻⁵; Supplementary Fig. 9f).

We next asked whether in MSS tumors clonal mutations hold immune selection signals whereas subclonal mutations do not, as they can accumulate freely when escape mechanisms are present. Clonal immune dN/dS was similarly lower than one for MSS escaped+ and escaped− tumors (Fig. 3d; 0.67 versus 0.68 immune dN/dS). When including subclonal mutations (that is all mutations), only escaped+ tumors had an immune dN/dS closer to one, whereas escaped− tumors retained immune selection signals (0.89 versus 0.69, P = 0.0035; Fig. 3d). To demonstrate that this effect was not due to high-TMB individuals from escaped+ tumors but rather driven by subclonal variants, we compared clonal versus all immune dN/dS within each patient. Escaped− tumors had similar immune dN/dS using all or clonal

mutations (all mutations: 0.61 versus clonal: 0.59 immune dN/dS, Fig. 3e), whereas escaped+ tumors had a significantly higher immune dN/dS when including subclonal mutations (0.70 versus 0.55 immune dN/dS, $P = 1.2 \times 10^{-4}$; Fig. 3e), demonstrating the temporal acquisition of immune-neutral mutations in escaped+ tumors, and providing evidence for sustained immune selection in escaped− tumors (immune-edited tumors).

To further validate that immune selection strength depends on immune surveillance, we compared the patient-specific immune dN/dS to CD8 T cell infiltration in escaped−, immune-edited tumors. Immune dN/dS and CD8 T cells were correlated in primary MSS immune-edited tumors but not in MSS immune-escaped tumors ($P = 0.018$; Fig. 3f and Supplementary Fig. 10a, or other CD8 metrics from Danaher et al.[33] ($P = 0.011$, Bonferroni $\alpha = 0.025$), Supplementary Fig. 10b), validating that the immunopeptidome holds CD8 T cell-mediated selective strength. Escape status in MSI and POLE tumors did not make a difference in immune infiltration, suggesting that besides a possible unknown escape mechanism, high mutation rate may mask weak negative selection, as recently proposed[30].

## Immune-edited tumors' resistance to immunotherapy

To address the clinical importance of immune evasion and immune dN/dS as a tumor antigenicity surrogate, we analyzed 308 metastatic cases subjected to immunotherapy with ICIs from the Hartwig Medical Foundation cohort[37] (Fig. 4a). Patients were sequenced before treatment and classified into complete response and partial response and into progressive disease or stable disease following RECIST1.1 guidelines. There were 79 responders recorded (partial response and complete response) and 229 nonresponders (progressive disease/stable disease). We estimated cohort- and patient-specific immune dN/dS using the HLA-A0201 immunopeptidome (Fig. 4b) or each individual's six HLA alleles (Fig. 4c), respectively. Nonresponders were strongly immune-edited (immune dN/dS < 1) compared to responders for both immunopeptidomes ($P = 0.034$). A total of 17 of 154 nonresponders had high-immune dN/dS (dN/dS > 2), putatively indicating positive selection in the immunopeptidome (Fig. 4c). However, our simulations suggest that high-immune dN/dS patients may artificially arise by carrying fewer synonymous mutations in the immunopeptidome than the rest of the cohort (Supplementary Fig. 11a) while having the same overall TMB (Supplementary Fig. 11b). Notably, immune dN/dS for nonresponders remained significantly lower than one when removing individuals with less than two, three or four synonymous mutations in the immunopeptidome respectively (Supplementary Fig. 11c).

Following our escape classification based on a list of escape genes (Supplementary Fig. 12), we found that responders were more frequently escaped than nonresponders (chi-squared $P = 8 \times 10^{-5}$; Fig. 4d). Responders had also a significantly higher TMB than nonresponders before treatment (Supplementary Fig. 13a; $P = 1.4 \times 10^{-4}$). Escaped+ tumors also had a significantly higher TMB compared to escaped− patients (Supplementary Fig. 13b; $P = 2.5 \times 10^{-18}$). However, within the escaped+ and escaped− group, TMB was not different between responders and nonresponders, suggesting that TMB is insufficient to predict clinical response when the escape status is considered (Fig. 4e). We hypothesized that immune dN/dS < 1 and absence of escape indicates strongly immune-edited tumors resistant to immunotherapies. Patient-specific immune dN/dS for responders and nonresponders separated by escape status were significantly different ($P = 0.008$), and escaped− nonresponders had immune dN/dS significantly lower than one when filtering out patients with zero (Fig. 4f; $P = 0.001$), one, two or three synonymous mutations in the immunopeptidome (Supplementary Fig. 13c).

To determine whether other dN/dS metrics were affected by our stratification, we restricted our cohort estimates to only driver genes (driver dN/dS), non-driver genes (global dN/dS), escape genes (escape dN/dS) and the immunopeptidome (immune dN/dS) in four patient

groups (Fig. 4g). Driver dN/dS was positive for escaped− and neutral for escaped+ tumors. Escape dN/dS was higher than one for escaped+, and expectedly zero for escaped− patients, given the absence of mutations in these genes. Global dN/dS was expectedly close to one for all groups as most somatic mutations in the genome are neutral[14]. Immune dN/dS was lower than 1 only for nonresponders/escaped− tumors, corroborating our hypothesis of primary resistance in immune-edited patients.

To evaluate the clinical importance of TMB, escape status and immune dN/dS, we investigated their impact on overall survival in ICI-treated patients. Initially, only escape status was significantly associated to clinical outcome (univariate Cox, TMB log-likelihood ratio test (LRT), $P = 0.51$, escape LRT $P = 0.006$, immune dN/dS LRT $P = 1$). Escaped+ patients displayed a significantly longer overall survival compared to escaped− patients ($P = 0.0023$; Supplementary Fig. 14a), especially in the nonresponder arm ($P = 1 \times 10^{-6}$; Fig. 4h). As we previously observed a nonresponder subgroup with high-immune dN/dS, we did not expect a linear association with clinical outcome. To account for this, we classified patients into immune low, neutral and high based on predefined immune dN/dS cutoffs (Methods) and removed high-immune dN/dS tumors unlikely to be genetically escaped (Supplementary Fig. 14b). Unedited escaped+ individuals had a significantly better overall survival compared to edited escaped− patients ($P = 0.0025$; Fig. 4i), validating these categories as predictive markers of ICI response. Alternatively, to avoid exclusion of patients and using predefined cut points, we estimated the absolute distance to neutrality (delta immune (d-immune) dN/dS). We showed that high-immune dN/dS individuals can be edited but have high dN/dS due to low mutation burden. In a multivariate cox regression analysis, escape status, age and d-immune dN/dS were significantly associated with overall survival, whereas TMB was not prognostic (Supplementary Fig. 14c). Importantly, escaped− high d-immune dN/dS individuals had the worst prognosis, whereas the escaped+ low d-immune dN/dS had the best prognosis (Supplementary Fig. 14d), further validating that unedited (immune-neutral) escaped+ tumors are the best candidates for ICI treatment. Remarkably, across different multivariate models including recently published metrics[38] (clonal and subclonal TMB, indel count, CXCL9 and CD274 expression), escape status and d-immune dN/dS remained as significant factors (AIC 448, log rank $P = 1 \times 10^{-6}$; Supplementary Fig. 14e).

## Immune dN/dS and escape predicts ICI response in melanoma

To further validate immune dN/dS as a predictive biomarker for immunotherapy, we analyzed a clinically annotated longitudinal cohort (Riaz cohort) of 68 metastatic melanoma patients[39], sequenced before (Pre) and during (ON) ICI therapy (Fig. 5a). We applied SOPRANO using each patient-specific HLA-I genotype to obtain immune dN/dS for 48 patients who had at least one synonymous mutation in the immunopeptidome (Supplementary Fig. 15a). We compared the selective pressure acting on the immunopeptidome Pre- and On-therapy for responders (R, complete responders or partial responders) and nonresponders (NR, stable disease or progressive disease). Treatment reduced mutation burden (Supplementary Fig. 15b) and immune dN/dS (Supplementary Fig. 15c) only in responders and only inside the immunopeptidome (ON $P = 0.024$, OFF $P = 0.95$), supporting a reduction in tumor volume directly linked to immune selection (Fig. 5b).

Next, we assessed whether immune dN/dS reflects tumor antigenicity in the context of response and escape in this longitudinal cohort. We classified patients based on the escape gene list plus loss of HLA heterozygosity and HLA germline status. A total of 42 out of the 88 escape genes were found mutated in 30 individuals, with most mutations being missense; 5 additional patients had loss of heterozygosity/homozygosity in the HLA region, resulting in 35 escaped+ patients versus 13 escaped− patients (Fig. 5c). Cohort immune dN/dS revealed that in Pre-treatment samples, escaped+ patients showed an average immune dN/dS of ~1 regardless of response, whereas escaped− patients

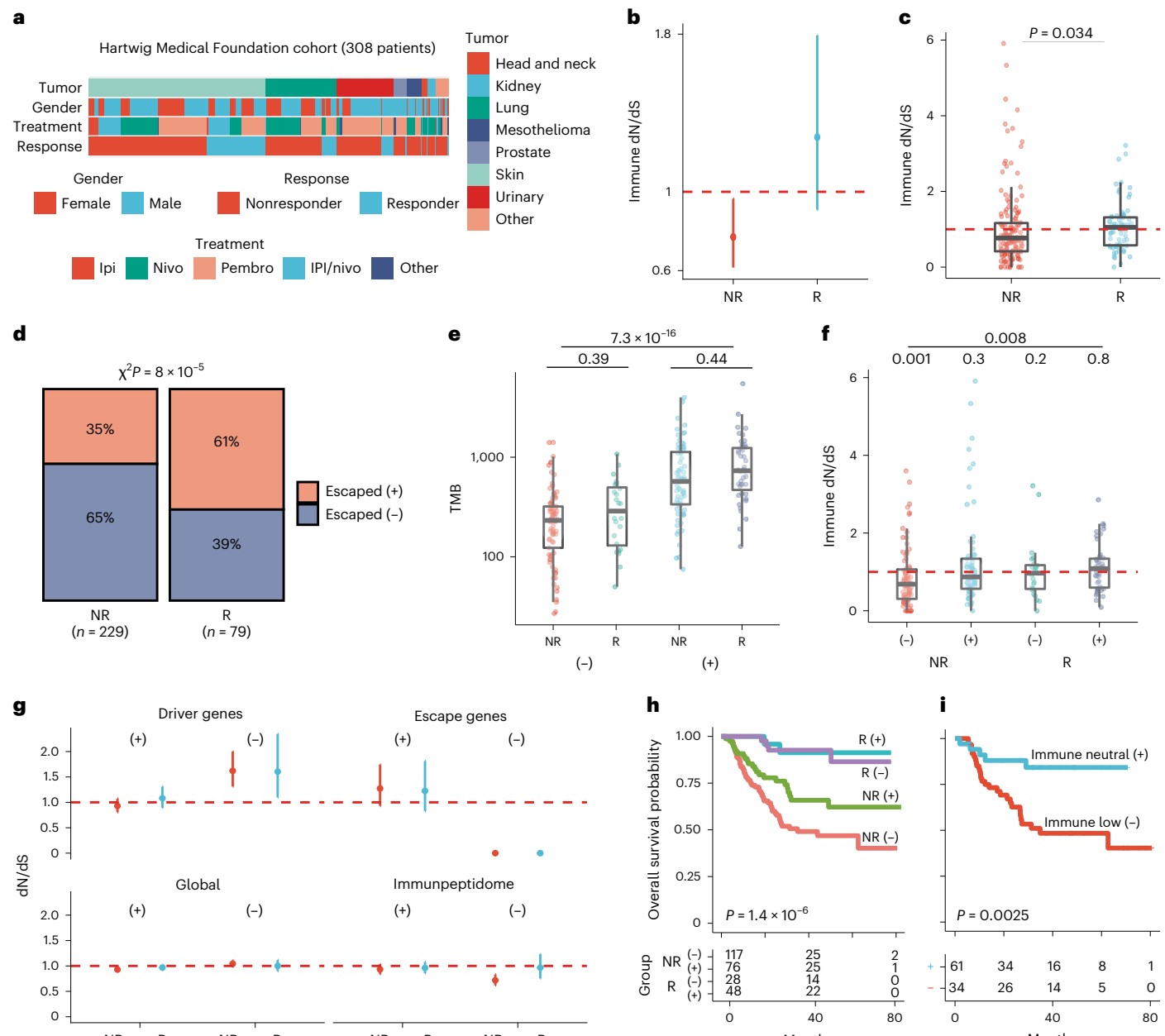

**Fig. 4 | Analysis of the Hartwig Medical Foundation metastatic cohort under immunotherapy. a**, A total of 308 patients with clinical response to immunotherapy based on RECIST1.1, including responders (R) and nonresponders (NR). Patients were primarily treated with ipilimumab (Ipi), nivolumab (Nivo), pembrolizumab (Pembro) or a combination of ipilimumab plus nivolumab (IPI/nivo). **b**, Cohort immune dN/dS for responders ($n = 79$) and non-responders ($n = 229$) using a common HLA-A0201 immunopeptidome reveal immune dN/dS lower than 1 for nonresponders consistent with a low overall tumor antigenicity. **c**, Comparison of individual immune dN/dS for responders (median = 1.05, $n = 67$) and nonresponders (median = 0.77, $n = 154$) using patient-specific HLA immunopeptidomes ($P = 0.034$). **d**, Proportion of escaped+ (NR $n = 81$, R $n = 48$) and escaped− (NR $n = 148$, R $n = 31$) tumors classified by clinical response. Responders were enriched in genetic escape mechanisms ($\chi^2 P = 8 \times 10^{-5}$). **e**, TMB for escaped (NR+ $n = 69$, R+ $n = 43$) and escaped− tumors (NR− $n = 85$, R− $n = 43$) classified by response status show that escaped+ tumors

have significantly more mutations than immune-edited tumors. **f**, Patient-specific immune dN/dS revealed a significant depletion of nonsynonymous mutations only in the immunopeptidome of escaped− nonresponders (NR−, median = 0.69, $P = 0.0012$). One-sample one-sided Wilcoxon signed rank test with mu = 1 (NR− $n = 85$, NR+ $n = 69$, R− $n = 24$, R+ $n = 43$). Boxplots represent the median, 25th percentile and 75th percentile, and whiskers correspond to 1.5 times the interquartile range. **g**, Cohort dN/dS for driver genes (196 genes from Martincorena et al.[14]), escape genes (Supplementary Table 3), all the exome (global dN/dS) and the immunopeptidome (immune dN/dS). All error bars include the point estimate plus the 95% confidence interval calculated using SOPRANO package. **h**, Kaplan–Meier curves of overall survival for responders and nonresponders grouped by escape status (log rank $P = 1.4 \times 10^{-6}$). **i**, Kaplan–Meier curves of overall survival for individuals classified based on immune dN/dS and escape status (−, escaped−; +, escaped+) (log rank $P = 0.0025$). Red dashed lines indicate immune-neutral dN/dS = 1.

who did not respond were immune edited (immune dN/dS = 0.6, 95 CI, 0.44, 0.82) (Fig. 5d). Importantly, only responder escaped+ patients showed immune dN/dS decrease during therapy, whereas the nonresponder escaped− patients sustained immune selection signals after

treatment (Supplementary Fig. 15b). To control if the observed immune selection was due to a bias on the mutation burden, we randomized the same number of genes found mutated in the cohort and recalculated immune dN/dS for escaped+ and escaped− 10,000 times. The observed

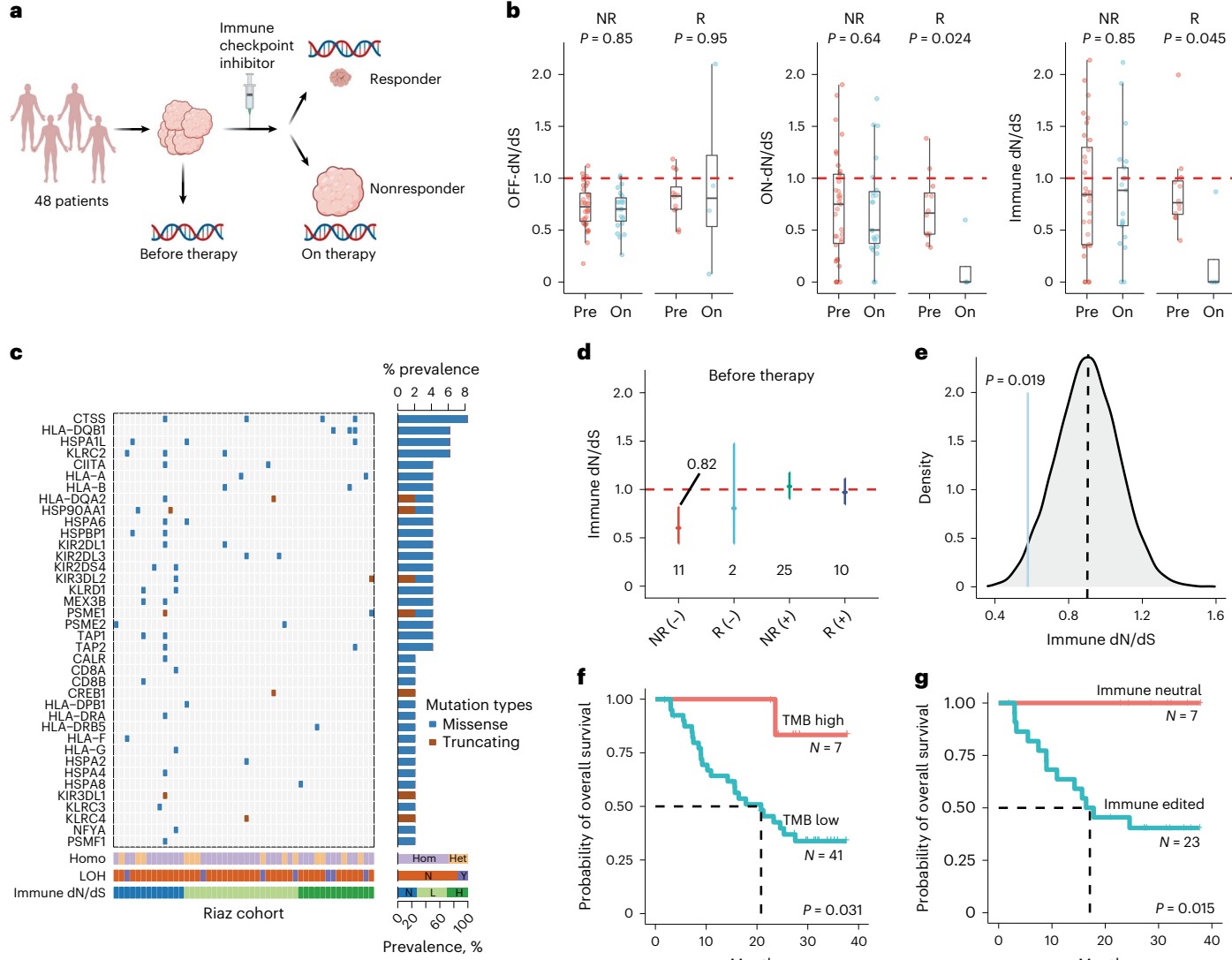

**Fig. 5 | Impact of immune dN/dS on outcome for ICI therapy in a metastatic melanoma cohort. a**, A clinically annotated cohort of 48 patients with sequencing data before (Pre) and after (On) receiving ICIs was obtained from Riaz et al.[39]. **b**, dN/dS distributions for nonresponders (n = 36) and responders (n = 12) before and after therapy. We estimated OFF-dN/dS (left), ON-dN/dS (middle) and immune dN/dS (right) using each patient's six HLA alleles. *P* values were calculated using two-sided Wilcoxon rank sum test and corrected using Benjamini-Hochberg. **c**, Mutations and their prevalence in genes classified as escaped in the Riaz cohort. Individuals were also classified according to: homozygosity status (Hom, homozygous; Het, heterozygous), somatic loss of HLA heterozygosity (N, no; Y, yes) and their immune dN/dS category (N, neutral; L, low; H, high). **d**, Pre-therapy immune dN/dS for escaped− and escaped+ cohorts classified as responders (R) and nonresponders (NR). Escaped−

nonresponders were the only group with cohort immune dN/dS less than 1. All escaped+ tumors before therapy displayed immune dN/dS equals to one. **e**, Immune dN/dS distribution for randomized escaped− tumors. The point estimate for nonresponders escaped− patients was significantly lower than the mean of randomized immune dN/dS values (exact *P* = 0.019). **f**, Kaplan–Meier curves of overall survival between high-TMB and low-TMB patients. High-TMB patients had significantly longer overall survival (log-rank *P* = 0.031) than low-TMB individuals. **g**, Kaplan–Meier curves of overall survival between immune neutral (escape+ and neutral immune dN/dS) and immune edited (escape− and low-immune dN/dS). The association between overall survival and immune dN/dS was more significant than with TMB (log-rank *P* = 0.015). Red dashed lines indicate immune-neutral dN/dS = 1.

immune dN/dS for escaped− patients was significantly lower than the randomized distribution (Fig. 5e; exact *P* = 0.019).

Finally, to test immune dN/dS predictive power independent of escape status, we compared overall survival using TMB and immune dN/dS before therapy as independent variables. As previously demonstrated in this cohort[39], high TMB showed significantly better response than low TMB patients (log rank *P* = 0.031; Fig. 5). We classified patients into immune dN/dS low (immune dN/dS < 0.82) and neutral (Fig. 5f). We also excluded immune-high tumors due to high confidence intervals (Supplementary Fig. 15e) and few synonymous mutations in the immunopeptidome (Supplementary Fig. 15f). Overall, we identified a

significant predictive difference between groups when including the immune-high group (log rank *P* = 0.026; Supplementary Fig. 16a) or when excluding it (log rank *P* = 0.015; Fig. 5g). Immune-neutral tumors showed the best response and low-immune dN/dS tumors had worst overall survival compared to neutral tumors. Additionally, d-immune dN/dS was also prognostic (Supplementary Fig. 16b). Immune-neutral tumors (low d-immune dN/dS) showed the best overall survival. A multivariate cox-hazard model with TMB, escape status and d-immune dN/dS demonstrated that immune-neutral patients had the lowest hazard ratio (HR = 0.25 (0.077–0.82), *P* = 0.023), whereas TMB was not prognostic (Supplementary Fig. 16c; HR = 0.84 (0.35–2.02), *P* = 0.7).

When combining the HMF and the Riaz cohort, we corroborated that immune-edited patients, those with low-immune dN/dS and without escape mechanisms, have the worst prognosis after receiving immunotherapy, when filtering for one, two, or at least three synonymous mutations in the immunopeptidome (Supplementary Fig. 16d).

## Discussion

Although immunoediting is widely recognized as an evolutionary process directed by our immune system that selects for lowly antigenic or escaped clones, its dynamics during carcinogenesis and response to treatment are poorly understood. Over the last decade, TMB has been regarded as an immunogenicity metric widely used to enroll patients for ICI treatment. However, TMB does not capture tumor-immune evolutionary dynamics and a large proportion of patients do not respond to immunotherapies despite their TMB status. In addition, a recent study has shown that low TMB colorectal cancers can also achieve clinical response to ICIs[9]. The increasing evidence supporting the adaptive immunity's role on sculpting the cancer genome[40], and the limitations of predictive biomarkers for ICI treatment, highlight the need for accurate metrics reflecting the underlying tumor evolutionary history.

Here, we propose that the nonsynonymous to synonymous mutations ratio, or dN/dS, estimated on the self-immunopeptidome can be used to differentiate between immune-edited versus immune-escaped tumors enabling treatment response prediction. We hypothesized that metastatic tumors with an evolved escape mechanism accumulate neoantigens and that their accumulated antigenicity is unmasked by immunotherapy. In contrast, immune-edited tumors have an overall low tumor antigenicity and will be less likely to respond to ICIs. This hypothesis was corroborated by our analysis of ICI response in more than 300 metastatic tumors. Our findings demonstrate the importance of evolutionary aware metrics over the standard TMB as predictive biomarkers for ICI treatment.

Importantly, during immunoediting, growing cells are under immune selection. However, negative selection in cancer has been a controversial topic[28,29,41]. Although some studies have shown an association between immune activity and selective pressures[2,13,24,42,43], others have claimed that there is limited evidence to prove this relationship[28,44]. As several studies have applied dN/dS as selection metric in cancer and normal tissue[13,14,45–48], we aimed to understand immune dN/dS dynamics to explain the lack of selection signals in cancer. Beyond proposed explanations, we demonstrate that by failing to classify individuals into edited and escaped, immune selection signals are lost. The low number of tumor types with strong immune selection suggests that a majority have some undetected immune escape mechanism, which will ultimately be possible to detect using immune dN/dS.

Precisely which genes and genetic alterations lead to immune evasion, or immune recognition, remain unknown. Escape or antigenic variation may arise in certain contexts due to other genetic events such as CN alterations[27], gene fusions[49] or frameshift events[50]. Discovery and profiling of non-canonical transcripts derived from viruses or aberrant splicing isoforms has only been possible recently thanks to new technologies (that is, long-read sequencing). In addition, absence of immune selection in tumors with no genetic escape suggests that other mechanisms mimicking evasion may also exist; that is, tumor clones in immune-privileged tissues growing under immune-neutral dynamics. Another limitation to detect immune selection stems from the chosen personal immunopeptidome; germline variants, expression status, wrong MHC-binding predictions or mutations that reduce native affinity can impact the precision of immune selection, that is, predicted immunopeptidomes with an excess of peptides not truly bound to MHC will mask selection signals. All these limitations render immune selection metrics highly conservative and highlight the urgent need for a better understanding of natural immunity during tumor progression.

Further refinement of the immunopeptidome and escape genes, by improving MHC-binding prediction methods and assessing the functional impact of escape events, and discovery of non-human/alternative derived antigens must be a priority to reveal all possible and functional tumor–immune interactions during cancer evolution.

## Online content

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

## Methods

### Analysis of primary untreated tumors from TCGA

**TCGA data.** Somatic mutation calls of 10,202 samples across 33 tumor types from TCGA consortium were obtained via GDC portal (https://portal.gdc.cancer.gov/) for four callers: MuTect2, VarScan2, MuSE and SomaticSniper. We compared our results between callers and used MuTect2 for all remaining analysis. GDC has a unified pipeline with multiple quality control including a panel of normal to filter false positive somatic calls[51] (that is, germline contamination). MAF files were converted to VCFs and reannotated using ensemblVEP (v89) using flag –pick option (best ensembl transcript). Only point mutations classified as synonymous, missense, start loss, stop gain, stop loss or frameshift mutations were considered.

Normalized immune scores for multiple cell subpopulations were obtained from three different studies (Rooney et al.[26], Danaher et al.[33] and Thorsson et al.[34]). For cohort analysis, median immune scores per tumor type were calculated based on the available data (z-normalized scores, cell-type scores based on expression of specific genes or Thorsson immune expression cluster scores).

Immune selection metrics defined as HBMRs and simulated HBMRs were obtained from a previous study[28], available for 19 tumor types. The CRC acronym comprises colon adenocarcinoma and rectum adenocarcinoma samples.

Six-digit HLA allele information for 9,736 samples was obtained from the controlled access files from Thorsson et al.[34] all deposited in https://gdc.cancer.gov/about-data/publications/panimmune.

Expression and CN data per gene were downloaded from the GDCquery API available in the R package TCGAbiolinks. We normalized expression values by dividing the upper-quantile fragments per kilobase of exon per million mapped fragments (FPKM) per gene per sample by the maximum value observed in each sample and then applying a log transformation. To determine whether an individual was escaped based on CN status, we estimated the median number of deletions (CN = 1 or CN = 0) and the median number of gains (CN > 2) for a subset of 1,097 patients with CN status and immune dN/dS available (minimum synonymous 1). Individuals with above the median number of deleted escape genes were classified as 'escaped Del'. A similar approach was used for escaped gains.

To determine the impact of missense/truncating events on CN burden, we obtained the aneuploidy score for deletions and amplifications from the Supplementary Table 2 in a previously published paper[35].

**SOPRANO.** SOPRANO (http://github.com/luisgls/SOPRANO) was built based on our original method published in Zapata et al.[13] to calculate selection in variant effect predictor annotated files. It calculates the ratio of nonsynonymous to synonymous mutation rate (dN/dS) inside (ON-target dN/dS) and outside (OFF-target dN/dS) a target region using a 192-trinucleotide context correction (SSB192). SOPRANO takes two input files: (1) a variant effect predictor annotated file with somatic point mutations and (2) a target protein coordinates bed file. Missense and synonymous mutations must be included. Other point mutations such as truncating (stop-loss/stop gain) or splicing variants can be included as nonsynonymous events. The remaining mutation types will be discarded, except for intronic mutations. SOPRANO can use intronic mutations to improve the background mutation rate estimated from OFF-dN/dS regions by dividing the observed number of intronic mutations by the length of the intronic regions of the gene (estimated from hg19 genome). The resulting mutation rate is then averaged with OFF-dN/dS.

SOPRANO can calculate dN/dS using somatic mutation data from a single patient or a cohort on any region of interest. SOPRANO allows for excluding driver genes and/or randomizing the target region. SOPRANO uses a set of Ensembl transcript identifiers and their respective FASTA file, enabling estimation of dN/dS in any genome, irrespective of the version. A major limitation of SOPRANO is the inability to estimate immune selection when no mutations are present inside the immunopeptidome. In these cases, it is not possible to differentiate between immune-editing or escape.

In this work, immune selection was estimated at multiple levels (described in Table 1). For cohort analysis, we used HLA-A0201 or proto-HLA based predictions. For patient-specific analysis, we built a self-immunopeptidome based on all peptides predicted to bind at least one of the six-HLA germline alleles as the target region. We defined immune dN/dS as the ratio between dN/dS inside (ON-target dN/dS) and outside the immunopeptidome (OFF-target dN/dS) to correct for technical artifacts (germline contamination, overfiltering or false-positive somatic calls) that can bias the neutral dN/dS (~1). Immune dN/dS is a normalized metric of immune selection assuming that most coding somatic mutations outside the immunopeptidome are nonantigenic. This assumption is supported by the fact that only 10% of missense mutations lead to neoantigens with high recognition probability; thus, the majority of mutations outside the immunopeptidome (excluding driver genes) should be under neutral dynamics (dN/dS ~ 1). We tested different immunopeptidomes and demonstrated that in multiple conditions OFF-dN/dS approximate to one (Supplementary Fig. 2).

**Estimation of edited neoantigens from immune dN/dS.** As an example, an immune dN/dS of 0.6 indicates that 40% of nonsynonymous mutations ($n_a$) were removed with respect to the observed number of synonymous events ($n_s$) multiplied by the ratio of nonsynonymous ($\mu_a$) and synonymous mutation rates ($\mu_s$) (Equation 1).

$$\frac{dN}{dS} = 0.6 => n_a = 0.6 * n_s * \frac{\mu_a}{\mu_s} \qquad (1)$$

This formula also allows us to calculate the number of neoantigens removed by selection inside the immunopeptidome of each individual using:

$$n_{a_{edited}} = \left( n_{a_{obs}} - n_{s_{obs}} * \frac{\mu_a}{\mu_s} \right) \qquad (2)$$

where $n_{a_{edited}}$ is the number of nonsynonymous mutations removed from the tumor based on the observed number of nonsynonymous mutations ($n_{a_{obs}}$) and the neutral expectation ($\frac{\mu_a}{\mu_s}$) This number represent a lower boundary for the number of immune-edited mutations given that mutations outside the immunopeptidome might have also been edited by immune selection (nonsynonymous in non-self regions transforming a non-binding peptide to a binding peptide).

**Patient-specific immunopeptidomes.** To construct a human immunopeptidome, we downloaded coding transcripts with HGNC symbol and Ensembl transcript ID from Ensembl Biomart (genes v90). We obtained all transcript lengths and ran bedtools (V2.26) makewindows to get all possible overlapping 9-mers. We then obtained the FASTA sequence for each of these 11,060,000 9-mer and ran netMHCpan4.0 (and netMHCpan4.1) using a list of HLA alleles (4-digit resolution for HLA-A, -B and -C). This list was restricted to the top 70 HLA alleles that have more than 1% population frequency in a list of 1,277 samples from the 1000 Genomes Project cohort[52]. We selected all possible peptides with a %rank < 0.5 (strong binders or SB) as our unfiltered immunopeptidome.

To increase the specificity of our estimates, we filtered the dataset by intersecting strong binder peptides with a list of T cell-positive assay peptides from the Immune Epitope Database, IEDB (http://iedb.org accessed on 05/02/2018). The peptide was kept if the 9-mer sequence had an exact match within any IEDB-positive peptide of length 9 or more. We kept transcripts with mean and/or median expression of more than 1 FPKM (globally expressed genes) calculated across 33 TCGA tumor types obtained from the Human Protein Atlas. The strong

binder list intersected with IEDB-positive assays can be obtained from the GitHub repository (github.com/luisgls/SOPRANO) as a bed file: allhlaBinders_exprmean1.IEDBpeps.bed. To obtain an HLA-A0201 immunopeptidome, we filter this list for regions associated to HLA-A0201. To obtain a proto-HLA, we performed the same filtering but filtering for A0201, A0101, B0702, B0801, C0701 and C0702.

To generate a patient-specific immunopeptidome, we search in the precomputed immunopeptidome database (i.e allhlaBinders_exprmean1.IEDBpeps.bed) for the six matching HLA alleles using a script provided in the repository. We created a bed file containing an Ensembl transcript ID and the start and end of the merged set of predicted HLA-binding peptides.

**Alternative immunopeptidomes.** To benchmark SOPRANO, we calculated ON, OFF and immune dN/dS using multiple immunopeptidomes (Supplementary Fig. 2) from a random set of 1,000 HLA-A0201 individuals. We compared strong binders predicted using netMHCpan4.0 or netMHCpan4.1 (available in the github repository). We compared (a) strong binders (%rank <0.5 defined) and (b) weak binders (%rank <2 defined as SB + WB, Supplementary Fig. 2, immunopeptidome C and E). To determine the impact of patient-specific expression, all non-expressed transcripts (FPKM = 0) from each TCGA individual were removed (group C, E and G in Supplementary Fig. 2).

**TCGA data versus immune dN/dS.** We obtained cohort (Supplementary Table 2) and per-patient (Supplementary Table 4) immune dN/dS on 33 tumor types using SOPRANO. We excluded cohorts with less than 10 mutations in the immunopeptidome, used ExonicIntronic mode and SSB192 correction method. For single individual analysis, we used ExonicOnly mode, and kept individuals with at least a single synonymous mutation in the immunopeptidome. Ggpubr and ggstatsplot packages from R available from repository CRAN (2020-02-29) were used for correlation analysis. We ran SOPRANO for non-escaped patients by excluding tumors with a missense or a truncating mutations in one of the genes labeled as escape genes (labeled as escaped + ).

To determine the best linear regression model, we checked collinearity and compared cohort and per-patient SOPRANO ON, OFF and immune dN/dS (ON/OFF) from tumors with available immune infiltrate data. As dN/dS is a continuous outcome variable, and our aim was to identify which immune metrics, with unknown random effects, explain dN/dS, we used generalized linear mixed models. We selected the best performing model by using the function stepAIC and checked for normality and heteroscedasticity using the gvlma package. TMB was defined as the logarithm base 10 of the point mutation number. When TMB was added to the model, it violated the assumptions for linear regression. We compared the expression levels of PD1 and PDL1 in a univariate and multivariate model with CD8 T cells.

**Escape status of the TCGA cohort.** We obtained a list of genes from the Antigen Processing and Presentation Machinery (hsa04612) downloaded from KEGG. We included escape genes used in Rosenthal et al.[2] that were not present in this list such as *ERAP1*, *ERAP2*, *IRF1* and *PDIA3*. We also added *FAS* and *MEX3B*, both genes associated to immune response[53,54]. The final list consisted of 88 genes (Supplementary Table 4). We then classified each individual as escaped+ if there was a missense or a truncating mutation in one of these escape genes. To determine whether these genes were under positive selection in the TCGA cohort, we ran dndscv[14] with default parameters using only escape genes (Supplementary Table 6).

**Immune dN/dS and escape status for MSI-rich cohorts from TCGA data (Lakatos cohort).** We obtained a high quality subset of MSI-rich tumors, CRC, UCEC and STAD. In this dataset, curated data for other escape mechanisms and MSI status was available (Lakatos et al.[32]). We excluded those patients that did not have an assigned escape

phenotype or did not have a molecular subtype assigned (MSS, MSI or POLE). We also explored CN alterations in escape genes and determined patient escape status based on the median number of CNAs present in each tumor type. Clonal and subclonal mutations for these patients were obtained from our previous study.

**Analysis of the metastatic cohort before immunotherapy**
**HMF cohort.** Hartwig somatic calls and metadata were obtained from the Hartwig Medical Foundation under license agreement DR-078. We selected 308 metastatic patients who underwent immunotherapy after biopsy and had a recorded clinical response in the 'first response' column from the metadata. Only mutation types classified as synonymous, missense, start loss, stop gain, stop loss or frameshift were included. We only used somatic calls with the flag PASS, removed indels and reannotated SNVs using ensemblVEP (v90, reference Grch37) with the option –pick (best transcript per gene). We used VATools V1.0.0 to parse the input for SOPRANO.

Raw clinical data were supplied by HMF and final consistency checks are still to be performed. The response evaluations were not performed as part of a clinical trial and the timing of the evaluations was variable. Patients were classified as responders and nonresponders based on the first response recorded after treatment was initiated using RECIST1.1 response criteria. Responders were all those labeled complete response (1 case), or partial response (78 cases). Patients with stable disease (98 cases) or progressive disease (131 cases) were classified as nonresponders. There were 79 patients with no data, 2 classified as clinical progression, 4 classified as non-determined and 3 cases classified as non-complete response/non-progressive disease who were not included in the analysis. The timing from biopsy to response was not included. No further classifications were performed. We used the same set of 88 'escape' genes to classify patients into escaped+ and escaped−.

Due to the availability of whole-genome sequences in the HMF, we used MOBSTER[20] to perform clonal/subclonal deconvolution and determine clonal versus subclonal status of somatic mutations. We included mutations with VAF > 8% and assumed clonal CN states 1:0, 1:1, 2:0, 2:1 and 2:2. The rest of parameters were default.

**HLA genotype and immunopeptidome for 308 HMF individuals.** We obtained 308 germline whole genome sequencing data either in FASTQ or CRAM file format from the Google Cloud bucket. CRAM files were pre-processed using samtools (v1.11) to convert them to fastq files. Fastq files were aligned to the genome using Yara Mapper[55] against a default HLA reference, which generated the input for OptiType[56] to obtain 4-digit HLA type for each patient. The references are based on the IMGT/HLA Release 3.14.0, July 2013, and have been processed as described in the publication of OptiType.

**Survival analysis of the HMF cohort.** To perform survival analysis in the HMF cohort, we used the difference between the biopsy date respect to the date of death to obtain the time variable for those patients that were deceased. For patients classified as alive, we subtracted the last date recorded in the cohort as the maximum date from the biopsy date. Patients were classified as alive if they did not have a date of death recorded on the clinical metadata. Neutral individuals were those with immune dN/dS between 1.22 and its reciprocal, 0.82, based on the upper quartile of immune dN/dS value. These values are in concordance with the analysis performed in the longitudinal metastatic cohort. We also explored the association between d-immune dN/dS, defined as the absolute distance to one, with overall survival.

**Driver, escape and global dN/dS.** To determine driver, escape and global dN/dS, we ran SSB192 (http://github.com/luisgls/SSB-dNdS) with default parameters on each subgroup of the HMF cohort. We calculated driver dN/dS using the list of 196 genes provided in Martincorena et al.[14]. We calculated escape dN/dS using the list of 88 genes

used in this study. Global dN/dS is the dN/dS values across the whole exome considering all point mutations. Immunopeptidome dN/dS is the value obtained from SOPRANO.

**Statistical analysis of the HMF cohort.** Statistical tests were performed using Wilcoxon rank-sum test for two distributions or Kruskal–Wallis test when more than two distributions were present using the R package ggstatsplot, unless explicitly stated in the paper. Multiple test correction for coxph univariate analysis was performed using Bonferroni. For survival analysis, we used the package ggsurv and applied multiple test correction for pairwise comparisons using Benjamini–Hochberg, unless stated otherwise.

### Analysis of melanoma cohort before and after immunotherapy

**Riaz cohort.** We obtained the somatic mutation data from Riaz et al.[39], a melanoma cohort that was sequenced before and during immunotherapy. RECIST1.1 classification and clinical outcomes was provided by the authors. We obtained SOPRANO (github.com/luisgls/SOPRANO) ON, OFF and immune dN/dS using a patient-specific immunopeptidome consisting of peptides predicted to bind the native MHC (six HLA alleles). We classified individuals based on their immune dN/dS as low (<0.82), neutral (0.82–1.21) or high (>1.21).

**Escape status in the Riaz cohort.** We used the same set of 88 escape genes from the previous analysis to classify patients into escaped+ and escaped− based on the presence of stop loss, frameshift, stop gain, start loss and missense mutations. In addition, we classified patients as escaped+ if they had loss of heterozygosity or were fully homozygous in the HLA region. For the final cohort analysis, we selected 48 patients that had more than 10 mutations globally, at least one synonymous mutation in the immunopeptidome and had clinical response data.

**Statistical analysis of the Riaz cohort.** We performed the statistical comparison between dN/dS distributions of pre- (Pre) and post- (On) treatment samples using Wilcoxon Mann–Whitney test. For the cohort analysis, we mixed all samples for each strata and recalculated the immune dN/dS based on the observed counts of nonsynonymous and synonymous mutations and sites. To validate that the immune dN/dS of the nonresponder escaped− group was not an artifact from the selection of escape genes, we sampled 10,000 times 42 genes (equivalent to the number of genes mutated in our list of 88 genes) from all genes and recalculated dN/dS for each of the four groups (NR−, NR+, R−, R+). We compared the mean estimate obtained from the observed cohort (NR−) against the randomized distribution of dN/dS for NR−.

**Survival analysis of the Riaz cohort.** For the survival analysis, we first performed cox regression on TMB and immune dN/dS. To select an optimal cutpoint, we used the function surv_cutpoint from the R package survminer. This function allowed us to divide numerical variables into categorical variables for TMB and immune dN/dS. In our first classification of immune categories, we used 0.82 as the maximum cut-off for immune-edited patients and the reciprocal, 1.21, as the minimum value for immune-high patients. We compared the size of the confidence intervals for each patient to the immune dN/dS. To compare only immune-neutral and immune-edited patients, samples with high confidence intervals (CI > 5) and immune dN/dS higher than 1.21 were filtered for the analysis of survival. *P* values were obtained using the ggsurvplot function from the R package ggsurv.

**Simulations.** We used our freely available package (https://github.com/luisgls/dNdSSimulator) to simulate immune dN/dS under two extreme conditions: fully active immune system and no capacity for immune response (parameter 'pattack' set to 1 versus 0). Detailed description of how to run the simulator is on the website. In each condition, we simulated 1,000 tumors starting from an initial pool of five cells, a probability of immune recognition of 10% (1 out of 10 neoantigens will be recognized at each cell division with a probability set to pattack). Simulations were run until the 100th generation unless they reached carrying capacity before (number of cells > 2,000). In each simulation, at the final time point, population size and number of nonsynonymous/synonymous mutations in the immunopeptidome was recorded, and mutations present in less than 1% of cells were discarded. Only simulations with more than 1,000 cells were used to estimate immune dN/dS, and results were uploaded to GitHub repository (dNdSSimulator).

### Reporting summary
Further information on research design is available in the Nature Portfolio Reporting Summary linked to this article.

## Data availability
TCGA data were obtained from GDC portal (https://portal.gdc.cancer.gov/) and processed as described previously[21]. HBMR values of selection in the immunopeptidome were obtained from the supplementary material in Van Den Eynden et al.[28]. Normalized scores for immune cell infiltration were obtained from Rooney et al.[26], Danaher et al.[33] and Thorsson et al.[34]. Genes involved in the antigen presenting machinery were obtained from KEGG pathway hsa04612 (https://www.genome.jp/pathway/hsa04612). An assembled list of escape mechanisms for colon adenocarcinoma, rectum adenocarcinoma and STAD and UCEC was obtained from Lakatos et al.[32]. Somatic variant calls from 308 Hartwig Medical Foundation samples were downloaded from the Hartwig Data Portal under license agreement DR-075 (https://database.hartwigmedicalfoundation.nl/). Hartwig Medical Foundation Patient-level genome-wide germline and somatic data (raw BAM files and annotated variant call data) are considered privacy sensitive and available through an access-controlled mechanism. Somatic calls, clinical and HLA allele information from 68 patients with metastatic disease sequenced before and during immunotherapeutic treatment was obtained from the authors of Riaz et al.[39] and deposited in Zenodo (https://doi.org/10.5281/zenodo.7546705). SOPRANO results for each tumor type and patient are available as supplementary tables. Analyzed data, code and R markdown files to reproduce raw figures have been made available in Zenodo (https://doi.org/10.5281/zenodo.7416627).

## Code availability
SOPRANO is freely available at github.com/luisgls/SOPRANO. Simulator of stochastic branching process for immunoediting is available at http://github.com/luisgls/dNdSSimulator. Code for estimating positive selection in escape genes can be accessed from https://github.com/im3sanger/dndscv. Code for estimating driver, global and driver dN/dS can be obtained from http://github.com/luisgls/SSB_selection. We used the R programming language (environment 3.63, 2020-02-29) and standard R packages available at repositories such as CRAN (2020-02-29) and Bioconductor 3.12. Code and R packages needed for raw figures are available as R markdown files (https://doi.org/10.5281/zenodo.7416627). Bedtools 2.26 and R are needed for SOPRANO to run. To generate input files for SOPRANO, ensemblVEP v89 has been used. The software produced/used for this publication is fully described in Methods section. Tutorial to run SOPRANO is made available on http://github.com/luisgls/SOPRANO. samtools (v1.11) was used to convert fastq files. Yara Mapper (https://www.seqan.de/apps/yara.html) was used for read mapping. OptiType v1.3.3 was used to get HLA alleles. The references are based on the IMGT/HLA Release 3.14.0, July 2013. netMHCpan4.0 and 4.1 was used to predict MHC binding.

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

## Acknowledgements

L.Z. is supported by the European Union's Horizon 2020 research and innovation program under the Marie Skłodowska-Curie Research Fellowship scheme (846614). A.S. and T.A.G. are supported by the Wellcome Trust (202778/B/16/Z and 202778/Z/16/Z, respectively) and Cancer Research UK (A22909 to A.S., A19771 and DRCNPG-May21_100001 to T.A.G.). We acknowledge funding from the National Institute of Health (National Cancer Institute grant U54 CA217376) to A.S. and T.A.G. This work was also supported by a Wellcome Trust award to the Centre for Evolution and Cancer (105104/Z/14/Z) and by a AIRC/CRUK/FC Accelerator Award to A.S. (CRUK: A26815, AIRC: 2279). We thank M. Punta and C.-A. Garin for support and discussion. We thank S. Quesada and E. Ghorani for important insights on the project. This publication and the underlying study have been made possible partly on the basis of the data that Hartwig Medical Foundation and the Center of Personalised Cancer Treatment have made available to the study. Figures 1 and 5a were created with Biorender.com. G.C. acknowledges funding from AIRC under MFAG 2020-ID. 24913 project—P.I. Caravagna Giulio.

## Author contributions

L.Z. conceived, designed, implemented and performed all analysis. G.C., M.J.W., E.L., A.A. and B.W. provided support with the analysis. K.A.J., D.C., C.J., L.G. and S.M. provided bioinformatic support. D.C., T.A.C. and N.R. generated sequencing data and sample information for the melanoma cohort. T.A.G. and A.S. supervised the project. L.Z. wrote the first draft of the paper. L.Z., A.S. and T.A.G. wrote the final version of the paper with the help of all the authors.

## Competing interests

T.A.C. is a co-founder of Gritstone Oncology and holds equity. T.A.C. holds equity in An2H. T.A.C. acknowledges grant funding from Bristol Myers Squibb, AstraZeneca, Illumina, Pfizer, An2H and Eisai. T.A.C. has served as an advisor for Bristol Myers, MedImmune, Squibb, Illumina, Eisai, AstraZeneca and An2H. T.A.C. and D.C. hold ownership of intellectual property on using tumor mutational burden to predict immunotherapy response, with a pending patent, which has been licensed to PGDx. The other authors declare no competing interests.

## Additional information

**Correspondence and requests for materials** should be addressed to Luis Zapata, Trevor A. Graham or Andrea Sottoriva.

| | |
|---|---|

# Reporting Summary

## Statistics

For all statistical analyses, confirm that the following items are present in the figure legend, table legend, main text, or Methods section.

| n/a | Confirmed | |
|---|---|---|
| ☐ | ☒ | The exact sample size (*n*) for each experimental group/condition, given as a discrete number and unit of measurement |
| ☒ | ☐ | A statement on whether measurements were taken from distinct samples or whether the same sample was measured repeatedly |
| ☐ | ☒ | The statistical test(s) used AND whether they are one- or two-sided<br>*Only common tests should be described solely by name; describe more complex techniques in the Methods section.* |
| ☒ | ☐ | A description of all covariates tested |
| ☐ | ☒ | A description of any assumptions or corrections, such as tests of normality and adjustment for multiple comparisons |
| ☐ | ☒ | A full description of the statistical parameters including central tendency (e.g. means) or other basic estimates (e.g. regression coefficient) AND variation (e.g. standard deviation) or associated estimates of uncertainty (e.g. confidence intervals) |
| ☐ | ☒ | For null hypothesis testing, the test statistic (e.g. *F*, *t*, *r*) with confidence intervals, effect sizes, degrees of freedom and *P* value noted<br>*Give P values as exact values whenever suitable.* |
| ☒ | ☐ | For Bayesian analysis, information on the choice of priors and Markov chain Monte Carlo settings |
| ☒ | ☐ | For hierarchical and complex designs, identification of the appropriate level for tests and full reporting of outcomes |
| ☐ | ☒ | Estimates of effect sizes (e.g. Cohen's *d*, Pearson's *r*), indicating how they were calculated |

*Our web collection on statistics for biologists contains articles on many of the points above.*

## Software and code

Policy information about availability of computer code

| | |
|---|---|
| Data collection | Data collection was performed using the R programming language (environment 3.63, 2020-02-29) and the TCGAbiolinks package. SOPRANO code is freely available at github.com/luisgls/SOPRANO. Code for simulator of stochastic branching process for immunoediting is available at github.com/luisgls/dNdSSimulator. Code for estimating positive selection in escape genes can be accessed from https://github.com/im3sanger/dndscv. Code for estimating driver, global and driver dN/dS can be obtained from github.com/luisgls/SSB_selection. |
| Data analysis | We used the R programming language (environment 3.63, 2020-02-29), and standard R packages available at repositories such as CRAN (2020-02-29) and Bioconductor 3.12. Bedtools 2.26 and R are needed for SOPRANO to run. The software produced for this publication is made available as described in the methods section of the paper. Tutorial to run SOPRANO is made available on github.com/luisgls/SOPRANO.samtools (v1.11) was used to convert fastq files. Yara Mapper (https://www.seqan.de/apps/yara.html) was used for read mapping. OptiType v1.3.3 was used to get HLA alleles. The references are based on the IMGT/HLA Release 3.14.0, July 2013. netMHCpan4.0 and 4.1 was used to predict MHC binding. |

For manuscripts utilizing custom algorithms or software that are central to the research but not yet described in published literature, software must be made available to editors and reviewers. We strongly encourage code deposition in a community repository (e.g. GitHub). See the Nature Portfolio guidelines for submitting code & software for further information.

## Data

Policy information about availability of data

All manuscripts must include a data availability statement. This statement should provide the following information, where applicable:

- Accession codes, unique identifiers, or web links for publicly available datasets
- A description of any restrictions on data availability
- For clinical datasets or third party data, please ensure that the statement adheres to our policy

TCGA data was obtained from GDC portal (https://portal.gdc.cancer.gov/) and processed as described previously. HLA Binding Mutation Rates (HBMR) values of selection in the immunopeptidome were obtained from the supplementary material in Van Den Eynden et al. Normalized scores for immune cell infiltration were obtained from Rooney et al, Danaher et al and Thorsson et al. Genes involved in the antigen presenting machinery were obtained from KEGG pathway hsa04612 (https://www.genome.jp/pathway/hsa04612). Assembled list of escape mechanisms for COAD, READ and STAD and UCEC was obtained from Lakatos et al. Somatic variant calls from 308 Hartwig Medical Foundation (HMF) samples were downloaded from the Hartwig Data Portal under license agreement DR-075 (https://database.hartwigmedicalfoundation.nl/). HMF Patient-level genome-wide germline and somatic data (raw BAM files and annotated variant call data) are considered privacy sensitive and available through an access-controlled mechanism. Somatic calls, clinical and HLA allele information from 68 metastatic individuals sequenced before and during immunotherapeutic treatment was obtained from the authors of Riaz et al and deposited in Zenodo (10.5281/zenodo.7546705). SOPRANO results for each tumor type and patient are available as supplementary tables. Analyzed data, code and R markdown files to reproduce raw figures have been made available in Zenodo (10.5281/zenodo.7416627).

# Field-specific reporting

Please select the one below that is the best fit for your research. If you are not sure, read the appropriate sections before making your selection.

☒ Life sciences ☐ Behavioural & social sciences ☐ Ecological, evolutionary & environmental sciences

For a reference copy of the document with all sections, see nature.com/documents/nr-reporting-summary-flat.pdf

# Life sciences study design

All studies must disclose on these points even when the disclosure is negative.

| | |
|---|---|
| Sample size | All sample sizes are described in the manuscript. Table 1 contains information on sample size for data used. No sample size calculation was performed for patient selection as all samples available were used. A minimum number of mutations was used for estimating immune dN/dS as described in the manuscript. |
| Data exclusions | No data exclusions. |
| Replication | R Markdown files and data needed to replicate our results are deposited in synapse. |
| Randomization | We randomize genes as escape using same gene sample number to demonstrate the effect of immune selection in escape genes. This analysis involving randomization of samples is described in the manuscript |
| Blinding | n/a. Data collection involved selecting all samples available from three cohorts. |

# Reporting for specific materials, systems and methods

We require information from authors about some types of materials, experimental systems and methods used in many studies. Here, indicate whether each material, system or method listed is relevant to your study. If you are not sure if a list item applies to your research, read the appropriate section before selecting a response.

## Materials & experimental systems

| n/a | Involved in the study |
|---|---|
| ☒ ☐ | Antibodies |
| ☒ ☐ | Eukaryotic cell lines |
| ☒ ☐ | Palaeontology and archaeology |
| ☒ ☐ | Animals and other organisms |
| ☒ ☐ | Human research participants |
| ☒ ☐ | Clinical data |
| ☒ ☐ | Dual use research of concern |

## Methods

| n/a | Involved in the study |
|---|---|
| ☒ ☐ | ChIP-seq |
| ☒ ☐ | Flow cytometry |
| ☒ ☐ | MRI-based neuroimaging |

