## [Peer Review File · Nature Genetics]

Peer Review Information

Manuscript Title: Immune selection determines tumour immunogenicity and influences response to checkpoint inhibitors

Corresponding author name(s): Luis Zapata

Reviewer Comments & Decisions:

Decision Letter, initial version:
--

5th Jan 2022

Dear Dr Zapata,

I hope that you and your family had a happy and healthy Christmas.

Please accept my apologies for the delay in returning this to you. Thank you for bearing with me. Your Article, "Immunoediting dynamics determine tumour immunogenicity and drive response to checkpoint inhibitors" has now been seen by 3 referees. You will see from their comments copied below that while they find your work of considerable potential interest, they have raised quite substantial concerns that must be addressed. In light of these comments, we cannot accept the manuscript for publication, but would be very interested in considering a revised version that addresses these serious concerns.

We hope you will find the referees' comments useful as you decide how to proceed. If you wish to submit a substantially revised manuscript, please bear in mind that we will be reluctant to approach the referees again in the absence of major revisions.

To guide the scope of the revisions, the editors discuss the referee reports in detail within the team, including with the chief editor, with a view to identifying key priorities that should be addressed in revision and sometimes overruling referee requests that are deemed beyond the scope of the current study. In this case, our assessment is that the reviewers are supportive of the concept you are presenting, but they are all asking for further characterisation, validation, and benchmarking to show

that your conclusions are justified and that your method is robust. We ask that you address all their concerns in full, ideally experimentally, or textually where appropriate.

If you choose to revise your manuscript taking into account all reviewer and editor comments, please highlight all changes in the manuscript text file. At this stage we will need you to upload a copy of the manuscript in MS Word .docx or similar editable format.

*2) If you have not done so already please begin to revise your manuscript so that it conforms to our Article format instructions, available [here](http://www.nature.com/ng/authors/article_types/index.html). Refer also to any guidelines provided in this letter.

[redacted]

If you wish to submit a suitably revised manuscript we would hope to receive it within 6 months. If you cannot send it within this time, please let us know. We will be happy to consider your revision so long as nothing similar has been accepted for publication at Nature Genetics or published elsewhere. Should your manuscript be substantially delayed without notifying us in advance and your article is eventually published, the received date would be that of the revised, not the original, version.

Thank you for the opportunity to review your work.

Sincerely,

Safia Danovi
Editor
Nature Genetics

Referee expertise:

Referee #1: cancer immunogenomics, computational biology

Referee #2: cancer immunology, evolution

Referee #3: cancer genomics, evolution, computational biology

Reviewers' Comments:

Reviewer #1:

Remarks to the Author:

Summary

In their submitted work entitled “Immunoediting dynamics determine tumour immunogenicity and drive response to checkpoint inhibitors”, the authors proposed a simple yet interesting metric of selection, the ratio of nonsynonymous to synonymous mutations, or “immune dN/dS”, estimated on the immunopeptidome. They analyzed genomic mutations in most TCGA tumors and a heterogeneous cohort of metastatic cancers, computed the immune dN/dS, characterized immune selection, and performed additional survival and correlation analyses to demonstrate its clinical importance, specifically, its impact on predicting treatment response to ICIs. The authors then proposed immune dN/dS as a surrogate of tumor immunogenicity. Overall, these data are interesting, and the metric (immune dN/dS) may have important clinical applications. However, the paper is descriptive and correlative, and it suffers from several major limitations.

1. It is important to understand the evolutionary dynamics of immune selection using longitudinal samples. The word “dynamics” showed up in the main title, abstract and multiple times in the main text. However, majority of the data present in this study (e.g., TCGA) were from samples collected at a single time point. The metastatic patient cohort is very heterogeneous in term of patients, treatments received, and cancer types, and only covered two time points. Although this could be limited by the availability of the in-house or public datasets, disappointingly, a dynamic picture of immune selection is not included in this study.
2. The paper is in general descriptive and correlative without functional validation of some of their key discoveries.
3. While there are many single-cell datasets publicly available, the correlation between immunopeptidome dN/dS and immune metrics was largely based on immune deconvolution of bulk expression data, with a very limited resolution. The bulk RNA-seq experiments typically measure total gene expression from heterogeneous tissues, including tumor, stroma, and immune compartments of the samples. It is known that the tumor cell purity varied greatly across TCGA tumor samples, so does the cellular compositions, however, this will be masked in bulk samples. CD8 T cells are a heterogeneous group of cells with diverse transcriptional states and functional phenotypes. However, this was not examined at single-cell resolution.
4. The results are interesting, but they triggered some other interesting questions, for example, which genes and/or genomic alterations lead to immune evasion and recognition and drive the selection process?

5. The authors showed that the immune-escaped tumors had a better response to immunotherapy than immune-edited tumors and they proposed immune dN/dS as a surrogate of tumor immunogenicity to predict response to ICIs. However, it is still uncertain from their results whether immune dN/dS outperforms other existing markers, such as various factors analyzed in this CELL paper from Litchfield et al. (PMID: 33508232) and biomarkers/signatures proposed by multiple other studies published over the past few years.

6. It is known that not all the mutant alleles are transcribed and only less than half (or a third) of the somatic mutations are detectable in their transcripts using RNA-seq data, and it is also known that transcriptome data are usually included to predict neoantigens. For this study, it is strongly suggested that the authors examine the relationship between the genomic dN/dS (from WES data) and the transcriptomic dN/dS (from RNA-seq data), and the impact of transcriptomic dN/dS on immune selection.

7. Some other genomic events such as splicing variants, gene fusions, and copy number alterations may also lead to immunogenic variation but this was not explored in this study. It would be interesting to know the presence and frequencies of these events in immune-edited tumors and immune-escaped tumors and their association (and impact on) with the immune dN/dS.

8. The definition of some terms and the classification approach are somewhat ambiguous, it is challenging even for readers with immunogenomics background. For example, the immune-edited tumors and immune-escaped tumors, the ON and OFF, the escape genes, the global and immune dN/dS, the heterogeneity of the immunopeptidome repertoire, the tumor immunogenicity, etc. How to properly separate the escaped from edited tumors? Is there a mixed type and the frequency of the mixed type? Overall, the samples were dichotomized and lacked of quantitative approach.

9. In addition, the key data processing steps also warrant further check. The Methods were in general lacked sufficient details regarding key steps, especially how the groups described above in #7 were defined. Moreover, it is unclear to me how the mutation data were filtered (quality filtering, common and rare SNPs, functional prediction of missense mutations), whether the small indels are included in dN. How the clonal and subclonal mutations are defined. And how reliable are the mutations included in dN/dS, i.e. how many (the %) of those mutations can be validated by independent platforms?

Reviewer #2:

Remarks to the Author:

Zapata et al. present an interesting analysis of public tumor datasets to investigate the relationship between tumor immune escape and response to immunotherapy. This issue is very important because

not all patients respond to immunotherapy and there are limitations of current predictive markers of response such as tumor mutation burden in certain tumor types. For example, renal cell carcinoma is within the top 25% of response rates to immunotherapy while having a relatively low tumor mutation burden (see Yarchoan et al. 2017 reference below). The authors use the metric of immune dN/dS which is the ratio of nonsynonymous to synonymous mutations restricted to coding regions in the genome that are predicted to bind to the most common HLA alleles or patient-specific HLA alleles (immunopeptidome). They demonstrate that a score of <1 correlates with CD8+ T cell infiltration and immune editing and selection. They suggest that this depletion of neoantigen+ cells from the tumor removes the targets that are needed for CD8+ T cells that are promoted by immunotherapy to recognize cancer cells as non-self and attack them. In contrast, tumors that maintain a neutral dN/dS score close to one still have sufficient neoantigen+ cells for recognition by CD8+ T cells in part because one or more immune escape mechanisms have stopped the process of their removal from the tumor. Immunotherapy is then able to overcome the escape mechanism in responders. These original findings are significant and of great interest to the field of cancer immunology and immunotherapy. The data are presented well in the figures and the conclusions are validated across several datasets.

Major Comments

Can the authors relate their findings to those summarized in Figure 1 in the following paper? Yarchoan M et al., Tumor mutational burden and response rate to pd-1 inhibition. *N Engl J Med.* 2017;377(25):2500–01. To what extent, for example, could immune escape explain the high response rate of renal cell carcinoma to immunotherapy despite low tumor mutation burden? Analysis and/or discussion of this would enhance the manuscript.

Can the authors comment on the extent to which analysis of only point mutations in the annotated coding immunopeptidome is a limitation? For example, it is known that tumor antigens can be encoded by viral proteins, mutations in alternate expressed reading frames, reverse strand transcription, human endogenous retrovirus insertion elements, peptide splicing, unique protein complexes on the cell surface that are not classic peptides presented by MHC (e.g. TRAIL/DR4), lipids presented by CD1, and metabolites presented by MR1.

Minor Comments

The font in the figures throughout the paper (starting with the cancer types in 2A and 2B) is in many cases extremely small. Please ensure the journal's minimum font size recommendations are followed. Please define MSS, MIS, and POLE in the main manuscript text at first use.

It is confusing to refer to immune edited as (edited-) on line 298. Should this be escape- or edited+? Please revise the incomplete sentences starting with "An" on line 316 and the sentence starting with "Similar" on line 323.

On line 411, the authors are referring to the Chi-squared analysis in 4D, but 4C is used in the text.

The y-axis in 5B should include Immune dN/dS.

Reviewer #3:

Remarks to the Author:

Zapata and colleagues present an analysis of immunoediting dynamics across cancers. Using 8543 primary tumours and 356 metastatic tumours, the authors explore the extent and importance of immune editing. The authors suggest negative selection is pervasive, and that the extent of immune editing can be prognostic and predictive of response to immune checkpoint blockade.

Given that there is considerable debate in the literature regarding the extent of neo-antigen directed immune editing and, moreover, the extent of negative selection, this is an important and timely analysis. However, this also means that it is absolutely imperative that the conclusions are justified, the methods are extremely rigorous, and validation is performed. As it stands, I think each of these areas require some additional work. Below, I have suggested some additional analysis that I think could help assess the strength of the findings. I would also suggest further validation would considerably strengthen this manuscript.

General comments:

- the definitions of immune editing and immune escape doesn't quite fit within the classical definitions as outlined by Schreiber. While I think I understand the distinction the authors are making, i think it would be useful to rephrase to avoid confusing readers.

- The authors state that "At a cohort level, we observe that the majority of tumours undergo neutral evolution as a consequence of immune escape, which masks ongoing immune selection in the non-escaped minority of cases." I do not wish to delve into a debate regarding neutral evolution. However, I think it is important to clarify that this does not mean the tumour is exclusively a result of neutral evolutionary dynamics. I do not believe the authors are trying to suggest this - but I think it is important to make clear for the reader.

- SOPRANO requires further validation. The extent to which the tool is different from the original publication is quite unclear. Overall, I think a more in-depth comparison between why the results differ between SOPRANO and the recently published results of Van Den Eyden would be very useful.

- In general, a more detailed explanation of the underlying logic behind SOPRANO is needed, even if this has been published before. For instance, it would be good to clarify the following:

1. a mutation can alter a peptide from non-binding to binding, in this case will it not be considered? (this is immunogenic, but will be missed)
2. how is a mutation that alters a peptide from binding to non-binding considered? (this is not immunogenic, but will be considered immunogenic)

3. are only missense mutations considered?

4. most mutations are not expressed - is this taken into account beyond looking at genes that are generally expressed?

- The cohort should be justified and explained in the main text. The 8543 tumours suddenly appears without any explanation? Given that immune cell scores are also available from Thorsten et al., and this includes more samples, the authors should consider analysing the full cohort.

- Why is %rank < 0.5 used? This is classically a 'strong binder' - do the results change if %rank < 2 is used instead? And, while I appreciate the goal of the paper is not to compare different binding predictions, the authors should consider adopting the latest binding prediction tools, e.g. netMHC4.1 and may wish to explore how the results change if IC50 is used instead of %rank. I appreciate this may be a large amount of work, but I think it would be useful to demonstrate that the results remain robust.

- The authors calculate the ratio between ON-dN/dS and OFF-dN/dS. This may mask some signal. For instance, both may show significant negative selection, or significant positive selection, and the ratio is less informative? A detailed exploration of key differences would be useful here.

- The immune dNdS is a very important part of the paper. As such, the explanation needs to be much clearer. I think a simulated data-set could be useful here to demonstrate expectations.

- to what extent does immune dNdS correlate with mutation burden?

- the dNdS analysis should permit an exploration of potentially how many neoantigens have been edited per tumour - what do the numbers suggest?

- Why not restrict the analysis to patients with HLA-A*02:01? (I would suggest either doing this or accurately computing the HLA alleles for each patient and using more precise scores)

- The authors find that CD8 T cell abundance is negatively correlated with immune dN/dS - this is very intriguing - is this the case within cancer types, or when using all together? A model taking into account cancer type would be useful here and other potential confounders would be useful here. Is there no correlation between non-immune dNdS and CD8 T cell abundance?

- How do the results change if CRC, STAT and UCEC are included? Further justification for excluding/including whole sets of tumours is needed.

- The analysis ignores copy number alterations in immune genes, does this improve the associations?

- The authors make a strong statement that the "reported lack of neoantigen depletion signals on the genome is an artefact of the heterogenous nature of cohorts that either have a large proportion of immune-escaped tumours (e.g. colorectal cancer) or come from immune-privileged tissues (Brain tumours)." It would be useful to more directly assess whether this is the case if other methods beyond SOPRANO are used here too.
- cut-offs with regards to number of synonymous mutations seems a little odd - is there a danger that this is enriching the cohort for certain types of tumours? It would be useful to explore different cut-offs more clearly.
- I'm a little unclear why HLA type isn't available for the entire cohort?
- I think it would be useful to explore how the analysis of response to ICB looks in the context of the full cohort (despite potential caveats).
- the results would be considerably strengthened by the inclusion of a validation cohort - and demonstration this is clearly independent of cancer type
- the authors should explore the extent to which immune editing is also prognostic in TCGA in tumours without ICB treatment.

DATA and code:

- I'm afraid I couldn't manage to navigate to where the data and code is to reproduce the figures and analysis. Although I have little doubt the authors will make this available - I need this to properly review the manuscript.

Author Rebuttal to Initial comments

Reviewer #1:

Remarks to the Author:

Summary

In their submitted work entitled "Immunoediting dynamics determine tumour immunogenicity and drive response to checkpoint inhibitors", the authors proposed a simple yet interesting metric of selection, the

ratio of nonsynonymous to synonymous mutations, or “immune dN/dS”, estimated on the immunopeptidome. They analyzed genomic mutations in most TCGA tumors and a heterogeneous cohort of metastatic cancers, computed the immune dN/dS, characterized immune selection, and performed additional survival and correlation analyses to demonstrate its clinical importance, specifically, its impact on predicting treatment response to ICIs. The authors then proposed immune dN/dS as a surrogate of tumor immunogenicity. Overall, these data are interesting, and the metric (immune dN/dS) may have important clinical applications. However, the paper is descriptive and correlative, and it suffers from several major limitations.

R: We thank the reviewer for their summary of our study and are glad that they find our work interesting. We address their insightful critique in detail below.

1. It is important to understand the evolutionary dynamics of immune selection using longitudinal samples. The word “dynamics” showed up in the main title, abstract and multiple times in the main text. However, majority of the data present in this study (e.g., TCGA) were from samples collected at a single time point. The metastatic patient cohort is very heterogeneous in term of patients, treatments received, and cancer types, and only covered two time points. Although this could be limited by the availability of the in-house or public datasets, disappointingly, a dynamic picture of immune selection is not included in this study.

R: We thank the reviewer for this important comment. For clarity and following this criticism, we replaced the word dynamics in some sections of the manuscript. We agree with the reviewer on the value of longitudinal datasets to understand evolutionary dynamics of cancer progression, however it is clear from the large corpus of work on cancer evolution that the tumour genome contains a surreptitious record of cancer evolutionary dynamics (reviewed in [1]), and our study continues in this vein.

In this work, we conceive cancer immunoediting as an intrinsically dynamic evolutionary process that constantly removes neoantigens and ultimately drives adaptation, leading to immune-edited and immune-escaped tumors (See Response Fig. 1, where we provide a toy model of the immune system and its interaction with cancer cells and immunogenic mutations).

Some specific examples within our study include:

- We hypothesized that escape events release immune pressure leading to a subclonal accumulation of neoantigens and ultimately an immune dN/dS closer to neutrality over time ($dN/dS \sim 1$). In figure 3A,3B and in new figures S4A,S4B, we demonstrate that escaped+ tumors harbour more mutations (including missense, truncating and copy number alterations) than non-escaped (Escaped-) tumors; while in Fig. 3C and new figure S4C we correlate TMB and immune dN/dS and demonstrate that tumors with higher TMB have also immune dN/dS values closer to one “only” in escaped+ tumors.
- In fig 3D, we demonstrate that immune dN/dS is consistently closer to one in escaped+ tumors when considering all mutations (Clonal plus subclonal signal), while clonal immune dN/dS remained significantly lower than one for both groups, suggesting that escape is a late event for microsatellite stable tumors.
- In our section for the immunotherapy cohorts, the longitudinal dataset provides two timepoints allowing us to demonstrate the impact of immunotherapy on immune selection signals before and after treatment for responders and non-responders. We observed that treatment specifically removes nonsynonymous mutations in the immunopeptidome of responders (ON-target dN/dS).

The reviewer highlights the heterogeneous nature of our datasets, with respect to cancer types, treatments, etc. We see this heterogeneity as a strength as it allows us to contrast immunoediting between different patient groups (and indeed this is a major focus on the manuscript). Nevertheless, we completely agree with the reviewer on the value of longitudinal datasets, particularly which document the immune response through therapy, and will seek to analyse these as more become available.

Response Figure 1. Cancer immunoediting model. A) Premalignant dividing cells accumulate mutations at each cell division. B) Once a driver event is acquired an increased proliferation fuels the emergence of tumors. C) Cells with antigenic mutations are eliminated by the innate immune system. D) Immunoediting leads to immune-edited and immune-escaped clones in clinically detectable tumors. E) Among others, passenger, driver, immunogenic and escape mutations accumulate on the growing clone with a specific selective advantage measured by dN/dS (Adapted from submitted manuscript).

2. The paper is in general descriptive and correlative without functional validation of some of their key discoveries.

R: It is important to mention that there have been multiple functional studies reporting that immunoediting occurs during somatic evolution of cancer in experimental murine models[2]. In the cancer evolution field, negative selection is currently an evolving concept where the prevailing view is that evolutionary dynamics are dominated by positive selection for driver alterations. In this work, we proposed a simple yet sound explanation for the recently reported lack of immune selection signals in tumor genomes[3]. We used multivariate modelling to demonstrate that when accounting for tumors with a large proportion of escape events, the association between immune activity and immune selection becomes apparent, suggesting that grouping heterogeneous patients into cohort calculations reduces the power to detect negative selection. As a note to the reviewer, we have also developed a stochastic model (currently under revision in a separate manuscript) where we have explored the impact of mixing immune-edited and escaped cohorts into single values and our conclusions support the hypothesis that the lack of power to detect selection comes from mixing heterogeneous evolutionary paths.

In this line, we appreciate that the concept of cancer immunoediting is currently widely accepted, however the metrics to quantitatively measure its impact are currently lacking. We provide a simple metric (dN/dS) that combined with the evidence of immune escape mechanisms allow us to interpret evolutionary trajectories of single tumors while also giving us a strong predictor of response to cancer immunotherapies. We are highly critical of our work and continue to validate our predictions in collaboration with others that have mouse models of cancer immunoediting where we can directly measure dN/dS. We are convinced that this work is laying the ground for further research with a strong impact on the clinical management of cancer patients.

3. While there are many single-cell datasets publicly available, the correlation between immunopeptidome dN/dS and immune metrics was largely based on immune deconvolution of bulk expression data, with a very limited resolution. The bulk RNA-seq experiments typically measure total gene expression from heterogeneous tissues, including tumor, stroma, and immune compartments of the samples. It is known that the tumor cell purity varied greatly across TCGA tumor samples, so does the cellular compositions, however, this will be masked in bulk samples. CD8 T cells are a heterogeneous group of cells with diverse transcriptional states and functional phenotypes. However, this was not examined at single-cell resolution.

R: We thank the reviewer for this important comment. We agree that deconvolution of bulk RNAseq is an imperfect process. However, we required large cohorts of data where there was both exome sequencing data and parallel measurement of immune infiltration. Whilst we would love to have single cell data from thousands of patients with all coding point mutations and phenotypic maps to concurrently measure clonal selection and the immune infiltrate, to the best of our knowledge no such dataset currently exists. Of course, generating such a dataset is a monumental undertaking and not something we can conceive of in the near future, but we entirely agree with the reviewer that such a dataset will be of immense value.

In our manuscript, we had to rely on deconvolution from bulk RNA-seq available from public cohorts. In this new version, as requested by another reviewer we have included more samples with immune infiltrates available from three different studies (shown in Response figure 2, and thoroughly explored in new supplementary Fig. 2 and 3). We have also included stromal fraction as a possible confounding factor in our regression model aiming to link immune dN/dS and immune infiltrates controlling for tumor purity. In this analysis, we corroborated that when removing patients with a truncating mutation in one of the

escape genes (escaped+), the association between Immune dN/dS and immune cell fraction became significant in the escaped- group (Response Fig 3A/3B, new Supp fig 5C-D-E-F-G). When we expanded this analysis to all tumor types available and multiple other cellular states, and used AIC model selection to distinguish the best model, we found that only Leukocyte fraction and NK cells resting are the top performing predictors for immune dN/dS in both scenarios (with and without escape patients), and the most significant results emerge when removing those escaped+ patients (Response Fig 3B, new Supp Fig 5G).

Response Figure 2. (New Supp. Fig 2) Association between immune dN/dS and HBMR from Van Den Eynden et al to multiple immune cell predictors. In all scenarios where MSI rich tumors were removed, we observe a significant association to immune cell scores. This association is not significant with TMB.

Response Figure 3 (New Supp Fig 5F,5G). Model comparison between different predictors and independent variables in a multivariate analysis. A) Leukocyte fraction and stromal fraction as predictive variable for OFF, ON and Immune dN/dS in two scenarios : all tumor and patients - escaped+ and when removing escaped+ individuals and calculating immune dN/dS - escaped-. B) Multiple immune cell predictors for immune dN/dS in the two scenarios mentioned previously. We performed model selection using AIC criteria and determined that Leukocyte fraction and NK cells resting are the most significant variables.

Of note, a recent study has thoroughly compared the difference on cell composition obtained using bulk-sequencing and single-cell RNA strategies demonstrating comparable performance between these methods[4]. However, recognising the limitations of this approach, we have included the following statement in our manuscript: “To minimise possible biases from bulk-based cell deconvolution methods, we obtained published immune infiltration estimates reported in TCGA data from three different studies and calculated the median abundance for each immune subtype (Fig 2C, Supplementary Fig 2).”

4. The results are interesting, but they triggered some other interesting questions, for example, which genes and/or genomic alterations lead to immune evasion and recognition and drive the selection process?

R: We thank the reviewer for this interesting question. We have now included a new analysis in the main manuscript aiming to measure selection on mutations in the antigen presenting machinery genes. Specifically, we estimated dN/dS for missense and truncating mutations in these genes across all the TCGA cohort. In the new main figure 2H (Response Fig. 4), we observed that these genes often have dN/dS statistically significantly greater than 1 indicating that mutations in these genes are positively selected (predominantly HLA-A and HLA-B, B2M and KIR-like genes).

For the reviewer only, we further explored in one tumor type, stomach adenocarcinoma, as part of the highly enriched tumors in evasion events, other genetic alterations (downregulation of expression, or copy number alterations) in these escape genes. We found that KIR-like genes are generally lowly or not expressed (Response Fig 5) and are affected by copy number gains in 10% of cases (Response Fig. 6). We identified that KIR-like genes, often neglected from genomic studies, may also drive immune escape via downregulation or copy number alterations. However, exploring the impact of these genes, especially if these genes are assumed to be expressed only in immune cell lineages, is an endeavour that requires further validation that we aim to explore in a separate study.

Response Figure 4 (New main fig 2H). Volcano plot for dN/dS and significance of genes associated to the antigen presenting machinery using missense (left) or truncating mutations (right). In red, significant genes after multiple test correction.

Response Figure 5. Gene expression matrix for escape genes in TCGA individuals with stomach adenocarcinoma.

Response Figure 6. Copy number changes matrix for escape genes in TCGA individuals with stomach adenocarcinoma.

As our cohorts from patients treated with immunotherapy were smaller, we had insufficient data to reliably assess positive selection on the antigen presenting genes. However, we have added a map of mutation (and mutation types) prevalence for these genes on both of our immunotherapy cohorts and

we observed that KIR-like genes, ERAP2 and HLA genes are frequently mutated (Response Fig 7, and Response Fig 8).

Response Figure 7 (New Supp Fig 10). Landscape of genetic alterations in escape genes for individuals from the Hartwig Medical Foundation cohort.

Response Figure 8 (Main Fig 5C). Landscape of mutations in the list of escape genes for the Riaz cohort.

It is possible that other genes or genomic alterations are involved in immune evasion, but their identity is currently unknown. We note in our discussion: “There is currently not a comprehensive list of genes and genetic alterations that lead to immune evasion. It is possible that many of the mutations considered here

do not actually affect a cell's immunogenicity. Moreover, escape or immunogenic variation may arise in certain contexts due to other genetic events such as copy number alterations, gene fusions, frameshift events or epigenetic changes. Also, it is probable that some mutations in escape genes may confer resistance to immunotherapies, as evidenced in some studies where mutations in B2M are enriched in non-responders[5], highlighting the need for a better understanding of the functional impact of mutations in genes associated to immune evasion. Notably, our choice of "escape" genes is supported by the recent discovery of positively selected mutations in genes involved in the antigen presenting machinery pathway, such as the HLA genes described in this study."

5. The authors showed that the immune-escaped tumors had a better response to immunotherapy than immune-edited tumors and they proposed immune dN/dS as a surrogate of tumor immunogenicity to predict response to ICIs. However, it is still uncertain from their results whether immune dN/dS outperforms other existing markers, such as various factors analyzed in this CELL paper from Litchfield et al. (PMID: 33508232) and biomarkers/signatures proposed by multiple other studies published over the past few years.

R: We thank the reviewer for this suggestion. The most significant genetic factors identified by Litchfield et al[6] were clonal TMB followed by total TMB and CXCL9 expression. In our previous version, we already found that immune dN/dS is a better prognostic metric than total TMB. Following the reviewer's suggestion, we have now extended this analysis to include age, gender, clonal TMB, subclonal TMB, indel burden, CXCL9 and CD274 (PDL1) expression. We found that our multivariable model including escape status and immune dN/dS had better prognostic value (Response fig 9, concordance 0.74, Log-rank P-value 1e-06, and AIC 446), compared to when these variables were not included (0.72, log-rank P-value 4e-06 and AIC 452). We have included these results as a new supplementary figure 12E.

Response Figure 9 (D as new Supp Fig 12E). Multivariate cox regression models and hazard ratios. A) Basic model using variables identified in Litchfield et al. B) Same factors as in model A, but including immune dN/dS. C) Same as in model A plus escape status. D) Model A including immune dN/dS and escape status.

6. It is known that not all the mutant alleles are transcribed and only less than half (or a third) of the somatic mutations are detectable in their transcripts using RNA-seq data, and it is also known that transcriptome data are usually included to predict neoantigens. For this study, it is strongly suggested that

the authors examine the relationship between the genomic dN/dS (from WES data) and the transcriptomic dN/dS (from RNA-seq data), and the impact of transcriptomic dN/dS on immune selection.

R: We thank the reviewer for this comment. It is important to note that our predefined immunopeptidome contains transcripts that have a median and mean FPKM larger than one across multiple tissues. Now, we have compared immune dN/dS between our predefined immunopeptidome and by using a patient specific expression immunopeptidomes. We found that the number of mutations removed by this approach was very small and made no difference in our immune dN/dS estimates (Response figure 10).

However, this strategy was suitable for the analysis presented here and might not be useful for other datasets, therefore we have now made available in SOPRANO an unfiltered immunopeptidome that can be tailored by user-specific filtering using individual expression estimates. To further prove the impact of expression on dN/dS we also compared our filtered immunopeptidome to the unfiltered version (Response Fig 11, New Supp Fig 1J,1K,1L). We observed that unfiltered immune dN/dS was closer to neutrality and higher than the filtered immune dN/dS, corroborating that unexpressed regions evolve neutrally with respect to the immune system (described in Lines 197).

Response Figure 10. Immune dN/dS distribution for all individuals using the pre-filtered patient specific immunopeptidome available in GitHub (left) versus the immunopeptidome filtered by patient specific transcript expression.

Response Figure 11. (New Supp fig 1J,K,L,M) Immune and OFF dN/dS for expression or binding affinity based filtered immunopeptidome (mean and median < 1FPKM, strong binder 0,5% rank, weak binder 2% rank). Filtered immune dN/dS is

significantly lower than in the unfiltered set. Unfiltered dN/dS was expectedly closer to one due to the absence of selective forces on unexpressed variants, similar to when considering weak binders as part of the immunopeptidome.

7. Some other genomic events such as splicing variants, gene fusions, and copy number alterations may also lead to immunogenic variation but this was not explored in this study. It would be interesting to know the presence and frequencies of these events in immune-edited tumors and immune-escaped tumors and their association (and impact on) with the immune dN/dS.

R: We thank the reviewer for this valuable comment. We have now included new supplementary figure 4A and 4B (Response figure 12) characterizing the frequency of copy number alterations and splicing variants in escaped(+) versus escaped(-) tumors. We have also added this sentence to our manuscript (Lines 261-266): “tumors with a single alteration in one of the predefined escape genes harbour more splice, stop loss and non-sense mutations than those without an alteration in one of these genes. Moreover, irrespective of that low point mutation burden tumors (which are less likely to be escaped) usually have more copy number alterations[7] , we found that escaped(+) tumors have also more amplifications and deletions than escaped(-), suggesting a global relaxation of selection on these genomic events.”

Response Figure 12 (New Supp Fig 4A). Mutation burden comparison across all TCGA between escaped(+) and escaped(-) tumors for different mutation types.

8. The definition of some terms and the classification approach are somewhat ambiguous, it is challenging even for readers with immunogenomics background. For example, the immune-edited tumors and immune-escaped tumors, the ON and OFF, the escape genes, the global and immune dN/dS, the heterogeneity of the immunopeptidome repertoire, the tumor immunogenicity, etc. How to properly separate the escaped from edited tumors? Is there a mixed type and the frequency of the mixed type? Overall, the samples were dichotomized and lacked of quantitative approach.

R: We apologise for confusion caused by our lexicon. We have rephrased several sections in our manuscript to improve clarity. One of the key concepts is the distinction between immune-edited and immune-escaped tumors. We have included introductory lines to clarify the distinction between these two groups at the beginning of the introduction as requested by another reviewer. We now propose to separate these groups based on immune dN/dS and the presence of an escape mechanism: immune-edited will have immune dN/dS < 1 and no evidence of escape, while immune-escaped will have neutral ≈ 1 immune dN/dS and evidence of escape. In two other possible outcomes, such as low immune dN/dS and

evidence of escape or high immune dN/dS and no evidence of escape, results should be interpreted cautiously and further research is needed to understand the evolutionary relationship between the immune system and cancer cells in these scenarios. From our simulations (unpublished and under revision), we know that higher-than-one immune dN/dS can arise in scenarios where the number of synonymous mutations in the immunopeptidome is 1 or 0 and the immune system is active in at least 85% of cases. To use a quantitative approach to differentiate between immune edited and immune escaped tumors, will require a thorough understanding of all possible mechanisms of immune evasion and their consequences for the evolutionary fates of each cell. To simplify our interpretations, we have consistently modified our manuscript and labelled escaped+ and escaped- to refer patients with or without a mutation in the list of genes selected respectively.

9. In addition, the key data processing steps also warrant further check. The Methods were in general lacked sufficient details regarding key steps, especially how the groups described above in #7 were defined. Moreover, it is unclear to me how the mutation data were filtered (quality filtering, common and rare SNPs, functional prediction of missense mutations), whether the small indels are included in dN/dS. How the clonal and subclonal mutations are defined. And how reliable are the mutations included in dN/dS, i.e. how many (the %) of those mutations can be validated by independent platforms?

R: We thank the reviewer for highlighting our previous lack of clarity in the methodological description. We have modified our method section accordingly to describe all key data processing steps. In brief, somatic calls from 10202 TCGA patients were downloaded from the GDC. GDC provides results from four somatic variant callers, MuSE, MuTect2, VarScan2 and SomaticSniper and no consensus. Thus, we compared immune dN/dS between these four somatic callers and found no major differences except for a few tumor types (Response Figure 13). Specifically, we have included the following sentence in the methods sections: "GDC has a unified pipeline including multiple quality control steps such as a panel of normal filter to remove false positive somatic calls (i.e. germline contamination). MAF files were converted to VCFs and reannotated using ensembl-VEP release 89 with the flag `-pick` option (best ensembl transcript). Only point mutations classified as synonymous, missense, start_lost, stop gained, stop lost or frameshift mutations were considered for further analysis." To note, It has been shown that over filtering of somatic mutations leads to false signals of positive selection reflected by higher than one dN/dS values[8]. In our dataset, we observed that OFF dN/dS were consistently above one for most tumor types (Response Fig 24, new supp fig 1B) possibly associated to the strict filtering performed on these somatic calls and therefore rendering our estimates of negative selection highly conservative.

In this version and following the concern raised by the reviewer, we have also estimated immune dN/dS using the mutations from these four callers (Response Fig 13 and new Fig S1F). We have added the

following sentence to the manuscript (lines 181-185) “When comparing four different somatic callers (Mutect2, MuSe, SomaticSniper and VarScan2), immune dN/dS was similar for most tumor types except for Adrenocortical carcinomas (ACC) and diffuse leukemias (DLBC) where the number of mutations in the immunopeptidome was the lowest (Fig S1F).”

In addition, we have also clarified that the clonality of variants was used only for CRC, STAD and UCEC and was obtained from our previous study[9] . Clonality of variants for HMF cohort was obtained by running our MOBSTER software on whole genome sequences[10].

Response Figure 13 (New Supp Fig 1F). Immune dN/dS estimated from somatic mutations called using four different somatic callers.

Reviewer #2:

Remarks to the Author:

Zapata et al. present an interesting analysis of public tumor datasets to investigate the relationship between tumor immune escape and response to immunotherapy. This issue is very important because not all patients respond to immunotherapy and there are limitations of current predictive markers of response such as tumor mutation burden in certain tumor types. For example, renal cell carcinoma is within the top 25% of response rates to immunotherapy while having a relatively low tumor mutation burden (see Yarchoan et al. 2017 reference below). The authors use the metric of immune dN/dS which is the ratio of nonsynonymous to synonymous mutations restricted to coding regions in the genome that are predicted to bind to the most common HLA alleles or patient-specific HLA alleles (immunopeptidome). They demonstrate that a score of <1 correlates with CD8+ T cell infiltration and immune editing and selection. They suggest that this depletion of neoantigen+ cells from the tumor removes the targets that are needed for CD8+ T cells that are promoted by immunotherapy to recognize cancer cells as non-self and attack them. In contrast, tumors that maintain a neutral dN/dS score close to one still have sufficient neoantigen+ cells for recognition by CD8+ T cells in part because one or more immune escape mechanisms have stopped the process of their removal from the tumor. Immunotherapy is then able to overcome the escape mechanism in responders. These original findings are significant and of great interest to the field of cancer immunology and immunotherapy. The data are presented well in the figures and the conclusions are validated across several datasets.

R: We thanks the reviewer for this comprehensive summary of our study and their very positive opinion on the impact of the results.

Major Comments

Can the authors relate their findings to those summarized in Figure 1 in the following paper? Yarchoan M et al., Tumor mutational burden and response rate to pd-1 inhibition. N Engl J Med. 2017;377(25):2500–01. To what extent, for example, could immune escape explain the high response rate of renal cell carcinoma to immunotherapy despite low tumor mutation burden? Analysis and/or discussion of this would enhance the manuscript.

R: We thank the reviewer for highlighting the interesting study from Yarchoan and colleagues. We have performed an additional analysis on the data reported in Yarchoan et al. We observed from figure 1 in Yarchoan et al that tumors with high ($>30\%$) and medium ($>20\%$) objective responses are among the tumors with the highest proportion of escape events caused by point mutations in an antigen presenting

associated gene (See table below). An exception to this observation is kidney tumors, that do not display a high burden of point mutations in these genes, but they have reportedly a large fraction of patients with partial/total loss of HLA-I genes (Romero tissue antigens 2006, Jian frontiers 2021). We then looked at immune-dN/dS in kidney cancers vs melanomas and observed that the proportion of tumors with immune dN/dS above one for increasing cut-offs is significantly higher in kidney cancer (Response fig 14). Thus, we suggest that these tumors are enriched for other mechanisms of immune escape by HLA-I deletion, or a cryptic escape mechanism that renders the tumours still sensitive to immunotherapy.

Response Figure 14. Proportion of patients with immune dN/dS above the selected cut-off for Kidney Carinomas (KIRC) and melanoma (SKCM).

Table 1. List of tumor types, proportion of individuals with a point mutation in an escape gene, and objective responses from Yarchoan et al.

Tumor Type	Percentage Escaped	Objective Response
LUSC	0.79	High
SKCM	0.525	High
UCEC	0.366	Mid
BLCA	0.345	-
LUAD	0.335	Mid
STAD	0.323	Mid
CESC	0.301	Mid
HNSC	0.254	Mid
OV	0.232	Low
CRC	0.225	Mixed
LIHC	0.203	Mid
GBM	0.183	Low
KIRC	0.128	High
BRCA	0.126	Low
KIRP	0.117	Mid
PRAD	0.068	Low
LGG	0.059	Low
THCA	0.039	-

Can the authors comment on the extent to which analysis of only point mutations in the annotated coding immunopeptidome is a limitation? For example, it is known that tumor antigens can be encoded by viral proteins, mutations in alternate expressed reading frames, reverse strand transcription, human

endogenous retrovirus insertion elements, peptide splicing, unique protein complexes on the cell surface that are not classic peptides presented by MHC (e.g. TRAIL/DR4), lipids presented by CD1, and metabolites presented by MR1.

R: We thank and agree with the reviewer on this important comment. We have added a section in our discussion (lines 699-705) *"In addition, antigens derived from other mechanisms besides point mutation might also play an important role on tumor immunogenicity. Non-canonical transcripts derived from viruses or aberrant splicing isoforms can result in potent immune reaction forcing tumors to escape by other means beyond MHC complex alterations. However, the prevalence of these events remains unknown, and the necessary technologies to comprehensively profile them just recently have become available (such as long-read sequencing)."*

Importantly, and thanks to the suggestion from the reviewers we have analysed the burden of, other genomic alterations such as splice variants, stop loss/gain and aneuploidy in escaped versus non-escaped tumors (new supplementary fig 4A and 4B, described as well in response figure 12). Specifically, we have added this sentence: "globally tumors with a single alteration in one of the predefined escape genes harbour more splice, stop loss and non-sense mutations than those without an alteration in one of these genes. Moreover, irrespective of that low point mutation burden tumors (which are less likely to be escaped) usually have more copy number alterations[7], we found that escaped(+) tumors have also more amplifications and deletions than escaped(-), suggesting a global relaxation of selection on these genomic events".

Minor Comments

The font in the figures throughout the paper (starting with the cancer types in 2A and 2B) is in many cases extremely small. Please ensure the journal's minimum font size recommendations are followed.

R: In our new version of the manuscript we have improve the font size to follow minimum standard sizes (5pt).

Please define MSS, MIS, and POLE in the main manuscript text at first use.

R: We have modified the text accordingly to define these tumor subtypes at first use.

It is confusing to refer to immune edited as (edited-) on line 298. Should this be escape- or edited+?

R: We agree with the reviewer that this notation can be confusing. We have modified this section and refer immune-escaped tumors as escaped+ and non-escaped as escaped-.

Please revise the incomplete sentences starting with “An” on line 316 and the sentence starting with “Similar” on line 323.

R: This has been done.

On line 411, the authors are referring to the Chi-squared analysis in 4D, but 4C is used in the text.

R: Corrected.

The y-axis in 5B should include Immune dN/dS.

R: We have added immune dN/dS to the figure.

Reviewer #3:

Remarks to the Author:

Zapata and colleagues present an analysis of immunoediting dynamics across cancers. Using 8543 primary tumours and 356 metastatic tumours, the authors explore the extent and importance of immune editing. The authors suggest negative selection is pervasive, and that the extent of immune editing can be prognostic and predictive of response to immune checkpoint blockade.

Given that there is considerable debate in the literature regarding the extent of neo-antigen directed immune editing and, moreover, the extent of negative selection, this is an important and timely analysis. However, this also means that it is absolutely imperative that the conclusions are justified, the methods are extremely rigorous, and validation is performed. As it stands, I think each of these areas require some additional work. Below, I have suggested some additional analysis that I think could help assess the strength of the findings. I would also suggest further validation would considerably strengthen this manuscript.

We thank the reviewer for their positive assessment of our work and for their statement that this is a timely and important analysis. In the revised version of the manuscript, we fully address the reviewer's suggestions for additional analyses and details on the methodologies.

General comments:

- the definitions of immune editing and immune escape doesn't quite fit within the classical definitions as outlined by Schreiber. While I think I understand the distinction the authors are making, i think it would be useful to rephrase to avoid confusing readers.

R: We agree with the reviewer on the importance of the definitions. We used immune-editing to describe the dynamic process of selection against neoantigen-carrying clones, not the full process of elimination, equilibrium and escape as described by Schreiber. We added a paragraph in the introduction including these concepts and defined the two possible outcomes of this process, namely immune-edited versus immune-escaped clones (Lines 51-58): *"The classical model of immunoediting determines three major phases, elimination, equilibrium and escape. During the elimination phase, tumors will progressively remove all antigenic mutations generating immune-edited clones. During equilibrium and escape, multiple*

immune evasion mechanism may arise allowing the accumulation of antigenic mutations in immune-escaped clones. However, the prevalence of immune-edited and immune-escaped tumors and their impact on signatures of selection and clinical outcomes remains unknown."

- The authors state that "At a cohort level, we observe that the majority of tumours undergo neutral evolution as a consequence of immune escape, which masks ongoing immune selection in the non-escaped minority of cases." I do not wish to delve into a debate regarding neutral evolution. However, I think it is important to clarify that this does not mean the tumour is exclusively a result of neutral evolutionary dynamics. I do not believe the authors are trying to suggest this - but I think it is important to make clear for the reader.

R: We apologise for the unclear sentence. The sentence was only meant to imply that the accumulation of mutations is neutral in immunopeptidome regions. This does not imply that the tumour is evolving neutrally overall. We have modified this sentence (Lines 136) to: *"At a cohort level, we observe that the majority of tumours accumulate neoantigens neutrally as a consequence of immune escape. When escaped tumours are considered together with non-escaped tumours, the signal of ongoing immune selection in the non-escaped minority of cases is masked by the neutral signal from the immune escaped majority"*.

- SOPRANO requires further validation. The extent to which the tool is different from the original publication is quite unclear. Overall, I think a more in-depth comparison between why the results differ between SOPRANO and the recently published results of Van Den Eyden would be very useful.

R: We thank the reviewer for this comment. We have rephrased the manuscript (lines 126-128) as *"The original version of the method lacked a normalisation step to account for the heterogeneity in genome composition. Here, we implemented a trinucleotide-based mutational context correction (Somatic Substitution Bias using 192 trinucleotides or SSB192)." We hope this clarifies the main difference from the original version.*

We remind the reviewer that the first section of the manuscript was dedicated to compare immune selection from our method with the results provided in the study of Van Den Eynden (Main Figure 2C, new supplementary Fig 1G). Here, it is important to note two things:

- As the VDE study did not provide a method to calculate immune selection, we used their published metrics and compared their results with our immune dN/dS. Our method was consistent with their strategy, dN/dS results were strikingly similar and the major differences come from the choice of the predefined set of regions associated to the immunopeptidome (HLA-A0201 or protoHLA).
- Next, we argued that removing escape-rich tumors will increase the association between immune selection and lymphocytic infiltration. Then, besides our own immune dN/dS, we used their metric to demonstrate that Rooney's CD8 T cells (Main Figure 2F and G) and cytolytic activity (Supp Fig 2A), Danaher's CD8 T Cells (Supp Fig 2B) and Thorsson's lymphocyte infiltration score (Supp Fig 2C) increased their significance.

However, to further validate our strategy with that from VDE, we have now included a patient specific comparison of mutations found in the ON and OFF regions using both strategies and added the following sentence (Lines 192): *"In addition, when comparing mutation burden between these two approaches, we found a small difference on the number of nonsynonymous mutations inside the immunopeptidome but not outside (Fig S1H, Pearson $r=0.93$ vs Fig S1I, Pearson $r=0.98$)"*.

Response Figure 15 (New Supp Fig 1H,I). Nonsynonymous and synonymous mutation burden comparison between the immunopeptidomes used in this publication and those found in Van Den Eynden et al 2019)

- In general, a more detailed explanation of the underlying logic behind SOPRANO is needed, even if this has been published before. For instance, it would be good to clarify the following:

1. a mutation can alter a peptide from non-binding to binding, in this case will it not be considered? (this is immunogenic, but will be missed)

R: We thank the reviewer for this important point. We have currently expanded the introduction to SOPRANO in the manuscript (Lines 155) and provided an extended description of the method (Lines 914-969).

In brief, we developed SOPRANO aiming to estimate dN/dS inside and outside any predefined region from a set of transcripts. This region could be a protein domain shared by multiple transcripts, a binding motif, a drug receptor sequence, or as in this manuscript, a set of native peptides with binding affinity to an HLA-allele. SOPRANO requires two mandatory arguments: a) the annotated Variant Effect Predictor output file with somatic calls, and b) the bed file containing the target transcript regions. These two files can be filtered a priori based on user choice, such as expression or binding affinity of the somatic mutation or the immunopeptidome.

To estimate immune selection, we reasoned that the self-immunopeptidome provides the background region where somatic mutations will be under immune selective pressure. We describe this in our manuscript (Lines 90-99): *“as nonsynonymous mutations can also generate neoantigens by transforming self-peptides to non-self peptides, we hypothesized that immune selection can be measured by calculating dN/dS on the self-immunopeptidome (Immune dN/dS). The self-immunopeptidome can be defined as all genomic regions that generate peptides that bind to the major histocompatibility complex (MHC) and therefore are natively exposed to the immune system. These self-peptides do not elicit an immune response due to central tolerance, but when somatically mutated they should be recognised as non-self and potentially initiate an immune reaction imprinting signals of selection on the genome.”*

However, we recognize that it is possible that mutations outside the target region also change a peptide from non-binding to binding and these events will not be accounted (mutations inside are always counted).

As other studies have focused on estimating selection using all observed somatic neoantigens, we explored whether our immune dN/dS correlates with these metrics. Specifically, we obtained immune selection estimates from Marty et al[11] where to demonstrate that MHC complexes restrict the emergence of oncogenic mutations during cancer evolution they develop a metric called odds ratio (OR). OR indicates the probability of neoantigen presentation based on the MHC complex of each patient. Thus, we compared our immune dN/dS with OR, and observed a significant correlation between both values across TCGA cohort (Response Fig 16), suggesting that both metrics provides similar estimates of immune selection but using two different approaches.

In addition, as requested in another comment, we have also filtered the somatic mutation calls based on their binding affinity, and demonstrated that increasing the repertoire of peptides included in the

immunoepitome pushes the signal of immune dN/dS towards neutrality, also supporting the notion that a less specific immunoepitome weakens evidence of immune selection.

Response Figure 16. Immune dN/dS versus Odds Ratio of immune selection described in Marty et al.

2. how is a mutation that alters a peptide from binding to non-binding considered? (this is not immunogenic, but will be considered immunogenic)

R: We thank the reviewer for their point. As clarified now in the manuscript, SOPRANO calculates ON and OFF target selection based on the two input files: somatic calls and target region. It is then possible to filter mutations based on their binding affinity a priori. We explored this option by selecting 1000 patients with HLA-A0201 allele and filtered all their nonsynonymous mutations with less than 500nM affinity score (only value available in TCGA_peptideMHC_binding_MC3_v0.2.8.CONTROLLED_genomicCoord_withExpression_170403.tsv) obtained from Thorsson et al repository (<https://gdc.cancer.gov/about-data/publications/panimmune>).

We then ran SOPRANO using the filtered set of nonsynonymous mutations and the HLA-A0201 immunopeptidome and compared to the unfiltered set. We observed a significant correlation between both values (response figure 17), however the median immune dN/dS was significantly higher in the filtered set due to a significant reduction on the OFF dN/dS while the ON dN/dS was reduced slightly (Response fig 18). This result supports our assumption that the majority of mutations in the OFF-target regions are not neoantigens (and hence filtered out in this step). In summary, we consider that our study provides a conservative metric that might underestimate the true extent of immune selection in cancer.

Response Figure 17. Immune dN/dS values for unfiltered versus filtered somatic calls. Nonsynonymous mutations were filtered based on their binding affinity.

Response Figure 18. Distribution of ON, OFF and Immune dN/dS values for unfiltered versus filtered somatic calls. Nonsynonymous mutations were filtered based on their binding affinity. OFF dN/dS values reflect that the majority of nonsynonymous mutations outside the immunopeptidome are not immunogenic.

3. are only missense mutations considered?

R: dN/dS is a metric borrowed from comparative genomics where only nonsynonymous and synonymous mutations were used. In our implementation, we have also included truncating mutations as they can have an important impact on the selective process. In the future, our goal is to develop a new model that considers other mutation types such as splicing, copy number alterations, and methylation changes. For this, the major challenge would be the development of proper background mutation rate models for each mutation type.

4. most mutations are not expressed - is this taken into account beyond looking at genes that are generally expressed?

R: We thank the reviewer for their input. In this work, the immunopeptidome is composed with a core set of protein transcripts that have a minimum median and mean expression of more than 1FPKM across multiple tissue types (downloaded from the human protein atlas). This filtering reduces the number of unfiltered transcripts in the immunopeptidome in almost half (from 19000 to a bit more than 10000 transcripts). We have clarified our strategy in the manuscript, and have now also compared to a patient specific expression filter demonstrating almost identical results (Response Fig 19). Interestingly, we also demonstrate that when using an unfiltered set, immune dN/dS becomes closer to one (Response Fig 11), supporting the absence of immune selection on unexpressed variants.

We now have also provided the unfiltered immunopeptidome for those users that wish to filter based on their own expression data.

Response Figure 19 (Same as response fig 10). Immune dN/dS distribution for all individuals using the pre-filtered patient specific immunopeptidome available in GitHub (left) versus the immunopeptidome filtered by patient specific transcript expression.

- The cohort should be justified and explained in the main text. The 8543 tumours suddenly appears without any explanation? Given that immune cell scores are also available from Thorsten et al., and this includes more samples, the authors should consider analysing the full cohort.

R: Initially, we performed our analysis on the set of tumor types with previous metrics of immune selection (Van Den Eynden et al) and immune cell scores (Rooney et al) available, hence the number. However, we agree with the importance of computing immune dN/dS on all available data and compare with other sources of immune cell scores. We computed ON, OFF, and immune dN/dS using different immunopeptidomes and at two levels (patient specific or cohort) as described in Figure 1 for all ~10200 TCGA patients from 33 tumor types available. We found that at the cohort level that the leukocyte fraction and lymphocyte infiltration are significantly correlated to immune dN/dS when considering individuals

without a mutation in an escape gene (Fig S5B,5C), whereas immune dN/dS was not associated with the level of immune infiltration in tumours with an immune escape mutation (Fig S5D,E).

It is important to note that when analysing patient specific metrics, many patients are discarded due to an absence/very low number of mutations in the immunopeptidome (zero non-synonymous or zero synonymous leading to infinite or zero immune dN/dS). In cohort analysis (where mutations are aggregated across patients) the low mutation burden tumours can still contribute to the analysis.

Response Figure 20 (New Supp Fig 5B,C,D,E). Comparison of dN/dS values and leukocyte fraction obtained from Thorsson et al using all tumor types. A) Immune dN/dS versus leukocytes using all individuals. B) Immune dN/dS versus leukocytes using non-escaped tumors. C) Immune dN/dS and leukocytes using escaped tumors. D) OFF-target dN/dS versus leukocytes using all individuals.

- Why is %rank < 0.5 used? This is classically a 'strong binder' - do the results change if %rank < 2 is used instead? And, while I appreciate the goal of the paper is not to compare different binding predictions, the authors should consider adopting the latest binding prediction tools, e.g. netMHC4.1 and may wish to explore how the results change if IC50 is used instead of %rank. I appreciate this may be a large amount of work, but I think it would be useful to demonstrate that the results remain robust.

R: We thanks the reviewer for their questions. Following the reviewer suggestions, we have taken 1000 individuals with HLA-A0201 allele, and compare the impact of the netMHC version (Response Fig. 21) and the cut-off to be considered binder (Response Fig 22). We have also clarified our strategy to develop the immunopeptidome and included the following in the manuscript:

“We selected all possible peptides with a %rank < 0.5, considered strong binders, as the unfiltered immunopeptidome. This cut-off was selected based on previous evidence stating that truly immunogenic neoantigens are enriched in strong binders. We reasoned that if a more relaxed cut-off was chosen (i.e %rank < 2), specificity will be compromised reflected on a more neutral immune dN/dS. We validated this by selecting a set of 1000 individuals with HLA-A0201 allele and run SOPRANO on two HLA-A0201 immunopeptidomes (%rank < 0.5 versus %rank < 2). In addition, we compare whether our results were affected by the version of netMHCpan and results were highly correlated despite a slightly higher immune dN/dS for the latest version.”

Respect to which binder metric to use, we understand that rank has been widely demonstrated as a more robust metric than IC50[12]. To demonstrate this, we plot the relationship between rank and affinity for different HLA-alleles and observed that the absolute value of IC50 is highly dependent on the HLA-allele, creating a difficulty if one wishes to mix the patient specific immunopeptidome of 6 alleles. In contrast, rank is agnostic of the allele and allow us to mix different regions coming from different alleles with confidence (Response figure 23).

Response Figure 21. Immune dN/dS calculated on strong binders predicted using netMHC4.1 and netMHC4.0.

Response Figure 22 (New Supp Fig 1J). Immune dN/dS when using immunopeptidome predicted with Strong binder only or together with weak binder for expression based unfiltered and filtered datasets.

Response Figure 23. Binding affinity versus rank for multiple HLA alleles.

- The authors calculate the ratio between ON-dN/dS and OFF-dN/dS. This may mask some signal. For instance, both may show significant negative selection, or significant positive selection, and the ratio is less informative? A detailed exploration of key differences would be useful here.

R: We thank the reviewer for their comment. We now clarified our logic behind the use of the ratio between ON and OFF dN/dS and improve our description in the manuscript methods section (Lines 1185): *“To correct for biases arising from technical artefacts (germline contamination, over filtering, false positive somatic calls) that can push up or down the neutral dN/dS expectation (~1) and excluding driver genes, we calculated the ratio between dN/dS using mutations inside (ON-dN/dS) and mutations outside the immunopeptidome (OFF-dN/dS), and defined this ratio as immune dN/dS. Immune dN/dS is a normalized metric of immune selection that assumes most of mutations outside the immunopeptidome are non-immunogenic. This assumption is supported by the fact that only 10% of missense mutations leads to neoantigens with high recognition probability, thus the majority of mutations outside the immunopeptidome (excluding driver genes) should be under neutral dynamics (dN/dS ~ 1).”*

To exemplify this, we have added the plot for ON, OFF and immune dN/dS for the whole TCGA cohort (Response Fig 24). We find most tumor types with OFF regions displaying positive selection signals at the cohort level. This could reflect either that the majority of somatic calls have been over filtered, or that there is a large proportion of unknown driver events, both cases pushing dN/dS values above one. A particular example is melanoma which display a strong signal of negative selection in the OFF dN/dS. This could be due to a technical correction bias (not including a pentanucleotide model for correction), or that melanomas are highly immunogenic and combined with their high mutation rate the likelihood of mutations leading to neoantigens outside the immunopeptidome is more likely.

Response Figure 24 (New Supp fig 1A,B,C). ON, OFF and immune dN/dS values for all 33 tumors types.

- The immune dNdS is a very important part of the paper. As such, the explanation needs to be much clearer. I think a simulated data-set could be useful here to demonstrate expectations.

R: We agree with the reviewer on the importance of the immune dN/dS concept to our manuscript. We have rephrased the manuscript to clearly state that dN/dS can be calculated in any genomic region, and our definition of immune dN/dS refers to the calculation of dN/dS in a predefined immunopeptidome.

We have indeed developed a dN/dS simulator (manuscript in revision at another journal), that explores the dynamics of the immune dN/dS metric in the presence of an active immune system (i.e. Naïve CD8 T cells). The simulator is based on a stochastic branching process, and includes model A: a single cell model, where a single nonsynonymous mutation in the immunopeptidome in a single cell is sufficient to generate an immune response and model B: a clonal model, where a nonsynonymous mutation in the immunopeptidome needs to reach a certain clone size to be detected by the immune system.

The figure (Response figure 25) below taken from our simulations indicate that at increasing levels of immune activity, immune dN/dS approaches to zero for two different models (A: Single cell immunogenicity, B: Clone size dependent immunogenicity). In addition, we demonstrate that immune dN/dS remains lower than one even when using mutations above certain allele frequency threshold (1%, 5%, 10% or 50% minimum allele frequency) when the immune system is active.

In addition, we simulated a mixture of immune-edited and immune-escaped tumors demonstrating that in cohort studies (Response Figure 26), escaped tumors will mask the signal of the immune-edited tumors, providing an explanation for the low number of tumors with significantly low immune dN/dS.

Response Figure 25. Simulation of immune dN/dS and immune activity under different allele frequency cut-offs, for different strengths of immune attack. This figure features in a related manuscript currently under review at a different journal.

Response Figure 26 (Main Fig 1). Simulation of immune dN/dS for different mixtures of escaped versus non-escaped patients. A) Cohort estimates B) individual estimates.

- to what extent does immune dN/dS correlate with mutation burden?

R: In our analysis, we show the dependency between immune dN/dS and TMB in several situations. We mainly demonstrate that immune dN/dS does not correlate with TMB in non-escaped (escaped-) tumors but it does in escaped+ tumors reflecting an immune neutral accumulation of nonsynonymous mutations (Response fig 28). In escaped tumours, the magnitude of the deviation of immune-dN/dS from 1 tumors does correlate with TMB. This happens because prior to immune escape, immune-dN/dS was less than 1, and following immune escape mutations accumulate neutrally in the immunopeptidome, and so immune-dN/dS tends towards 1 as more and more mutations accumulate.

Response Figure 27 (New Supp Fig X). Immune dN/dS versus TMB for non-escaped versus escaped Microsatellite stable tumors using clonal and all mutations.

Response Figure 28 (New Supp fig 4B). Immune dN/dS distance to neutrality versus TMB for non-escaped and escaped tumors.

- the dNds analysis should permit an exploration of potentially how many neoantigens have been edited per tumour - what do the numbers suggest?

R: We thank the reviewer for this suggestion. We have calculated using the formula in lines 955 to estimate the number of neoantigens edited. The results are part of the supplementary table 1 and we have added the following sentence to the results (Lines 169-170): “the range of depleted neoantigens varied from 0 in all tumors with immune dN/dS above one to 151 in uterine carcinoma with a median of 8 neoantigens (Supp. Table 1)”

Table 2 (Added to Supp Table 1). Number of neoantigens removed by immune selection

Tumor Type	ACC	BLCA	BRCA	CESC	CHOL	CRC	DLBC	ESCA	GBM	HNESC	KICH	KIRC	KIRP	LAML	LGG	LIHC	LUAD	LUSC	MESC	OV	PAAD	PCPG	PRAD	SARC	SKCM	STAD	TGCT	THCA	THYM	UCEC	UCS	UVM
NeoantigensRemoved	0	56	21	10	8	55	0	27	0	10	0	10	14	0	0	10	124	64	13	7	0	0	0	0	131	9	0	0	0	151	5	0

- Why not restrict the analysis to patients with HLA-A0201? (I would suggest either doing this or accurately computing the HLA alleles for each patient and using more precise scores)

R: Our manuscript explores HLA-A0201, protoHLA and patient specific HLA as possible immunopeptidomes. We have also performed the analysis using cohort or patient-specific analysis. These combinations were described in table response 3 (also main table 1). To demonstrate the differences between filtering strategies and selection, we have followed the suggestion of the reviewer and calculated immune dN/dS only in patients with HLA-A0201 allele.

Table 3. Analysis of tumor datasets with SOPRANO

Dataset	Tumor type	#Patients	Immunopeptidome	dN/dS
TCGA	33 Tumor types	10172	HLA-A0201, Proto-HLA, Patient Specific	Cohort/Individual
MSI	CRC, STAD, UCEC	1507	Patient specific	Individual
Hartwig	Multiple	308	HLA-A0201, Patient specific	Mixed
Riaz	Melanoma	68	Patient specific	Mixed

- The authors find that CD8 T cell abundance is negatively correlated with immune dN/dS - this is very intriguing - is this the case within cancer types, or when using all together? A model taking into account cancer type would be useful here and other potential confounders would be useful here. Is there no correlation between non-immune dNdS and CD8 T cell abundance?

R: This is an important point that we have thoroughly re-analysed in this new version (See response figure 3 and 20). We have included new supplementary fig 5 exploring multiple associations between immune dN/dS and lymphocyte infiltration.

As we described in the manuscript we observed this association at the population level (using all together) when excluding escape-rich tumors (CRC, STAD and UCEC), or when excluding patients labelled as escaped (Supp Fig 2, Supp fig 5). We now included all tumor types in this analysis as well and observed that the signal is still significant for immune dN/dS, is stronger for ON dN/dS and not significant for OFF dN/dS (Response Figure 30, New supp Fig 3A,B,C).

Response Figure 29 (New supp fig 3A,B,C). Immune dN/dS, ON dN/dS and OFF dN/dS using all 33 tumor types.

- How do the results change if CRC, STAT and UCEC are included? Further justification for excluding/including whole sets of tumours is needed.

R: We have modified the manuscript to clarify this (lines205 - 210): “We hypothesized that the low number of tumour types displaying strong immune selection is due to a large proportion of immune-escaped tumours masking the signal. We recently reported a high frequency of immune evasion events in three tumour types, colorectal (CRC), stomach (STAD), and uterine cancer (UCEC) and so we first compared the association between immune dN/dS with immune infiltrates when including or excluding these tumor types.”

Response Figure 30 (Supp Fig 5A). Multivariable analysis for association of immune population and dN/dS when escape-rich tumors are included.

- The analysis ignores copy number alterations in immune genes, does this improve the associations?

R: We thank the reviewer for this suggestion. We obtained copy number status of escape genes from GDC (TCGAbiolinks) in three MSI-rich tumors. Interestingly, as we can observe in response figure 32, the most prevalent changes are copy number gains in these genes. As we cannot associate a loss of function of the gene function to a copy gain, and most the genes linked together are affected at chromosome level, we focused on deletions on these genes. We estimated the number of genes deleted per patient, and every patient with a number of genes larger than the median number of genes deleted was classified as escapedCNV deletion. We performed a similar strategy for gains and compared the immune dN/dS values

between escaped and non-escaped based on the CNV status and point mutations in escape genes (Response Figure 33).

Response Figure 31 (Supp Fig 7). Landscape of copy number variation in escaped genes for four tumor types.

Response Figure 32 (Supp Fig 4D). Immune dN/dS for escaped versus non-escaped based on copy number deletions/gains or missense/truncating mutations in MSI-rich tumor types.

- The authors make a strong statement that the "reported lack of neoantigen depletion signals on the genome is an artefact of the heterogenous nature of cohorts that either have a large proportion of immune-escaped tumours (e.g. colorectal cancer) or come from immune-privileged tissues (Brain tumours)." It would be useful to more directly assess whether this is the case if other methods beyond SOPRANO are used here too.

R: We agree with the reviewer and have modified this paragraph to "In summary, by removing tumors enriched in immune escape events we corroborated the association between immune activity and immune selection, suggesting that the reported lack of neoantigen depletion signals on the genome is an artefact of the heterogenous nature of cohorts that have a large proportion of immune-escaped tumours (e.g. colorectal, endometrial and stomach cancers)". This statement is supported by our simulations that

indicate that if 75% of individual escaped tumors are mixed with non-escaped, the signal of immune dN/dS will not be significantly different from neutrality (See response fig 18 above).

- cut-offs with regards to number of synonymous mutations seems a little odd - is there a danger that this is enriching the cohort for certain types of tumours? It would be useful to explore different cut-offs more clearly.

R: We thank the reviewer for this comment. Our simulations indicate that immune-edited tumors with no synonymous mutations in the immunopeptidome will always (falsely) report a signal of positive selection in the immunopeptidome; when there is one synonymous mutation 25% of cases report positive selection, and when there are 2 or 3 synonymous, this value is reduced to 12% and 4% respectively (Response fig 34). To determine whether we enriched for some tumor types due to this cut-off, we plot the number of samples left after removing patients with 0, 1, and 2 synonymous mutations in the immunopeptidome (Response fig 35). We observed no large differences across tumor types.

Response Figure 33. Simulation of immune dN/dS at different number of synonymous mutations in the immunopeptidome.

Response Figure 34. Number of mutations when filtering tumors with 1, 2 or 3 synonymous mutations in the immunopeptidome.

- I'm a little unclear why HLA type isn't available for the entire cohort?

R: We apologise for the confusion here. In the earlier versions of our manuscript we did not have the HLA calls from Hartwig Medical foundation patients. During the course of this project, we were able to run Optitype to get the patient's specific set of HLA alleles. We have now removed this sentence from the manuscript to avoid confusion.

- I think it would be useful to explore how the analysis of response to ICB looks in the context of the full cohort (despite potential caveats).

R: As the Hartwig and the Riaz cohorts come from different clinical trials and sequencing protocols, they have different clinical annotations, metadata, sequencing depth, coverage, technology and, as the reviewer suggest, other potential caveats. However, we have consolidated our estimation of immune dN/dS and escape status for both cohorts and test a multivariate predictor considering Age, gender, immune dN/dS and escape status. The results are as shown below for the reviewer and confirm that immune-dN/dS is a strong predictor of ICB response (Response fig 36).

Response Figure 35 (New Supp Fig 14). Survival curves for HMF and Riaz cohort combined at different cut-offs for synonymous mutations.

- the results would be considerably strengthened by the inclusion of a validation cohort - and demonstration this is clearly independent of cancer type

Response Figure 36. Cox proportion hazard ratios for models excluding or including escape status and immune dN/dS when considering cancer subtype as an additional variable.

R: We agree with the reviewer that such analysis will strengthen the clinical potential of our study, but we do not have access to a third such dataset. Reassurance that immune-dNdS measures meaningful biology is given by comparing the efficacy of the best predictive biomarkers found in Litchfield et al (CXCL9 and clonal TMB) to a multivariate model that also includes escape and immune dN/dS (Response fig 37). As shown in the figure, the inclusion of escape status and immune-dNdS (dist2N) improves the model performance. Nevertheless, we recognise that further studies are required to fully validate these two

metrics for wide applicability in a clinical setting. We are currently recruiting and collecting data within our institution to achieve this aim.

- the authors should explore the extent to which immune editing is also prognostic in TCGA in tumours without ICB treatment.

R: This is an interesting comment. We tested whether immune editing is prognostic in TCGA patients (response fig 38). From a subset of tumors, we identified that only cervical and uterine carcinomas have a significant difference between immune-edited (defined as low immune dN/dS and escaped-) and escaped (defined as neutral immune dN/dS and escaped+). Interestingly, both of these tumors are anatomically close suggesting a common immune mediated selective process. Nonetheless, when considering age or tumor stage immune dN/dS was not prognostic in any cancer. To note, given that TCGA patients have not been selected for immunotherapy treatment, it is difficult to properly perform an analysis establishing the prognostic value of immune dN/dS in this set of primary untreated tumors.

Response Figure 37. Survival curves for TCGA data when classifying patients into immune-edited and immune-escaped patients.

DATA and code:

- I'm afraid I couldn't manage to navigate to where the data and code is to reproduce the figures and analysis. Although I have little doubt the authors will make this available - I need this to properly review the manuscript.

R: We apologise for this, the code and data to reproduce our analysis is deposited in [synapse.org](https://www.synapse.org) repository under the Sottoriva lab section – EVOESCAPE_ALLDATA: <https://www.synapse.org/#!Synapse:syn22150271>

- [1] N. McGranahan and C. Swanton, "Clonal Heterogeneity and Tumor Evolution: Past, Present, and the Future.," *Cell*, vol. 168, no. 4, pp. 613–628, 2017.
- [2] R. D. Schreiber, L. J. Old, and M. J. Smyth, "Cancer immunoediting: Integrating immunity's roles in cancer suppression and promotion," *Science (80-.)*, vol. 331, no. 6024, pp. 1565–1570, 2011.
- [3] J. Van den Eynden, A. Jiménez-Sánchez, M. L. Miller, and E. Larsson, "Lack of detectable neoantigen depletion signals in the untreated cancer genome," *Nat. Genet.*, vol. 51, no. 12, pp. 1741–1748, 2019.
- [4] F. Avila Cobos, J. Alquicira-Hernandez, J. E. Powell, P. Mestdagh, and K. De Preter, "Benchmarking of cell type deconvolution pipelines for transcriptomics data," *Nat. Commun.*, vol. 11, no. 1, 2020.
- [5] C. S. Grasso, M. Giannakis, D. K. Wells, T. Hamada, X. J. Mu, M. Quist, J. A. Nowak, R. Nishihara, Z. R. Qian, K. Inamura, T. Morikawa, K. Nosho, G. Abril-Rodriguez, C. Connolly, H. Escuin-Ordinas, M. S. Geybels, W. M. Grady, L. Hsu, S. Hu-Lieskovan, J. R. Huyghe, Y. J. Kim, P. E. Krystofinski, M. D. Leiserson, D. J. Montoya, B. B. Nadel, M. Pellegrini, C. C. Pritchard, C. Puig-Saus, E. H. Quist, B. J. Raphael, S. J. Salipante, D. S. Shin, E. Shinbrot, B. Shirts, S. Shukla, J. L. Stanford, W. Sun, J. Tsoi, A. Upfill-Brown, D. A. Wheeler, C. J. Wu, M. Yu, S. H. Zaidi, J. M. Zaretsky, S. B. Gabriel, E. S. Lander, L. A. Garraway, T. J. Hudson, C. S. Fuchs, A. Ribas, S. Ogino, and U. Peters, "Genetic mechanisms of immune evasion in colorectal cancer," *Cancer Discov.*, p. CD-17-1327, 2018.
- [6] K. Litchfield, J. L. Reading, C. Puttick, K. Thakkar, C. Abbosh, R. Bentham, T. B. K. Watkins, R. Rosenthal, D. Biswas, A. Rowan, E. Lim, M. Al Bakir, V. Turati, J. A. Guerra-Assunção, L. Conde, A. J. S. Furness, S. K. Saini, S. R. Hadrup, J. Herrero, S. H. Lee, P. Van Loo, T. Enver, J. Larkin, M. D. Hellmann, S. Turajlic, S. A. Quezada, N. McGranahan, and C. Swanton, "Meta-analysis of tumor- and T cell-intrinsic mechanisms of sensitization to checkpoint inhibition," *Cell*, vol. 184, no. 3, p. 596–614.e14, 2021.
- [7] G. Ciriello, M. L. Miller, B. A. Aksoy, Y. Senbabaoglu, N. Schultz, and C. Sander, "Emerging landscape of oncogenic signatures across human cancers," *Nat. Genet.*, vol. 45, no. 10, pp. 1127–1133, 2013.
- [8] I. Martincorena, K. M. Raine, M. Gerstung, K. J. Dawson, K. Haase, P. Van Loo, H. Davies, M. R. Stratton, and P. J. Campbell, "Universal Patterns of Selection in Cancer and Somatic Tissues.," *Cell*, vol. 171, no. 5, p. 1029–1041.e21, 2017.
- [9] E. Lakatos, M. J. Williams, R. O. Schenck, W. C. H. Cross, J. Househam, L. Zapata, B. Werner, C. Gatenbee, M. Robertson-Tessi, C. P. Barnes, A. R. A. Anderson, A. Sottoriva, and T. A. Graham, "Evolutionary dynamics of neoantigens in growing tumors," *Nat. Genet.*, vol. 52, no. 10, pp. 1057–1066, 2020.
- [10] G. Caravagna, T. Heide, M. J. Williams, L. Zapata, D. Nichol, K. Khkhaidze, W. Cross, G. D. Cresswell, B. Werner, A. Acar, L. Chesler, C. P. Barnes, G. Sanguinetti, T. A. Graham, and A. Sottoriva, "Subclonal reconstruction of tumors by using machine learning and population genetics," *Nat. Genet.*, vol. 52, no. 9, pp. 898–907, 2020.
- [11] R. Marty, S. Kaabinejadian, D. Rossell, M. J. Slifker, J. van de Haar, H. B. Engin, N. de Prisco, T. Ideker, W. H. Hildebrand, J. Font-Burgada, and H. Carter, "MHC-I Genotype Restricts the Oncogenic Mutational Landscape," *Cell*, vol. 171, no. 6, p. 1272–1283.e15, 2017.

- [12] S. Paul, D. Weiskopf, M. A. Angelo, J. Sidney, B. Peters, and A. Sette, "HLA class I alleles are associated with peptide-binding repertoires of different size, affinity, and immunogenicity," *J. Immunol.*, vol. 191, no. 12, pp. 5831–5839, Dec. 2013.

Decision Letter, first revision:

14th Jun 2022

Dear Dr Zapata,

I apologise for the delay in returning this decision to you.

Your Article, "Immunoediting dynamics determine tumour immunogenicity and drive response to checkpoint inhibitors" has now been seen by 3 referees. You will see from their comments below that while Reviewers #1 and #2 are satisfied with your revisions, Reviewer #3 continues to raise some important points. We are interested in the possibility of publishing your study in *Nature Genetics*, but would like to consider your response to these concerns in the form of a revised manuscript before we make a final decision on publication.

Overall, you'll see that Reviewer #3 continues to think that the work is potentially field forwarding. However, they have voiced some issues with the overall rigour of the analyses (for example, with respect to the need for additional validation for SOPRANO) and they are also concerned that not all your claims are sufficiently supported by your data.

Reviewer #3 represents an ideal 'end-user' of your work and as such, we are keen that the final product is convincing to them. As you know, you are operating in a contentious space, and as we realise that the paper will likely be provocative, it's especially important that the study is completely robust before we can commit to moving forward. As such, we encourage you to consider Reviewer #3's comments carefully, and address them in full.

Please do not hesitate to get in touch if you would like to discuss these issues further.

We therefore invite you to revise your manuscript taking into account all reviewer and editor comments. Please highlight all changes in the manuscript text file. At this stage we will need you to upload a copy of the manuscript in MS Word .docx or similar editable format.

*2) If you have not done so already please begin to revise your manuscript so that it conforms to our Article format instructions, available [here](http://www.nature.com/ng/authors/article_types/index.html). Refer also to any guidelines provided in this letter.

[redacted]

We hope to receive your revised manuscript within four to eight weeks. If you cannot send it within this time, please let us know.

Sincerely,

Safia Danovi
Editor
Nature Genetics

Reviewers' Comments:

Reviewer #1:

Remarks to the Author:

Most of my comments have been addressed and the manuscript has been well improved. The manuscript could potentially benefit from further validation, but I do not see this as a big issue. I would suggest the authors state the limitation of immune deconvolution analysis as the correlation between immunopeptidome dN/dS and immune metrics was largely based on immune deconvolution of bulk expression data.

Reviewer #2:

Remarks to the Author:

The authors have comprehensively responded to the extensive comments and critiques of all reviewers and have greatly improved the manuscript as a result. The concerns I raised in my major and minor comments to the authors have been adequately addressed. I strongly recommend the article for acceptance.

Reviewer #3:

Remarks to the Author:

In my original review, I commented that 'Given that there is considerable debate in the literature regarding the extent of neo-antigen directed immune editing and, moreover, the extent of negative selection, this is an important and timely analysis. However, this also means that it is absolutely imperative that the conclusions are justified, the methods are extremely rigorous, and validation is performed. As it stands, I think each of these areas require some additional work.' The authors have certainly strengthened their manuscript. However, I still feel that further work is required to remove ambiguity both in the text and analysis. I do not question that this is potentially important work, however, I continue to have concerns regarding the rigor of a) the SOPRANO method development & benchmarking and b) the analyses performed.

In my original review I stated that SOPRANO requires further validation. And, I still feel this is the case. The authors have gone some way to address this concern, but more is needed. For instance, a strength of SOPRANO is the reference database of the genomic coordinates of peptides expected to bind to each HLA. However, no comparison of SOPRANO performance using patient-specific HLA vs prototype vs HLA-A0201 was performed, nor were these conditions benchmarked against HBMR scores for the same patients. The HLA-specificity offered by SOPRANO will be most relevant when measuring immune selection in individual tumours (rather than at the cohort level), however the authors do not see a correlation between immune infiltrate and immune dN/dS in their individual tumour TCGA analysis (lines 258-259). Conversely in the Lakatos et al. cohort patient specific immune dN/dS vs CD8 T cell infiltration was significantly associated in MSS escaped- tumours but not in MSS escaped+ (Fig 3F, lines 396-398). What differences between these analyses account for the discrepant results? The accuracy of patient-specific HLAs? Availability of paired RNAseq to do patient-specific expression filtering? These will be important limitations to acknowledge for future users of the tool.

Related to the above. While I appreciate that the authors agree with the value of a simulated dataset to explore cancer immune evolution - the response to the comment is not particularly helpful for the review process. Either the authors should provide the full details of the simulations (and preferably also include this manuscript), or else I cannot really review this aspect and it simply highlights that the authors agree more work is needed but do not feel the work should appear in this manuscript. Overall, I think simulations would be very useful in this manuscript too.

A weakness of the paper is the lack of consistent/consensus definition of 'escaped' tumors. For example, tumors with HLA LOH are counted as escaped in the Lakatos et al. cohort but not identified in the TCGA cohort. Given the central argument that the fraction of escaped tumors will obscure signals of immune editing, reporting the prevalence of such escaped tumors (in each TCGA tumor type for example) would be important and is not done here. In general, it would be useful to unequivocally demonstrate that the results reflect key differences in immune edited tumors, not simply mutation burden. For instance, if the authors repeat the same analysis using synonymous mutations in immune genes - do they observe a completely different pattern? Or if they use an equivalent gene set (i.e. similar size), but of random genes?

More generally, the authors classify each tumor as "escaped" if there was a missense or a truncating mutation in one of the escape genes. Using the dN/dS scores for these genes the authors should be able to estimate the likely number of false positives. Indeed, it seems likely that many of the 'escaped' tumors are not really escaped as many missense mutations will effectively be passengers. This should be commented on.

The authors helpfully added the following sentence: The range of depleted neoantigens varied from 0 to 151 with a median of 8 neoantigens edited in each tumor type (Supp Table 1). Unless I am misinterpreting these results, this suggests that neoantigen immune editing; i.e. selection against neoantigens does not have a profound impact on the accumulation of mutations. For instance, in LUAD, the tumor type with by far the strongest signals of Immune dNdS, the authors report loss of 124 neoantigens. This suggests in $>3/4$ of the cohort there are likely no neoantigens lost whatsoever. Likewise, in HNSC it is $<5\%$ of the cohort that have any evidence of loss of neoantigens. I'm not sure how this squares with the importance of immune dN/dS? Furthermore, it would be important to directly evaluate this in the context of immune edited/unedited tumors. A figure showing the balance/mix of edited to non-edited tumors would be useful - and how many neoantigens are likely removed.

The statement 'To date, the only FDA-approved biomarkers of response include tumor mutation burden (TMB) and mismatch repair (MMR) proficiency status' (lines 72-74) omits the other key biomarker used to guide immunotherapy treatment decisions: immunohistochemistry PD-L1 status. This biomarker should be considered in any predictive/survival analyses including tumor types where PD-L1 status guides immunotherapy treatment.

Statistical tests - on a number of occasions they authors show comparisons of p-values to indicate significance. I fear on occasion this is overstating the importance of their findings. For instance, the authors state their immune dN/dS is more significant than TMB - but the authors are comparing different sets of tumors, and moreover do not perform a formal comparison of the methods. Likewise, the authors state that "We found that our multivariable model including escape status and immune dN/dS had better prognostic value (Response fig 9, concordance 0.74, Log-rank P-value $1e-06$, and AIC 446), compared to when these variables were not included (0.72, log-rank P-value $4e-06$ and AIC

452).” while this may be true - I would assume these models are not significantly different from each other; i.e. we cannot conclude with confidence one is better than the other. In general, it would be useful to show p-values rather than p=ns

Response Figure 21. Immune dN/dS calculated on strong binders predicted using netMHC4.1 and netMHC4.0. - Without further explanation, I find this figure very difficult to interpret.

The finding that SKCM displays significant negative selection in OFF dN/dS is exactly the reason why exploring a ratio of ratios may be misleading. I think in this respect the immune dN/dS has the potential to confuse, and it would be clearer to focus on ON dN/dS while also reporting OFF dN/dS. Immune dNdS masks important results.

The code for SOPRANO is publicly available in GitHub, however the running scripts are filled with hardcoded paths that make reproducibility particularly challenging. Also there are multiple versions and scripts available so it is unclear which should be used.

Minor comments

If patients with low numbers of synonymous mutations need to be removed from the analysis due to the fact that the resulting scores are inaccurate (10% of patients in the analysis, lines 436-438), ideally this inaccuracy would be captured statistically (e.g. by wide confidence intervals around the scores) or formally addressed in another way. Documentation of this limitation must be made clear for future users.

Could further information be provided regarding how intronic mutations are used to estimate the OFF-dN/dS rates as these will have different background rates to synonymous mutations?

Fig 1B does not clearly explain what HLA-alleles are used in the prototype or why these were selected (this is only explained later in the text). If the 6 HLA-alleles in the figure are the prototype, is the prototype homozygous for HLA-A0201 or is this a typo? The HLA nomenclature is different in the text than the figure (:). Fig 1C does not acknowledge that escaped tumors can be immune-edited or not.

There is still insufficient definition of terms in the text: ‘weak binders’ (line 202) – what thresholds were used?, ‘unexpressed regions’ (line 202) – using the cohort expression filter or patient-specific expression profiles?, global dN/dS (OFF) – does ‘global’ = ‘OFF’? (line 227 figure legend), ‘the median abundance for each immune subtype’ (line 232, Fig 2C reference in text does not correspond to figure) – Histologically scored? What was quantified? Total immune infiltrate or were specific markers for immune cell types included?, ‘copy number status’ – of the immune genes? Del and amp status or just Del?, ‘driver, global, escape, and immune loci’ (line 488) – each set of genes needs defining.

MMR proficient is used in the introduction then instead of MMR deficient in the results ‘MSI rich’ tumours and later microsatellite stable (MSS) (line 326) is used which may confuse readers unfamiliar with these terms. The conventional terms are MSI-high or MMR deficient. Moreover, why were entire tumour types where MSI-high tumours can occur removed rather than only tumours known to be MSI-high?

The statement ‘Given one of the major side effects of immunotherapies is autoimmunity, an upregulation of dendritic cell activity in combination with ICIs may be a good avenue to minimise autoimmunity and increase treatment efficacy’ (lines 301-303) seems unrelated/unsubstantiated by

the presented results

Fig 3C y axis should be log immune dn/ds (I think). There is a variety of immune dN/dS scales used (e.g. non-log, log, difference from 1) so these need to be clearly labeled as such. In general, for many of the figures the axis are not clearly labeled.

Author Rebuttal, first revision:

Reviewers' Comments:

Reviewer #1: Remarks to the Author:

Most of my comments have been addressed and the manuscript has been well improved. The manuscript could potentially benefit from further validation, but I do not see this as a big issue. **I would suggest the authors state the limitation of immune deconvolution analysis as the correlation between immunopeptidome dN/dS and immune metrics was largely based on immune deconvolution of bulk expression data.**

R: We thank the reviewer for their positive assessment of our work and are grateful for their input that has significantly improved our original manuscript. We have included the following statement in the discussion: "We note the limitations of our cohort analysis and uncertainty inherent to the immune deconvolution of bulk expression data used in this study."

Reviewer #2:

Remarks to the Author:

The authors have comprehensively responded to the extensive comments and critiques of all reviewers and have greatly improved the manuscript as a result. The concerns I raised in my major and minor comments to the authors have been adequately addressed. I strongly recommend the article for acceptance.

R: We thank the reviewer for their valuable contributions to our manuscript, and are delighted that they now recommend our work for acceptance.

Reviewer #3:

Remarks to the Author:

In my original review, I commented that 'Given that there is considerable debate in the literature regarding the extent of neo-antigen directed immune editing and, moreover, the extent of negative selection, this is an important and timely analysis. However, this also means that it is absolutely imperative that the conclusions are justified, the methods are extremely rigorous, and validation is performed. As it stands, I think each of these areas require some additional work.' The authors have certainly strengthened their manuscript. However, I still feel that further work is required to remove ambiguity both in the text and analysis. I do not question that this is potentially important work, however, I continue to have concerns regarding the rigor of a) the SOPRANO method development & benchmarking and b) the analyses performed.

R: We thank the reviewer for their continued thorough and very valuable critique of our study. We are pleased that they judge our work to be timely and important, and find that the strength of our study has been improved through this revision process. We address their remaining concerns below.

In my original review I stated that SOPRANO requires further validation. And, I still feel this is the case. The authors have gone some way to address this concern, but more is needed. For instance, a strength of SOPRANO is the reference database of the genomic coordinates of peptides expected to bind to each HLA.

However, no comparison of SOPRANO performance using patient-specific HLA vs prototype vs HLA-A0201 was performed, nor were these conditions benchmarked against HBMR scores for the same patients.

R: We thank the reviewer for this comment, and we indeed agree on the importance of the immunopeptidome. SOPRANO estimates selection based on the input files provided, mutation and the immunopeptidome. Importantly, the immunopeptidome is defined by the user.

To address this concern, we summarise the distributions for ON, OFF and immune dN/dS for several immunopeptidomes and pipelines (Response Figure 1, new Fig S2). Specifically, we examine: 1) the choice of netMHC4.0 versus netMHC4.1, 2) HLA-A0201 versus patient-specific HLA, 3) None, patient-specific or global expression of transcripts, 4) binding affinity of nonsynonymous events, 5) inclusion of predicted

strong versus strong and weak binders. The median value for each distribution demonstrates the impact of the chosen immunopeptidome.

The lowest immune dN/dS values (suggesting stronger immune selection) were obtained when using patient-specific estimates (F and G). The highest immune dN/dS values (indicating weaker selection) are obtained when using only HLA-A0201 predicted peptides, no transcript expression filtering, and adding weak binders (D). By either removing weak binders (B) or filtering by transcript expression (C and E), immune dN/dS is reduced (indicating stronger selection), suggesting that weak binders and non-expressed transcripts are not under immune selection. Additionally, the inclusion of IEDB T-cell positive filtering of peptides also reduces immune dN/dS (A). We are confident that these analyses strongly support the fact that immune dN/dS is measuring selection against neoantigens.

Furthermore, we noted no significant differences between filtering by global expression (F) or patient-specific expression (G). In all groups, OFF dN/dS ranged between 0.94 and 1, remained close to neutrality, and was unaffected by the choice of tool (0.98), expression (0.98), or whether strong or weak binder regions were used (OFF dN/dS \sim 1).

In a separate analysis, we used the immunopeptidome of group C (expressed strong binders of HLA-A0201) and removed all mutations that were not predicted neoantigens prior to calculating immune dN/dS. We observed that ON dN/dS was reduced by 10% (from 0.75 to 0.67), whereas OFF dN/dS was reduced by 50% (from 0.98 to 0.49), thereby indicating that the HLA-A0201 immunopeptidome is 5 times more likely to carry a neoantigen based on the predictions of netMHC.

Response Figure 38. SOPRANO ON, OFF and immune dN/dS for different immunopeptidomes. A) Original immunopeptidome used in this study based on predicted strong binders to HLA-A0201 using netMHCpan4.0. Transcripts were filtered based on global expression and similarity to IEDB T-cell positive epitope assays. B) Immunopeptidome based on predicted strong binders to HLA-A0201 using netMHCpan4.1. C) Transcripts from Immunopeptidome B filtered by patient specific expression. D) Transcripts from Immunopeptidome B including predicted weak binders. E) Transcripts from Immunopeptidome D filtered by patient specific expression. F) Original immunopeptidome used in this study based on predicted strong binders to patient specific HLA alleles using netMHCpan4.0 filtered by global expression. G) Transcripts from immunopeptidome F filtered by patient specific expression.

Response Figure 39. SOPRANO ON and OFF dN/dS for immunopeptidomes without (left) and with (right) filtering out mutations not predicted to be strong binders to the HLA-A0201 allele using netMHCpan4.1.

To examine the impact of patient-specific immunopeptidomes, we plotted the quantile-quantile distribution for the selection estimates in group A (calculated on globally expressed transcripts predicted strong binders of HLA-A0201) versus group F (calculated on globally expressed transcripts predicted strong binders of the patient specific HLA alleles) (Response Figure 3A). We observed that immune dN/dS values have almost identical distribution in both immunopeptidomes.

As noted in our previous response, HBMR scores were not available for individual samples, only the ratio of nonsynonymous events inside and outside the patient specific HLA allele (called dNHLA/dNnonHLA) were available. Consequently, we used this value from Van Den Eynden et al. to benchmark our immune dN/dS. We plotted the QQ distribution using group F against dNHLA/dNnonHLA (from Van Den Eynden et al.), and observed that SOPRANO patient-specific estimates detect stronger negative selection than the estimates provided in literature (Response Figure 3B).

Response Figure 40. QQ plot comparing A) SOPRANO HLA-A0201 versus SOPRANO patient specific immune dN/dS values and B) published patient specific immune selection metric from Van Den Eynden et al. 2019 (y-axis) and the SOPRANO patient-specific immune dN/dS (X-axis).

In summary, our results reflect the importance of the choice of the immunopeptidome and validates immune dN/dS as a metric of selection. Including neoantigens from weak binders and non-expressed regions expectedly push dN/dS towards immune neutrality. Such results support the notion that regions not subjected to immune selection should average out the selective contribution of regions under true selection. A proto-HLA consisting of the most frequent HLA alleles is only present in 2% of TCGA individuals. Consequently, every HLA-allele associated region that is not an HLA allele of the individual will mask the signal (described in lines 202-216, 1117 and 1122). We thank the reviewer for these suggestions that has significantly improved our manuscript.

The HLA-specificity offered by SOPRANO will be most relevant when measuring immune selection in individual tumours (rather than at the cohort level), however the authors do not see a correlation between immune infiltrate and immune dN/dS in their individual tumour TCGA analysis (lines 258-259).

R: As stated in our revised version, we did not find a correlation in our patient-specific TCGA comparison of individual estimates of immune selection to immune infiltrates. However, there are several caveats in this comparison given the heterogeneity associated with each individual with different tumor types, purity, variant clonality, copy number status, under/over calling escape status, and other possible

confounding biases. To tackle this limitation, we stratified individuals by immune categories described by Thorsson et al. and found that only two groups show a difference in immune dN/dS between escape strata - those with high infiltration of immune cells and high proliferation (Figures S4D and S4E). We note that annotating escape mechanisms in TCGA data is challenging, and in fact, our findings at the cohort level are striking given this level of heterogeneity. Nonetheless, to circumvent this limitation, we used an orthogonal classification of escape mechanisms provided in the literature for a subset of tumors (Lakatos cohort).

Conversely in the Lakatos et al. cohort patient specific immune dN/dS vs CD8 T cell infiltration was significantly associated in MSS escaped- tumours but not in MSS escaped+ (Figure 3F, lines 396-398).

- What differences between these analyses account for the discrepant results?
- The accuracy of patient-specific HLAs? Availability of paired RNAseq to do patient-specific expression filtering?
- These will be important limitations to acknowledge for future users of the tool.

R: We thank the reviewer for this concern and apologise for generating confusion. Although we have calculated immune dN/dS for all samples from TCGA, the Lakatos cohort is a subset of these data consisting of 879 CRC, STAD and UCEC individuals with high quality data (i.e. high purity allowing for better clonal classification), multiple annotated escape mechanisms, including escape data derived from RNA-seq, curated information about the MSI status of each patient, and clonality of each variant.

For the remainder of TCGA, we have only classified them into escape versus non-escape using the 88 gene set, which could be undercalling immune escape in the bulk of the TCGA data. However, we showed that when removing patients with a mutation in one of these escape genes, the association at the cohort level becomes significantly stronger (Figure S7), supporting the idea that escape masks signals of selection.

We have now added this comment to our results (Lines 322-328): *“When removing escaped individuals in the cohort analysis, we identified an association between immune selection and lymphocyte activity. However, as we did not observe a strong association between patient-specific estimates of immune selection and their immune infiltrates, we hypothesized that our classification of escape could underestimate true immune evasion. Consequently, we analysed a curated subset of TCGA individuals with orthogonal annotation of escape mechanisms.”*

Related to the above. While I appreciate that the authors agree with the value of a simulated dataset to explore cancer immune evolution - the response to the comment is not particularly helpful for the review process. Either the authors should provide the full details of the simulations (and preferably also include this manuscript), or else I cannot really review this aspect and it simply highlights that the authors agree more work is needed but do not feel the work should appear in this manuscript. Overall, I think simulations would be very useful in this manuscript too.

R: Our dN/dS simulator is freely available on GitHub (github.com/luisgls/dNdSsimulator). The repository also contains a tutorial that can be followed to generate individual simulations and estimate immune dN/dS. As previously noted, we have a second manuscript in revision with another journal that includes a comprehensive analysis of the model. We share that manuscript here for the reviewer's interest.

In brief, we developed a stochastic branching process of cancer evolution coupled with an agent-based model of immunoediting that simulates the acquisition of mutations, their fitness consequences, and whether the immune system will purge them from the population. We specifically simulate synonymous and nonsynonymous mutations occurring in the immunopeptidome (genomic portion that generates immunogenic events), and therefore we can calculate immune dN/dS. After 1000 simulations, we can separate these runs with zero, one, or more synonymous events inside the immunopeptidome and estimate immune dN/dS (Response Figure 4) and its corresponding confidence interval for each simulated tumor (Response Figure 5).

We observed that approximately 25%, 12% and 4% of immune edited tumours with 1, 2 or 3 synonymous mutations in the immunopeptidome, respectively, displayed a misleading immune dN/dS above 1 (Response Figure 4). The confidence interval length shrinks as more synonymous mutations are present in the IP (Response Figure 5). For instance, with 10 synonymous mutations, the length is 0.15, whereas with 100 mutations the length is 0.025.

Response Figure 41. Simulation of immune dN/dS at different number of synonymous mutations in the immunopeptidome.

Figure 42. Relationship between the length of the confidence interval and the number of synonymous mutations in the immunopeptidome.

A weakness of the paper is the lack of consistent/consensus definition of ‘escaped’ tumors. For example, tumors with HLA LOH are counted as escaped in the Lakatos et al. cohort but not identified in the TCGA cohort. Given the central argument that the fraction of escaped tumors will obscure signals of immune editing, reporting the prevalence of such escaped tumors (in each TCGA tumor type for example) would be important and is not done here.

R: We do acknowledge this limitation on our discussion and agree with the reviewer on the complexity of labelling escape. In our manuscript, we either use the mutational status of 88 genes associated to escape or the immune evasion status reported previously for a subset of TCGA patients. In Supplementary Table 1, we report the fraction of patients in each tumor type with a missense/truncating mutation in an escape gene. To overcome this limitation, we used a cohort with high quality (high purity) data from TCGA that we previously analysed (Lakatos et al.). Although it would be possible to estimate loss of heterozygosity (i.e. running LOHHLA on all bam files from TCGA) across the whole TCGA cohort, this would require an extensive amount of time, and given the same issue reported previously, we might under or over call escape by using this approach. As stated on the discussion, in the future, we need to work on best practices for comprehensively identifying possible immune evasion mechanisms and how to assess their impact on tumor evolution. In this work, we provide a simple quantitative metric that could be used as a proxy of immune escape.

In general, it would be useful to unequivocally demonstrate that the results reflect key differences in immune edited tumors, not simply mutation burden. For instance, if the authors repeat the same analysis using synonymous mutations in immune genes - do they observe a completely different pattern? Or if they use an equivalent gene set (i.e. similar size), but of random genes?

R: We thank the reviewer for this clever suggestion. To validate the impact of the “escape” genes, we followed the suggestion of the reviewer and compared individuals with a missense/truncating mutation to individuals with a synonymous mutation (and not a missense or truncating) in one of these genes. We observed that a missense or truncating mutation in an escape gene leads to significantly more truncating mutations (P-val~ 5e-03, P-val 4.48e-08, Response Figure 6, new Fig S6) compared to the control (synonymous in the same gene). Following another suggestion from the reviewer, we repeated the analysis using only escape genes that were significantly under positive selection across the TCGA cohort (Figure 2H), which revealed a similar pattern (Response Figure 7).

Interestingly, when performing the same analysis but comparing ON, OFF and immune dN/dS values, we observed that both ON- and OFF-dN/dS are lower for tumors with an escape event (missense/truncating) compared to the control (synonymous) (Response Figure 8). When using the top genes, immune dN/dS approached significance ($p=0.09$), suggesting that incorrectly labelling escape can have a major impact on the interpretation of immune selection based solely on immune dN/dS.

Figure 43. Mutation burden comparison using missense (top) or truncating (bottom) versus synonymous mutation in an escape gene.

Figure 44. Mutation burden comparison using truncating, missense versus synonymous mutations in one of the top escape genes.

Figure 45. ON, OFF and immune dN/dS comparison using truncating, missense versus synonymous mutations in one of the escape genes.

More generally, the authors classify each tumor as "escaped" if there was a missense or a truncating mutation in one of the escape genes. Using the dN/dS scores for these genes the authors should be able to estimate the likely number of false positives. Indeed, it seems likely that many of the 'escaped' tumors are not really escaped as many missense mutations will effectively be passengers. **This should be commented on.**

R: We thank the reviewer for this comment. In our previous answer, we showed that what the reviewer suggested might be an important confounding factor. However, driver discovery is complex and it was not in the scope of this manuscript and, therefore, we avoid classifying tumors based on the dN/dS significance of certain genes. We reasoned that it is also possible that some genes that are not under significant selection in a majority of samples may still have an impact in individual samples. From the list of escape genes, we now provide their dN/dS values (Supplementary Table 6) and noted that all, but two, have a dN/dS above one.

We have a full section on our discussion on the limitations of the escape genes landscape used in this study, and how to improve it (Lines 692 – 714).

The authors helpfully added the following sentence: The range of depleted neoantigens varied from 0 to 151 with a median of 8 neoantigens edited in each tumor type (Supp Table 1). Unless I am mis-interpreting these results, this suggests that neoantigen immune editing; i.e. selection against neoantigens does not have a profound impact on the accumulation of mutations. For instance, in LUAD, the tumor type with by far the strongest signals of Immune dNdS, the authors report loss of 124 neoantigens. This suggests in $>3/4$ of the cohort there are likely no neoantigens lost whatsoever. Likewise, in HNSC it is $<5\%$ of the cohort that have any evidence of loss of neoantigens. I'm not sure how this squares with the importance of immune dN/dS?

R: We thank the reviewer for this comment as the number reported is derived from the immunopeptidome, which is only a small portion of the coding genome. We have modified this sentence in the manuscript: "The range of depleted neoantigens inside the immunopeptidome varied from 0 to 151, which equates to a range of 0 to 1208 across the exome with a median of 60 neoantigens edited in each tumor type (Updated Supp Table 1). This number is a conservative estimate since it is calculated cohort-wise only within the HLA-A0201 immunopeptidome. If neoantigens accrue uniformly across the exome, then multiplying this number by ~ 8 (equivalent to the size of the whole exome divided by the size

of the immunopeptidome, and the likelihood of harbouring a neoantigen) gives more instructive insight into the level of immunoediting across a cancer genome.

We must note that when analysing patient-specific immunopeptidomes, 40% of individuals had no mutations in the immunopeptidome. Consequently, it is impossible in this patient group to differentiate whether neoantigens were previously removed or simply never occurred.

Furthermore, it would be important to directly evaluate this in the context of immune edited/unedited tumors. A figure showing the balance/mix of edited to non-edited tumors would be useful - and how many neoantigens are likely removed.

R: In Response Figure 9, we show immune dN/dS values for edited (Top, no truncating or missense mutation in one of the 88 escape genes) compared to escaped (Bottom). BRCA, CESC, CRC, HNSC, LUAD, PAAD, PRAD, STAD, and UCEC report a significantly higher immune dN/dS in the escaped compared to the edited (U-Mann Whitney), and no tumor type has a significantly higher immune dN/dS in the edited compared to the escaped, validating that escape events lead to more neutral immune dynamics. We have added the median immune dN/dS and the respective p-value for both categories to Supplementary Table 1. As individual estimates can suffer from large confidence intervals due to low or no mutations in the immunopeptidome, estimating the number of neoantigens lost globally is challenging. We have clearly stated this throughout the manuscript, and our goal was to demonstrate immune selection across the population rather than in single patients.

Figure 46. Immune dN/dS for multiple tumor types separated by escape status (TOP: No evidence of escape, Bottom: Immune escaped)

The statement ‘To date, the only FDA-approved biomarkers of response include tumor mutation burden (TMB) and mismatch repair (MMR) proficiency status’ (lines 72-74) omits the other key biomarker used to guide immunotherapy treatment decisions: immunohistochemistry PD-L1 status. This biomarker should be considered in any predictive/survival analyses including tumor types where PD-L1 status guides immunotherapy treatment.

R: We apologise for this misunderstanding but our statement “FDA-approved biomarkers” should have been replaced with: “FDA-approved genomic biomarkers” as our manuscript focuses on what can be derived from (routine) sequencing data. Nevertheless, we have now also included PD-L1 in this sentence.

Statistical tests - on a number of occasions they authors show comparisons of p-values to indicate significance. I fear on occasion this is overstating the importance of their findings. For instance, the authors state their immune dN/dS is more significant than TMB - but the authors are comparing different

sets of tumors, and moreover do not perform a formal comparison of the methods. Likewise, the authors state that “We found that our multivariable model including escape status and immune dN/dS had better prognostic value (Response fig 9, concordance 0.74, Log-rank P-value $1e-06$, and AIC 446), compared to when these variables were not included (0.72, log-rank P-value $4e-06$ and AIC 452).” while this may be true - I would assume these models are not significantly different from each other; i.e. we cannot conclude with confidence one is better than the other. In general, it would be useful to show p-values rather than p=ns

Figure 47. Multivariate cox regression models and hazard ratios. A) Basic model using variables identified in Litchfield et al. B) Same factors as in model A, but including immune dN/dS. C) Same as in model A plus escape status. D) Model A including immune dN/dS and escape status.

We have modified p-values=ns in the manuscript to reflect the numerical p-value. With respect to model selection, it is our understanding that these two statistics, p-value and an information criteria, are the accepted way to compare models. We note that the multivariate model including immune escape and

immune-dNdS was more significant, but notably, also had a lower AIC (despite being more complex) than the model without these parameters in the Hartwig tumor cohort.

Response Figure 21. Immune dN/dS calculated on strong binders predicted using netMHC4.1 and netMHC4.0. - Without further explanation, I find this figure very difficult to interpret.

R: We apologise for the lack of clarity in our response. Similar to our strategy using netMHCpan4.0, we run netMHC4.1 on all possible coding 9-mers present in the human reference proteome (obtained from Ensembl Biomart Human website) using the top 70 HLA alleles. We created a global immunopeptidome where we could extract the relevant HLA alleles for each individual immunopeptidome, and calculated immune dN/dS based on this. We have also added a section in our methods to explain alternative immunopeptidomes and consider that this figure can now be interpreted correctly.

The finding that SKCM displays significant negative selection in OFF dN/dS is exactly the reason why exploring a ratio of ratios may be misleading. I think in this respect the immune dN/dS has the potential to confuse, and it would be clearer to focus on ON dN/dS while also reporting OFF dN/dS. Immune dNdS masks important results.

R: We agree with the reviewer about the importance of reporting both ON and OFF dN/dS, and our method reports both values consistently so the user can decide on how to interpret dN/dS. It is important to note that SKCM is known to have a strong bias on a UV signature that does not depend on the trinucleotide context, but rather on the pentanucleotide context, which can lead to an overinterpretation of negative selection if not accounted for (See Martincorena et al. 2017). We refer to this issue in the method section: "We then defined immune dN/dS as the ratio between dN/dS inside (ON-target dN/dS) and outside the immunopeptidome (OFF-target dN/dS). We do so to correct for technical artefacts (germline contamination, over filtering, false positive somatic calls) that can bias up or down the neutral dN/dS expected (~1) in OFF-target regions."

The code for SOPRANO is publicly available in GitHub, however the running scripts are filled with hardcoded paths that make reproducibility particularly challenging. Also there are multiple versions and scripts available so it is unclear which should be used.

R: We thank the reviewer for noting this. We have removed old versions of the master script from the repository. We have also updated the README file, added immunopeptidomes, and clearly described the steps needed to use our tool.

Minor comments

If patients with low numbers of synonymous mutations need to be removed from the analysis due to the fact that the resulting scores are inaccurate (10% of patients in the analysis, lines 436-438), ideally this inaccuracy would be captured statistically (e.g. by wide confidence intervals around the scores) or formally addressed in another way. Documentation of this limitation must be made clear for future users.

R: We thanks the reviewer for pointing this limitation. In fact, we have added a sentence in the manuscript stating (Lines 177-180): “Moreover, when analysing patient-specific immunopeptidomes, a large proportion of individuals had no mutations in the immunopeptidome. It is therefore a challenge in these patients to differentiate whether neoantigens were previously removed or simply never occurred.” This limitation is indeed captured statistically as SOPRANO provides the confidence intervals for each individual and in TCGA a very low proportion of individuals have confidence intervals not spanning one, thereby supporting the use of cohort metrics instead of individual metrics when we come across a low number of mutations.

In addition, as previously shown in Response Figure 4, when there is one synonymous event on the immunopeptidome of immune-edited patients, the power to detect immune selection is 75%. SOPRANO will attempt to use all mutations provided and the confidence interval will be consistently reported for the user to decide on the interpretation.

Other studies exploiting dN/dS use the same strategy of putting together multiple patients to estimate selection. To our knowledge, we are the first attempting to calculate somatic selection in single patients and demonstrating a link with response to therapy.

Could further information be provided regarding how intronic mutations are used to estimate the OFF-dN/dS rates as these will have different background rates to synonymous mutations?

R: We amended the manuscript and our GitHub repository to state that this strategy is experimental. We have included a description of this feature of SOPRANO in the documentation of the tool. Specifically (Lines 960-965): “SOPRANO can leverage intronic mutations to improve the background mutation rate estimated from OFF dN/dS regions. SOPRANO simply estimates intronic mutation rate by dividing the observed number of intronic mutations by the length of the intronic regions of the gene. The resulting mutation rate is then averaged with OFF-dN/dS.” We benchmarked OFF dN/dS calculated using exonic-only versus exonic-intronic, and report no significant differences between modes (Response Figure 11).

Figure 48. OFF dN/dS comparison between Exonic Intronic and Exonic Only modes for SOPRANO.

Fig 1B does not clearly explain what HLA-alleles are used in the prototype or why these were selected (this is only explained later in the text). If the 6 HLA-alleles in the figure are the prototype, is the prototype homozygous for HLA-A0201 or is this a typo? The HLA nomenclature is different in the text than the figure (:). Fig 1C does not acknowledge that escaped tumors can be immune-edited or not.

R: We thank the reviewer for this insightful comment. We have modified the figure and caption to reflect the possible use of different immunopeptidomes based on the HLA alleles and the fact that immune edited tumors can transit towards an escape state (Response Figure 12). We have also modified the nomenclature of HLA alleles consistently throughout the manuscript.

Figure 49. Overview of immune selection calculation using SOPRANO. A) Estimates can be performed at the cohort or at the single individual level. B) In each case, it is possible to estimate immune selection on a single HLA-allele (i.e. HLA-A0201), a generic combination of HLA alleles (protoHLA) or the specific HLA-immunopeptidome. C) Two immunoediting outcomes determine the evolutionary trajectories of clonal growth, fully immune-edited tumors with strong signals of immune selection can transit towards fully immune-escaped tumors where signals are absent. D) When estimating immune selection using dN/dS, it is possible to aggregate all mutations and generate a single cohort estimate or a distribution of values per patient. In both cases, mixing escaped with edited tumors leads to loss of signals of immune selection.

There is still insufficient definition of terms in the text: ‘weak binders’ (line 202) – what thresholds were used?, ‘unexpressed regions’ (line 202) – using the cohort expression filter or patient-specific expression profiles?, global dN/dS (OFF) – does ‘global’ = ‘OFF’? (line 227 figure legend), ‘the median abundance for each immune subtype’ (line 232, Fig 2C reference in text does not correspond to figure) – Histologically scored? What was quantified? Total immune infiltrate or were specific markers for immune cell types included?, ‘copy number status’ – of the immune genes? Del and amp status or just Del?, ‘driver, global, escape, and immune loci’ (line 488) – each set of genes needs defining.

R: We have modified the manuscript accordingly in each section, and in the methods section, to provide clarification of the terminology used.

- Weak binders: (Lines 208) immunopeptidome defined as all the genomic coordinates that have an affinity of %rank < 2.
- Unexpressed regions: (Lines 207) no filtering of transcripts based on global or individual expression.
- OFF and global dN/dS: OFF is an output of SOPRANO, so we have corrected this in the manuscript. Global is an output of other tools to estimate dN/dS across the whole exome.
- Driver, Global, escape and immune loci: This nomenclature was used to differentiate between different strategies to calculate dN/dS. We have clarified this in the manuscript. In brief, immune

loci refer to the immunopeptidome. Driver and escape dN/dS refer to dN/dS calculated on a subset of genes, either driver or escape genes. (Lines 1203)

- Immune infiltrates (Lines 914): immune infiltrate estimation from published studies. Normalised immune scores for multiple cell subpopulations were obtained from three different studies (Rooney et al, Danaher et al, and Thorsson et al¹). For cohort analysis, we estimated the median of the immune score for each tumour type (Rooney et al. provides z-normalised scores based on ssGSEA, Danaher et al. provides cell type scores based on expression of specific genes, and Thorsson et al. uses published tumor immune expression signatures, later converted into gene signatures clusters).
- Copy number status: As described in the method section (Lines 929), expression data and copy number data per gene were downloaded from the GDCquery API available in the R package TCGAbiolinks. We used this metric to determine the status of the escape gene (Figures S8 and S5D). In contrast, to determine the impact of missense/truncating events in the burden of copy number alterations (Figure S5B), we obtained the aneuploidy score for each patient from Taylor et al.

MMR proficient is used in the introduction then instead of MMR deficient in the results ‘MSI rich’ tumours and later microsatellite stable (MSS) (line 326) is used which may confuse readers unfamiliar with these terms. The conventional terms are MSI-high or MMR deficient. Moreover, why were entire tumour types where MSI-high tumours can occur removed rather than only tumours known to be MSI-high?

R: We appreciate the comment from the reviewer. We have modified the introduction to refer to microsatellite instability (MSI) status instead of mismatch repair (MMR). Again, the rationale to use the entire tumor subtype instead of only tumors classified as MSI-high or MSS was to conservatively estimate the effect of immune evasion at the cohort level agnostic of the individual genetic annotation of escape. This observation provided us the link to continue exploring in more detail the effect of escape in the subsequent sections, as stated in the manuscript: “We recently reported a high frequency of immune evasion events in three tumor types where microsatellite instability (MSI) is common: colorectal (CRC), stomach (STAD), and uterine cancer (UCEC) , and so we compared the association of immune dN/dS and immune infiltrates when including or excluding these tumor types.” The detailed analysis of individual tumors classified as MSI or MSS and escape and non-escaped in these three tumor subtypes was the scope of section 3 (Figure 3).

The statement 'Given one of the major side effects of immunotherapies is autoimmunity, an upregulation of dendritic cell activity in combination with ICIs may be a good avenue to minimise autoimmunity and increase treatment efficacy' (lines 301-303) seems unrelated/unsubstantiated by the presented results

R: To avoid confusion with the results presented, and given this finding is unrelated to the main message of our study, we have removed it from the manuscript.

Fig 3C y axis should be log immune dn/ds (I think). There is a variety of immune dN/dS scales used (e.g. non-log, log, difference from 1) so these need to be clearly labeled as such. In general, for many of the figures the axis are not clearly labeled.

R: We have modified the main figures to properly label the axis. Immune dN/dS is the value not transformed unless specified in the axis which could be Immune dN/dS (Log) or d-Immune dN/dS as the difference from 1.

Decision Letter, second revision:

19th Aug 2022

Dear Dr Zapata,

I apologise for the delay in returning this decision to you. Thank you for bearing with me.

Your Article, "Immunoediting dynamics determine tumour immunogenicity and drive response to checkpoint inhibitors" has now been seen by Reviewer #3. You will see from their comments below that while they find your work of interest, some important points are raised. We are interested in the possibility of publishing your study in Nature Genetics, but would like to consider your response to these concerns in the form of a revised manuscript before we make a final decision on publication.

In addition to the points raised by Reviewer #3, we discussed as an editorial team the value of adding further simulations and your model to this manuscript. We appreciate that the model forms the basis of another paper which likely precludes inclusion in this paper. If this is no longer the case, we would be happy to include these data here. However, we would also invite you to include any further simulation data.

We therefore invite you to revise your manuscript taking into account all reviewer and editor comments. Please highlight all changes in the manuscript text file. At this stage we will need you to upload a copy of the manuscript in MS Word .docx or similar editable format.

*2) If you have not done so already please begin to revise your manuscript so that it conforms to our Article format instructions, available [here](http://www.nature.com/ng/authors/article_types/index.html). Refer also to any guidelines provided in this letter.

[redacted]

We hope to receive your revised manuscript within four to eight weeks. If you cannot send it within this time, please let us know.

Sincerely,

Safia Danovi
Editor
Nature Genetics

Reviewers' Comments:

Reviewer #3:

Remarks to the Author:

The authors have considerably improved their manuscript, and I appreciate the new additions to the manuscript. I still have a few (minor) comments:

- I think it would be more informative to provide the number of edited neoantigens per tumor rather than per tumor type. I appreciate this is a very conservative figure, but I think it is more informative than the total median for the tumor type.
- I'd suggest throughout the authors should show ON-dNdS / Off-dNdS and not simply immune dNdS.
- I think some further multiple testing correction is needed for some of the p-values (particularly in relation to the survival section).

Author Rebuttal, first revision:

Reviewers' Comments:

Reviewer #3:

Remarks to the Author:

The authors have considerably improved their manuscript, and I appreciate the new additions to the manuscript. I still have a few (minor) comments:

- I think it would be more informative to provide the number of edited neoantigens per tumor rather than per tumor type. I appreciate this is a very conservative figure, but I think it is more informative than the total median for the tumor type.

R: We thank the reviewer for this useful comment. We have now adapted our equation 2 to estimate the number of edited neoantigens per patient and have added this information to supplementary table 1 (sum of neoantigens edited per tumor type, shown below) and supplementary table 4 (detailed number of edited neoantigens per sample). We have also amended the manuscript (Lines 177-Lines188). Here is the table with the sum of neoantigens removed per tumor type and the average per patient.

SOPRANO	NeoantigensRemoved_Total	Average_perPat
ACC	45	0.5
BLCA	622	1.5
BRCA	879	0.9
CESC	326	1.1
CHOL	29	0.6
CRC	1392	2.6
DLBC	54	1.5
GBM	150	0.4
HNSC	575	1.1
KICH	28	0.4
KIRC	305	0.9
KIRP	245	0.9
LGG	256	0.5
LIHC	385	1.1
LUAD	760	1.3
LUSC	772	1.6
MESO	36	0.4
OV	427	1.0

PAAD	101	0.6
PCPG	39	0.2
PRAD	279	0.6
SARC	169	0.7
SKCM	535	1.1
STAD	731	1.7
TGCT	40	0.3
THCA	93	0.2
THYM	32	0.3
UCEC	2113	4.0
UCS	42	0.7
UVM	25	0.3

- I'd suggest throughout the authors should show ON-dNdS / Off-dNdS and not simply immune dNdS.

R: We thank the reviewer from this comment and we first have clarified that immune dN/dS is a definition based on the ON and OFF ratios (Lines 165): "We used the ratio between ON and OFF to conservatively demonstrate immune specific selection and termed such ratio "immune dN/dS"." In all analysis where we show immune dN/dS, we have ON and OFF estimations, and, when relevant we have already added both calculations to the supplementary material. Our focus is to determine selection acting inside the immunopeptidome (ON) and not selection outside (OFF). As previously discussed, the use of OFF dN/dS as a normalizing factor is to avoid technical biases rather than to provide biological meaningful interpretations. In most cohort analysis where there is no selection neither technical biases, OFF dN/dS should approximate one (Supplementary table 2). Thus, to simplify the narrative of our study, and given ON and Immune dN/dS are strongly correlated, we kept immune dN/dS in the manuscript as it was originally hoping that such clarification will be helpful for the interpretation of immune selection. If the reviewer finds that the inclusion of ON and OFF dN/dS in a particular analysis would strength our conclusions, we will review the comparison and include it.

- I think some further multiple testing correction is needed for some of the p-values (particularly in relation to the survival section).

R: The majority of plots and statistical analysis were generated using the package *ggstatsplot* that for all non-parametric tests perform Benjamini-Hochberg or Holm method correction by default (defined with parameter `p.adjust.method = "BH"`). For other figures we have confirmed that P-Values were in single test comparisons and others were properly corrected when performing pairwise comparisons or repeated against multiple variables. For the survival analysis we have added the function *pairwise_survdiff* for multiple test correction in our scripts when comparing between groups. For multiple univariate cox-regression p-values reported, we corrected the values using the Bonferroni method. If the reviewer thinks in a particular analysis that multiple test correction was not performed accordingly we would include this information.

Editorial comments:

In addition to the points raised by Reviewer #3, we discussed as an editorial team the value of adding further simulations and your model to this manuscript. We appreciate that the model forms the basis of another paper which likely precludes inclusion in this paper. If this is no longer the case, we would be happy to include these data here. However, we would also invite you to include any further simulation data.

R: We are grateful on the interest about our model of somatic evolution under immunoediting. We are currently working in a separate manuscript that precludes full inclusion of this work in the current manuscript. However, we have added a section in our methods describing the simulations performed using our freely available simulator. We describe two basic scenarios where the immune system is fully active and capable of recognizing neoantigens, and a second where neoantigens accumulate freely due to a non-active immune system, a scenario equivalent to immune escape. In the simplest scenario, we observed that median immune dN/dS under no immune selection is equal to one, and under strong immune selection is equal to 0.53 (shown in figure below). dN/dS can be affected by the allele frequency cut-off used to estimate selection, i.e. mutations in single cells or at very low population frequencies would be unseen by selective forces. We have published these results on the github repository under section: "Example for Immune active versus Immune inactive simulation".

Figure 50. Figure from github.com/luisgls/dNdSsimulator. We ran 1000 using default parameters of the single cell model simulator in two conditions: A) The immune system was fully active or B) it was inactive. We estimated immune dN/dS for each simulation and plot those that reached carrying capacity.

Decision Letter, second revision:

29th Sep 2022

Dear Dr. Zapata,

Thank you for submitting your revised manuscript "Immunoediting dynamics determine tumour immunogenicity and drive response to checkpoint inhibitors" (NG-A58772R2). It has now been seen Reviewer #3 and their comments are below. The reviewers find that the paper has improved in revision, and therefore we'll be happy in principle to publish it in Nature Genetics, pending minor revisions to satisfy our editorial and formatting guidelines.

We also wanted to flag an issue with your Supplementary information - you have files in your previous version that were not included with the current submission but are cited in the text. Please can you confirm that these remain unchanged and may be carried through? Alternatively, please could you send us the replacement files.

Sincerely,

Safia Danovi
Editor
Nature Genetics

Reviewer #3 (Remarks to the Author):

The authors have done a great job of addressing all my comments (and those of the other reviewers too).

Final Decision Letter:

25th Jan 2023

Dear Dr. Zapata,

I am delighted to say that your manuscript "Immune selection determines tumour antigenicity and influences response to checkpoint inhibitors" has been accepted for publication in an upcoming issue of Nature Genetics.

Due to the importance of these deadlines, we ask that you please let us know now whether you will be

difficult to contact over the next month. If this is the case, we ask you provide us with the contact information (email, phone and fax) of someone who will be able to check the proofs on your behalf, and who will be available to address any last-minute problems.

Your paper will be published online after we receive your corrections and will appear in print in the next available issue. You can find out your date of online publication by contacting the Nature Press Office (press@nature.com) after sending your e-proof corrections. Now is the time to inform your Public Relations or Press Office about your paper, as they might be interested in promoting its publication. This will allow them time to prepare an accurate and satisfactory press release. Include your manuscript tracking number (NG-A58772R3) and the name of the journal, which they will need when they contact our Press Office.

Please note that *Nature Genetics* is a Transformative Journal (TJ). Authors may publish their research with us through the traditional subscription access route or make their paper immediately open access through payment of an article-processing charge (APC). Authors will not be required to make a final decision about access to their article until it has been accepted. [Find out more about Transformative Journals](https://www.springernature.com/gp/open-research/transformative-journals)

Authors may need to take specific actions to achieve [compliance with funder and institutional open access mandates](https://www.springernature.com/gp/open-research/funding/policy-compliance-faqs). If your research is supported by a funder that requires immediate open access (e.g. according to [Plan S principles](https://www.springernature.com/gp/open-research/plan-s-compliance)) then you should select the gold OA route, and we will direct you to the compliant route where possible. For authors selecting the subscription publication route, the journal's standard licensing terms will need to be accepted, including [self-archiving-and-license-to-publish](https://www.nature.com/nature-portfolio/editorial-policies/self-archiving-and-license-to-publish). Those licensing terms will supersede any other terms that the author or any third party may assert apply to any version of the manuscript.

Please note that Nature Portfolio offers an immediate open access option only for papers that were first submitted after 1 January, 2021.

If you have posted a preprint on any preprint server, please ensure that the preprint details are

updated with a publication reference, including the DOI and a URL to the published version of the article on the journal website.

If you have not already done so, we invite you to upload the step-by-step protocols used in this manuscript to the Protocols Exchange, part of our on-line web resource, natureprotocols.com. If you complete the upload by the time you receive your manuscript proofs, we can insert links in your article that lead directly to the protocol details. Your protocol will be made freely available upon publication of your paper. By participating in natureprotocols.com, you are enabling researchers to more readily reproduce or adapt the methodology you use. [Natureprotocols.com](http://natureprotocols.com) is fully searchable, providing your protocols and paper with increased utility and visibility. Please submit your protocol to <https://protocolexchange.researchsquare.com/>. After entering your [nature.com](http://www.nature.com) username and password you will need to enter your manuscript number (NG-A58772R3). Further information can be found at <https://www.nature.com/nature-portfolio/editorial-policies/reporting-standards#protocols>

Sincerely,

Safia Danovi
Editor
Nature Genetics